# Learning to Discover at Test Time

**Mert Yuksekgonul** [* 1]   **Daniel Koceja** [* 1]   **Xinhao Li** [* 2]   **Federico Bianchi** [* 3]   **Jed McCaleb** [4]   **Xiaolong Wang** [2]
**Jan Kautz** [5]   **Yejin Choi** [5]   **James Zou** [† 1 3]   **Carlos Guestrin** [† 1]   **Yu Sun** [* 1 5]

## Abstract

How can we use AI to discover a new state of the art for a scientific problem? Prior work in test-time scaling, such as AlphaEvolve, performs search by prompting a frozen LLM. We perform reinforcement learning at test time, so the LLM can continue to train, but now with experience specific to the test problem. This form of continual learning is quite special, because its goal is to produce one great solution rather than many good ones on average, and to solve this very problem rather than generalize to other problems. Therefore, our learning objective and search subroutine are designed to prioritize the most promising solutions. We call this method Test-Time Training to Discover (TTT-Discover). Following prior work, we focus on problems with continuous rewards. We report results for every problem we attempted, across mathematics, GPU kernel engineering, algorithm design, and biology. TTT-Discover sets the new state of the art in almost all of them: (i) Erdős' minimum overlap problem and an autocorrelation inequality; (ii) a GPUMode kernel competition (up to $2\times$ faster than prior art); (iii) past AtCoder algorithm competitions; and (iv) denoising problem in single-cell analysis. Our solutions are reviewed by experts or the organizers. All our results are achieved with an open model, OpenAI gpt-oss-120b, and can be reproduced with our code, in contrast to previous best results that required closed frontier models. Our test-time training runs are performed using Tinker, with a cost of only a few hundred dollars per problem.

---

[*]Core contributors. [†]Joint advising. [1]Stanford University [2]UC San Diego [3]Together AI [4]Astera Institute [5]NVIDIA. Correspondence to: Mert Yuksekgonul <merty@stanford.edu>, Yu Sun <yusun@cs.stanford.edu>.

*Proceedings of the $43^{rd}$ International Conference on Machine Learning*, Seoul, South Korea. PMLR 306, 2026. Copyright 2026 by the author(s).

## 1. Introduction

To solve hard problems, humans often need to try, fail, stumble upon partial successes, and then learn from their experiences. Consider your first really hard programming assignment. You read the textbook and trained yourself on the book exercises, but this assignment just asked for so much beyond the basics in the book. You tried to guess the solution, but these attempts merely produced small signs of life. So you had to take a deep breath and learn from your failed attempts, which made your future attempts more intelligent. Finally, after hours of trying and learning, you understood the new ideas behind the assignment. And indeed, the next attempt worked!

In this example, the assignment was hard because it required new ideas beyond your training data (the text and exercises in the book). Now consider using AI to solve scientific discovery problems. This goal is even harder: By definition, discovery problems require ideas not only beyond the model's training data but also all existing knowledge of humanity. And out-of-distribution generalization is no easier for AI than for humans (Miller et al., 2020; Hendrycks et al., 2021; Ribeiro et al., 2020; Koh et al., 2021).

To offset this hardness, prior work has focused on test-time search in the solution space by prompting a frozen LLM to make many attempts, similar to how we tried to guess the solution to the assignment. In particular, evolutionary search methods, such as AlphaEvolve, store past attempts in a buffer and use them to generate new prompts via handcrafted and domain-specific heuristics (Novikov et al., 2025; Lange et al., 2025; Sakana AI, 2026; Yuksekgonul et al., 2025). While these prompts can help the LLM improve previous solutions, the LLM itself cannot improve, similar to a student who can never internalize the new ideas behind the assignment.

The most direct way for the LLM to improve is through learning. And indeed, while both learning and search scale well with compute (Sutton, 2019), learning has often superseded search in the history of AI for hard problems such as Go and protein folding (Silver et al., 2017; Jumper et al., 2021). We believe that this observation from history is still relevant today, as we scale compute at test time. So we continue to train the LLM, while it attempts to solve this

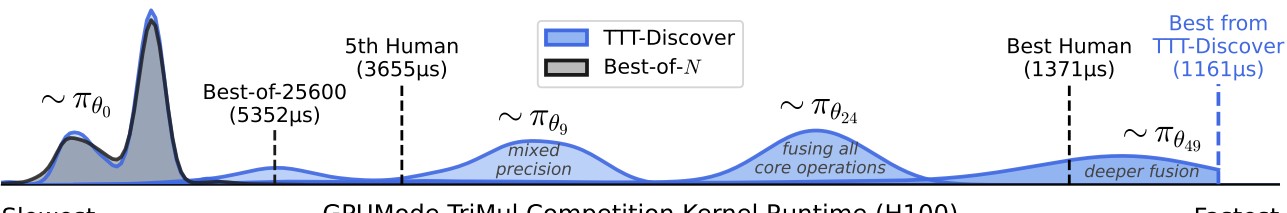

*Figure 1.* TTT-Discover continues to train an LLM on a single problem at test time. $\pi_{\theta_i}$ denotes the policy with the updates weights at test-time training step $i$. We plot the reward distribution at step $0, 9, 24,$ and $49$ (final), recorded while test-time training for the GPUMode TriMul competition. We generate 512 solutions at each step. As training progresses, the LLM generates better solutions that ultimately surpass the prior art (best human). For comparison, we plot the reward distribution of best-of-$N$ with the same total sampling budget.

very test problem. And these attempts, in turn, provide the most valuable training data: Recall that the test problem was hard because it was out-of-distribution. Now we have a data distribution specific to this problem.

At a high level, we simply perform Reinforcement Learning (RL) in an environment defined by the single test problem, so any technique in standard RL could be applied. However, our goal has two critical differences from that of standard RL. First, our policy only needs to solve this single problem rather than generalize to other problems. Second, we only need a single best solution, and the policy is merely a means towards this end. In contrast, the policy is the end in standard RL, whose goal is to maximize the average reward across all attempts. While the first difference is a recurring theme in the field of test-time training (Sun et al., 2020), the second is unique to discovery problems.

To take advantage of these differences, our learning objective and search subroutine strongly favor the most promising solutions. We call this method Test-Time Training to Discover (TTT-Discover). We focus on problems with continuous rewards, in mathematics (§4.1), GPU kernel engineering (§4.2), algorithm design (§4.3), and biology (§4.4). We report results for every problem we attempted, and TTT-Discover sets the new state of the art in almost all of them, using only an open model.

## 2. Preliminaries

All methods in this paper, including the baselines, share a common goal: Given a scientific problem at test time, the goal is to discover a new state-of-the-art solution with an LLM policy $\pi_\theta$, whose weights $\theta$ have already been trained (at training time). To formalize this goal, we first introduce how each scientific problem defines an environment, i.e., a Markov Decision Process (§2.1), which can then be used for search (§2.2) and learning (§3).

### 2.1. Discovery Problem

Our definition of the environment follows prior work in test-time scaling, such as AlphaEvolve (Novikov et al., 2025): A scientific problem comes in the form of a text description $d$, which we always feed as context to the policy. We define a state $s$ as a candidate solution, such as a kernel implementation of the PyTorch code in $d$. In our applications, the problem description also induces a continuous reward function $R(s) \in \mathbb{R}$, such as the inverse runtime of the kernel.

We denote $s_{\text{sota}}$ as the best-known solution among all existing candidates, and $r_{\text{sota}} = R(s_{\text{sota}})$ as the best-known reward. And in case there is no existing solution, $s_{\text{sota}}$ can be the empty string `<empty>`. For example, $s_{\text{sota}}$ can be the kernel currently at the top of the leaderboard. These notations allow us to formalize the notion of a discovery:

**Definition** (Discovery). *A discovery is an event where a state $s$ is found such that $R(s) > r_{sota}$. The larger the difference, the more significant the discovery.*

Under this formalism, we define a *discovery problem* as finding such a state $s$ with large $R(s) - r_{\text{sota}}$ within the environment defined by the scientific problem.

To produce a better solution, both search and learning methods use the LLM policy to generate an action $a \sim \pi_\theta(\cdot \mid d, s)$, where the choice of the initial solution $s$ (e.g., $= s_{\text{sota}}$) is an important part of the method's design. Similar to the reward function, the transition function $(s, a) \rightarrow s'$ of the environment is also induced by the problem description. Here, we consider only a single timestep since state reuse, which we will introduce soon, effectively subsumes multiple timesteps.

In all our applications, a valid action contains a piece of code and optionally some thinking tokens. For coding problems (e.g., kernel engineering), the environment produces $s'$ by simply parsing the code out of $a$. For problems in mathematics, the environment also needs to execute the code in $a$ after it is parsed. Table 4 provides an overview of the environments for all our applications.

## 2.2. Search Methods

The simplest search method, known as Best-of-$N$, samples i.i.d. rollouts from $\pi_\theta$:

**Best-of-$N$:**  $s = s_{\text{sota}}$ or `<empty>`,  $a_i \sim \pi_\theta(\cdot \mid d, s)$,

where the subscript, $i = 1, \ldots, N$, denotes the index of the rollout. By using $i$ instead of $t$ for the index, we indicate that the rollouts here are independent. One reasonable choice of the initial state $s$ is $s_{\text{sota}}$, assuming that a previous solution exists. But $s_{\text{sota}}$ might be too strong a prior towards exploitation. For example, conditioning on $s_{\text{sota}}$ might prevent the policy from exploring very different, but more promising directions that would ultimately produce better solutions under a large compute budget. To address this concern, we usually set $s = $ `<empty>`, the empty (or trivial) solution.

On the other hand, the policy might also explore a promising direction using $s = $ `<empty>`, but fail to fully exploit it. One technique to address this opposite concern is *state reuse*, which warm starts the policy with some of the previous solutions. Specifically, it maintains a buffer $\mathcal{H}_i$ of the previous solutions, and samples the initial solution $s_i$ from $\mathcal{H}_i$ using a search heuristic, `reuse`, which favors high-reward solutions but still assigns nontrivial likelihood to low-reward ones:

**State reuse:**  $s_i \sim \texttt{reuse}(\mathcal{H}_i)$,
$$a_i \sim \pi_\theta(\cdot \mid d, s_i), \mathcal{H}_{i+1} = \mathcal{H}_i \cup \{(s'_i, r_i)\}.$$

When we reuse a previous solution $s'_i$, we have effectively added an extra timestep to its trajectory.

Prior work, such as AlphaEvolve (Novikov et al., 2025), also reuses the actions, which can contain thinking tokens and intermediate results (e.g., code for math problems) that are not part of the states. As a consequence, the `reuse` heuristic also needs to convert the information from previous actions into natural language context $c_i$ that can be ingested by the LLM policy:

**State-action reuse:**  $s_i, c_i \sim \texttt{reuse}(\mathcal{H}_i)$,
$$a_i \sim \pi_\theta(\cdot \mid d, s_i, c_i), \mathcal{H}_{i+1} = \mathcal{H}_i \cup \{(s_i, a_i, s'_i, r_i)\}.$$

Prior work (Novikov et al., 2025; Lange et al., 2025; Yuksekgonul et al., 2025; Liu et al., 2024b) refers to state-action reuse as *evolutionary search*, because the `reuse` heuristic usually involves sophisticated designs motivated by evolution, including hand-crafted operations for mutation and cross-over, and domain-specific measurements of fitness and diversity.

## 3. Learning to Discover at Test Time

So far, the policy's experience with the test problem can only improve the next prompt $(d, s_i, c_i)$, but not the policy

$\pi_\theta$ itself, since $\theta$ remains frozen. We use this experience to improve the policy in an online fashion, by training $\pi_\theta$ on its own search attempts accumulated in the buffer $\mathcal{H}_i$.

Algorithm 1 outlines our method, where the two key subroutines to instantiate are `reuse` and `train`.

---

**Algorithm 1** TTT-Discover

1: **Input:** problem description $d$ and initial policy $\pi_{\theta_0}$.
2: $R, T = \texttt{get\_env}(d)$ {$d$ induces the reward and transition functions of the environment (§2.1)}
3: $\mathcal{H}_0 = \{(\texttt{<empty>}, R(\texttt{<empty>}), \{\})\}$ {Initialize buffer with the empty solution (§2.2)}
4: **for** $i = 0, 1, \ldots, N - 1$ **do**
5: $\quad s_i, c_i \sim \texttt{reuse}(\mathcal{H}_i)$ {Sample initial state and context with a `reuse` heuristic}
6: $\quad a_i \sim \pi_{\theta_i}(\cdot \mid d, s_i, c_i)$ {Sample action from policy}
7: $\quad s'_i = T(a_i)$ {Transition to next state}
8: $\quad r_i = R(s'_i)$ {Evaluate reward of next state}
9: $\quad \mathcal{H}_{i+1} = \mathcal{H}_i \cup \{(s_i, a_i, s'_i, r_i)\}$
10: $\quad \theta_{i+1} = \texttt{train}(\theta_i, (d, s_i, c_i, a_i, r_i))$ {Improve the model weights with `train`}
11: **return** $s_{i^*}$, where $i^* = \arg\max_{i=0,1,\ldots,N-1} r_i$

---

### 3.1. Naive RL at Test Time

Algorithm 1 falls under the formulation of reinforcement learning (RL). A natural baseline is to use a standard RL algorithm:

$$\texttt{train:} \quad \theta_{i+1} = \theta_i + \eta \nabla_\theta \mathbb{E}_{a \sim \pi_{\theta_i}(\cdot \mid s)}\big[R(s, a)\big],$$
$$\texttt{reuse:} \quad \mathcal{H}_i = \delta_{\text{init}}.$$

i.e., optimize for expected reward with no reuse, where $\delta_{\texttt{<empty>}}$ is a delta distribution with mass only on the initial state `<empty>`. We will use $\theta_i$ to denote the model weights for rollout $i$. We can straightforwardly apply popular RL algorithms, such as PPO or GRPO (Schulman et al., 2017; Guo et al., 2025), only in the environment defined by the single problem.

However, these algorithms are designed with the standard RL problem in mind. Discovery problems have important distinctions from standard RL problems.

In standard RL problems, the goal is to find a policy that maximizes the expected reward. This policy is to be deployed repeatedly in the same environment. The primary artifact is the policy.

In discovery problems, the goal is to find a single state that improves upon the state-of-the-art. We do not care about the average performance. There is no separate deployment

phase and thus the policy need not maintain robust performance in many states it may encounter starting from the same initial state distribution. In fact, a policy can have very low expected reward, so long as it reaches a new state-of-the-art once.

Due to these differences, the naive RL instantiation has important shortcomings.

**Objective function.** Naive RL optimizes average performance, and is indifferent to the state of the art. In discovery, however, success is determined by the maximum, and whether it improves upon the state of the art. Consider a kernel engineering problem where the state-of-the-art runtime is $2000\,\mu s$. Achieving $1900\,\mu s$ would require substantial optimization and perhaps a breakthrough. Yet, without complicated reward shaping, both would receive nearly the same reward.

**Short effective horizon.** Starting each attempt from scratch limits how far the policy can reach in an attempt. Reusing a previous solution effectively adds extra timesteps, extending the horizon. As a result, more complex solutions can emerge during training. In standard RL, a fixed initial state distribution makes sense as the policy must perform robustly from states it will encounter at deployment. Discovery has no such deployment phase.

**Exploration.** Exploration requires care at two levels. Optimizing for expected reward, the policy can collapse to safe, high-reward actions rather than risky ones that might achieve discovery. At the reuse level, naive prioritization can over-exploit a few promising states at the expense of diversity.

### 3.2. TTT-Discover

To address these shortcomings, we introduce two simple components.

**Entropic objective.** We define the entropic objective that favors the maximum reward actions:

$$J_\beta(\theta) = \mathbb{E}_{s\sim\text{reuse}(\mathcal{H})}\left[\log\mathbb{E}_{a\sim\pi_\theta(\cdot|s)}\left[e^{\beta(s)R(s,a)}\right]\right],$$

$$\nabla_\theta J_\beta(\theta) = \mathbb{E}_{\substack{s\sim\text{reuse}(\mathcal{H})\\ a\sim\pi_\theta(\cdot|s)}}\left[w_{\beta(s)}(a)\,\nabla_\theta\log\pi_\theta(a\mid s)\right],$$

$$w_{\beta(s)}(a) = \frac{e^{\beta(s)R(s,a)}}{\mathbb{E}_{\pi_\theta(\cdot|s)}\left[e^{\beta(s)R(s,a)}\right]}.$$

where we also shape advantages with a KL penalty: $A(a;s) = w_{\beta(s)}(a) - 1 - \lambda\log\frac{\pi_\theta(a|s)}{\pi_{\theta_0}(a|s)}$ (Schulman et al., 2017; Zhang et al., 2025b; Tang & Munos, 2025), and $-1$ is the baseline since $\mathbb{E}[w_{\beta(s)}] = 1$. Concurrent work (Jiang et al., 2025) also explored the entropic objective $J_\beta$ to maximize the pass@k performance for (training-time) RL with binary reward problems.

As $\beta\to\infty$, the entropic objective tends to the $\max$, which is intuitively what we want. However, too large $\beta$ early in training causes instabilities, while too small later makes advantages vanish as even smaller improvements become harder. Empirically, we found that setting a constant $\beta$ that works well across different tasks is challenging. Therefore, different than (Jiang et al., 2025), we set $\beta(s)$ adaptively per initial state by constraining the KL divergence of the induced policy; see Appendix B.1 for details.

**PUCT.** We select initial states using a PUCT-inspired rule (Rosin, 2011; Silver et al., 2016; 2017; 2018). Each state $s$ is scored by $Q(s) + c \cdot P(s) \cdot \sqrt{1+T}/(1+n(s))$, where $Q(s)$ is the maximum reward among states generated when the initial state was $s$ (or $R(s)$ if $s$ has not yet been selected). $P(s)$ is proportional to $s$'s rank in the buffer sorted by reward, $n(s)$ counts how many times $s$ or its descendants have been expanded, and $T$ is the total number of expansions, and $c$ is the exploration coefficient.

Rather than the mean (as in prior work), we use the maximum reward of children in $Q(s)$: we care about the best outcome starting from a state, not the average. The prior $P(s)$ captures the intuition that high-reward states are more likely to yield high-reward children—e.g., a fast kernel is more likely to seed a faster kernel than a slow one—while the exploration bonus prevents over-exploitation by keeping under-visited states as candidates. See Appendix B.2 for implementation details.

**Test-time Training to Discover.** TTT-Discover is simply a combination of $J_{\beta(s)}$ as the objective function, and PUCT.

**Implementation Details.** We run TTT-Discover with gpt-oss-120b (Agarwal et al., 2025) on Tinker (Lab, 2025) for 50 training steps. We use LoRA (Hu et al., 2022) with rank 32. At each step, we generate a batch of 512 rollouts, with 8 groups of 64 rollouts each. Each group of rollouts is generated using the same context and initial state selected from the reuse buffer. We use the entropic objective, and apply importance sampling ratio correction to the gradients due to the sampler/learner mismatch in the RL infrastructure (Yao et al.). We do not take any off-policy steps, i.e., take 1 gradient step on the entire batch. Assuming an average prompt length of 3000 tokens and 16000 sampling tokens on average, a training run with 50 steps and 512 rollouts costs around \$500 on Tinker. More training details are available in the Appendix B.

## 4. Applications

We evaluate TTT-Discover on problems in GPU kernel engineering, mathematics, algorithm design, and biology. We report our performance on every task we attempted. Besides potential impact, we pick domains with 2 criteria. First, we pick domains where we can compare our performance to

human experts. This is possible, for example, by comparing to the best submissions in human engineering competitions, or to the best results reported in academic papers. We also want to compare to AI baselines. As we discuss below, mathematics and algorithm design are discovery domains where prior work recently made progress (Novikov et al., 2025; Imajuku et al., 2025; Sakana AI, 2026).

In every application, we report the best known human results and the best known AI results. Importantly, we always report the Best-of-$N$ baseline that matches the sampling budget and the model that TTT-Discover uses. That is, since we perform 50 steps with 512 rollouts per step, and compare to the Best-of-25600 baseline. Although Best-of-25600 matches the total number of rollouts used by TTT-Discover, TTT-Discover incurs additional compute from its training updates. To account for this, we also evaluate a cost-matched Best-of-$N$ baseline and report results in Appendix G. For a closest evolutionary algorithm baseline, we also run OpenEvolve (Sharma, 2025), an open-source version of AlphaEvolve (Novikov et al., 2025), with the same 25600 sampling budget. We use the same context window budget and the Tinker client for gpt-oss-120b throughout the experiments. We caution that the context window limit led to a large number of rollouts in OpenEvolve to be truncated before the model completes its response, as OpenEvolve's prompts grow very large in length. However, to stay faithful to their implementation, we did not modify their prompts or rollouts.

## 4.1. Mathematics

We explore multiple open problems in mathematics. These are often problems where even small numerical improvements carry weight, since each result potentially rules out families of approaches and extends the frontier of what is mathematically known. Here, proofs are by construction: one can construct a concrete mathematical object – a step function or a sequence – that certifies, e.g., a bound can be achieved, making these problems amenable to search.

**Environment:** State $s$ is a construction, specifically a step function represented as an array, to certify a proof. The action $a$ consists of thinking tokens followed by code that either constructs a new $s$ or modifies an existing one. The dynamics execute the parsed code to produce the next state: $s' = \text{Python}(\text{Parse}(a))$. The reward is the bound certified by $s'$, or zero if $s'$ is not valid. Often, actions involve optimization algorithms to improve the constructions.

Throughout mathematics applications, we initialize the buffer with random states. Specifically, initial states are sampled uniformly at random within the problem's valid range. For each action, we give a 10-minute limit to execute the code given by the action. In the case of a timeout, the action gets a reward of 0. For minimization problems (cer-

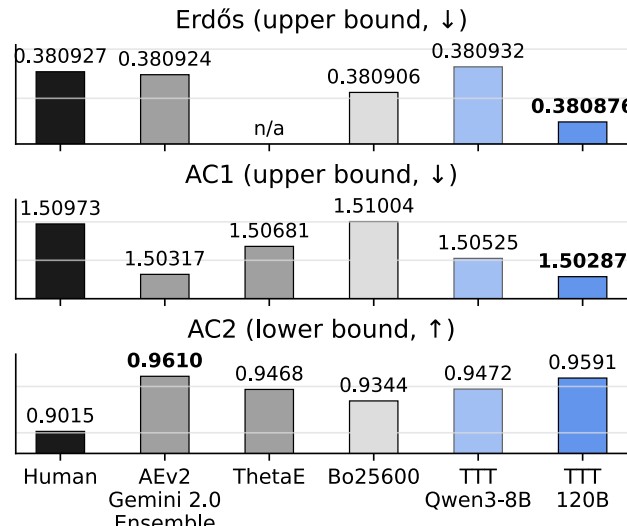

*Figure 2.* Results in mathematics problems. In the Erdős' Minimum Overlap Problem and First Autocorrelation Inequality (AC1), TTT-Discover sets the new state-of-the-art. We also report TTT-Discover with Qwen3-8B, for a better comparison to ThetaEvolve. Notable, TTT-Discover with Qwen3-8B outperforms not only ThetaEvolve, baselines including AlphaEvolve which uses Gemini-2.0 family models for the autocorrelation inequalities. Our state-of-the-art constructions are released and can be validated in our codebase. Full numbers are reported in Table 7.

tifying upper bounds), we set the reward proportional to $1/\text{bound}$ for the certified bound, and otherwise we set it proportional to bound. We report further details about the environment and the prompts we use in Appendix C.

**Previous state-of-the-art.** Such problems are recently explored in (Georgiev et al., 2025; Novikov et al., 2025). We report both the best known human results, and the recent progress by AI: AlphaEvolve (Novikov et al., 2025), AlphaEvolve V2 (Georgiev et al., 2025) which was released around 6 months after AlphaEvolve, ShinkaEvolve (Lange et al., 2025), and ThetaEvolve (Wang et al., 2025a).

We select one representative problem from each area in AlphaEvolve (Novikov et al., 2025): Erdős' minimum overlap problem (combinatorics), autocorrelation inequalities (analysis), circle packing (geometry).

**Circle Packing:** We report results for this problem in Appendix C.1, and use it mostly for another reference to compare to concurrent work.

### 4.1.1. ERDŐS' MINIMUM OVERLAP PROBLEM

This is a classic problem in combinatorial number theory, posed by Erdős in 1955, with connections to the distribution of sequences and difference sets. Partition $\{1, 2, \ldots, 2n\}$ into two sets $A$ and $B$ of equal cardinality $n$. Define $M_k$ as the number of solutions to $a_i - b_j = k$ for $a_i \in A, b_j \in B$,

and let $M(n) = \min_{A,B} \max_k M_k$ over all partitions. The problem is to bound $c = \lim_{n \to \infty} M(n)/n$. Bounds before AlphaEvolve were $0.379005 < c < 0.380927$, with the upper bound due to Haugland (Haugland, 2016) and the lower bound due to (White, 2023). AlphaEvolve (Novikov et al., 2025) improved the upper bound to $0.380924$.

Following (Novikov et al., 2025), we optimize step functions $f$ describing the density of $A$ throughout $[1, 2n]$. Due to a result of Swinnerton-Dyer (Haugland, 2016), density functions yield valid upper bounds on $\lim M(n)/n$ without constructing explicit partitions for large $n$. Validity checks require $f(x) \in [0, 1]$ and $\int f = 1$.

**Results.** We improve the upper bound on Erdős' Minimum Overlap Problem to $0.380876$, surpassing AlphaEvolve's recent construction with $0.380924$ (Novikov et al., 2025). Our improvement over AlphaEvolve is 16 times larger than AlphaEvolve's improvement over the previous state-of-the-art. Unlike AlphaEvolve's symmetric construction, our method discovered a 600-piece asymmetric step function. Surprisingly, the Best-of-25600 baseline also improved upon the AlphaEvolve construction. Insights on the discovered algorithm are reported in Appendix C.3.

### 4.1.2. AUTOCORRELATION INEQUALITIES

Autocorrelation inequalities are motivated by additive combinatorics (Barnard & Steinerberger, 2020). Improving these inequalities tightens a constant that propagates into sharper limits on how large a set can be while still avoiding repeated additive patterns (a central theme in additive combinatorics). Similar to the Erdős' minimum overlap problem, we will construct a step function $f$ to certify bounds.

**First autocorrelation inequality.** For nonnegative $f$ supported on $[-1/4, 1/4]$, define $C_1$ as the largest constant such that $\max_{|t| \leq 1/2}(f * f)(t) \geq C_1 \left( \int f \right)^2$ holds for all such $f$. The goal is to certify the tightest *upper bound* on $C_1$; any valid construction $f$ certifies $C_1 \leq \frac{\|f*f\|_\infty}{\|f\|_1^2}$. Until early 2025, the best known upper bound was $C_1 \leq 1.50973$ (Matolcsi & Vinuesa, 2010). AlphaEvolve improved this to $C_1 \leq 1.5053$, and AlphaEvolve V2 further improved it to $C_1 \leq 1.50317$, and ThetaEvolve refined AlphaEvolve's construction to get $C_1 \leq 1.50314$.

**Second autocorrelation inequality.** For nonnegative $f$, define $C_2 = \sup_{f \geq 0} \frac{\|f*f\|_2^2}{\|f*f\|_1 \|f*f\|_\infty}$. The problem is to certify the tightest known *lower bound* on $C_2$; any valid construction $f$ with ratio $r$ certifies $C_2 \geq r$. The best human bound was $C_2 \geq 0.8892$ (Matolcsi & Vinuesa, 2010). AlphaEvolve first improved this to $C_2 \geq 0.8962$, (Boyer & Li, 2025) improved this to $0.9015$, and AlphaEvolve V2 further improved it to $C_2 \geq 0.9610$ using a 50,000-piece step function.

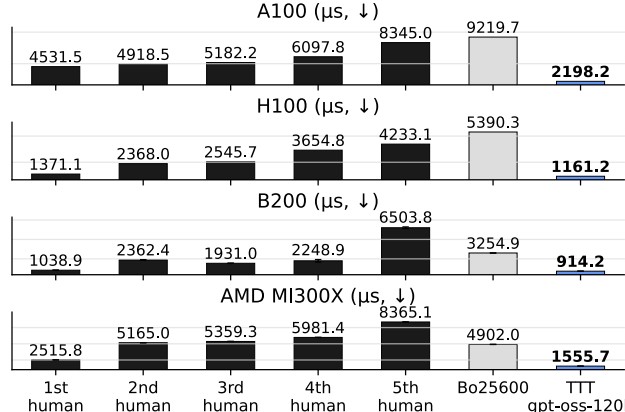

*Figure 3.* Results in GPU kernel engineering. TTT-Discover achieves state-of-the-art runtimes for the TriMul competition across all evaluated architectures (H100, A100, B200, MI300X). Full numbers are reported in Table 1.

**Results.** We improved the best known upper bound to prove $C_1 \leq 1.50286$, with a 30000-piece step function. The comparisons are reported in Figure 2. The previous state-of-the-art, ThetaEvolve, achieved their result by refining the AlphaEvolve V2 construction. In contrast, TTT-Discover found a new construction by starting from scratch. We visualize our and prior works' step functions in Figure 7. In the second autocorrelation inequality, we have not made a discovery. Our best construction certified a bound of $0.959$, where the AlphaEvolve construction had certified a tighter lower bound of $0.961$. Insights on the algorithms are reported in Appendix C.2.

For a better comparison to the concurrent work, ThetaEvolve, we also report TTT-Discover with Qwen3-8B (Yang et al., 2025). The Qwen3-8B variant they used, `DeepSeek-R1-0528-Qwen3-8B` that was released by DeepSeek, is not available on Tinker. Thus, we used the original Qwen model (`Qwen/Qwen3-8B`) that was reportedly worse than the DeepSeek variant. ThetaEvolve reports using 65 steps with 512 rollouts (32 groups of 16 rollouts) each, however we do not modify our hyperparameters otherwise and keep 50 steps of 512 rollouts each. For both inequalities, TTT-Discover with Qwen3-8B certified tighter bounds than ThetaEvolve, using a worse model and a smaller sampling budget.

**Expert Review**. We provide the review by Prof. Davide Torlo of Università di Roma La Sapienza in Appendix C.

### 4.2. Kernel Engineering

GPU kernels are the computational foundation of modern AI, every forward pass and backward pass ultimately executes as kernel code on hardware. We apply our method to GPU kernel optimization, where a new state-of-the-art kernel is a faster implementation than existing ones.

GPUMODE is an open community for kernel development that also hosts competitions for domain experts. We test our method on two competitions: TriMul (triangular matrix multiplication), a core primitive in AlphaFold's architecture (Jumper et al., 2021), and DeepSeek MLA (Multi-head Latent Attention), a key component in DeepSeek's inference stack (Liu et al., 2024a). Each GPU type for the TriMul competition (NVIDIA H100, A100, B200, AMD MI300X) has a separate leaderboard For The MLA competition, there is only an MI300X leaderboard.

As these competitions were conducted earlier, we retrospectively evaluate our performance while respecting competition standards. We prefer GPUMODE because their leaderboards are well-tested through human competitions with a robust evaluation harness (Zhang et al., 2025a), and their benchmarks avoid signal-to-noise issues where simple operations or small inputs cause overheads to dominate runtime.

**Environment:** The state $s$ is a GPU kernel code. The action $a$ consists of thinking tokens followed by kernel code written in Triton (Tillet et al., 2019). The dynamics parse the code from the action: $s' = \texttt{Parse}(a)$. For the initial state, we provide unoptimized kernels, detailed in Appendix D. The reward is proportional to the inverse of the geometric mean of runtimes on a fixed set of input shapes (following the leaderboard), or zero if the kernel fails correctness checks or times out. We evaluate runtime remotely on Modal to scale and ensure consistent hardware conditions. For TriMul, we evaluate the runtime only on H100s during training, even though we still evaluate the generated kernels for A100, B200, and MI300X for final report. Since MI300X is not available on Modal, for MLA-Decode we use H200s, and hope the kernels generalize to MI300X. Further details about the prompts and environments are in Appendix D.

**Results.** We report the runtime of the best kernels and the baselines in Figure 3. Our TriMul kernels achieve state-of-the-art across the board in all GPU types. For A100s, our best kernel is $50\%$ faster than the top human kernel, even though our reward function did not time the kernels on A100s. We uniformly achieve $> 15\%$ improvement over the best human submissions for all GPU types. Finally, we submit to the official TriMul A100/H100 leaderboard[1]. To compare TTT-Discovery with general post-training on a distribution of related tasks, we also report comparison with a baseline post-trained on a general GPU kernel optimization dataset, KernelBench, in Appendix H. We also show the improvement in solution validity rate across TTT-Discover steps on the TriMul task in Section I.

The discovered kernels for Trimul identify heavy memory

_______________________
[1]For TriMul B200/MI300X and MLA-Decode MI300X tasks, due to an infra problem on GPU Mode's server, we could not submit to the official leaderboard.

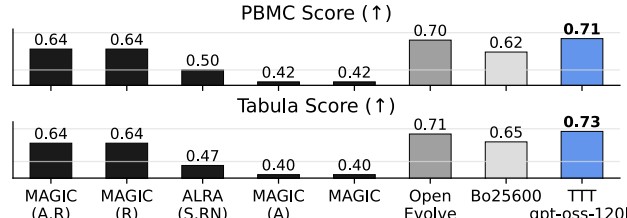

*Figure 4.* Results in single-cell RNA-sequencing denoising. TTT-Discover improves upon MAGIC, on the OpenProblems benchmark. Our discovered algorithm achieves state-of-the-art MSE scores on held-out datasets (PBMC and Tabula). Full numbers are reported in Table 3.

I/O incurred by frequent elementwise operations as a major bottleneck and leverage triton for operation fusion. Please refer to Appendix D.2 for an in-depth analysis, and Appendix D for more results on the MLA Decode kernels.

**Expert Review** Verbatim reviews from the GPUMode organizers are provided in D.

### 4.3. Algorithm Engineering

Due to space limitations, we include a summary of our results in competitive algorithm engineering tasks, and give all details in Appendix E. AtCoder Heuristic Contest (AHC) is a series of programming competitions focused on optimization problems drawn from real-world industrial challenges (AtCoder Inc., 2025), attracting hundreds of participants including industry experts. We attempted to evaluate on two past contests, ahc039 and ahc058. We select ahc039 because ShinkaEvolve (Lange et al., 2025) reported a solution that would have placed 2nd, and ahc058 because Sakana AI's ALE-Agent achieved the first-ever AI victory in an AHC (Sakana AI, 2026). We use the evaluation harness from ALE-Bench (Imajuku et al., 2025). We use the public test case generator to create local tests, select our best-performing algorithm, and submit it to be scored on the official platform. We report results in Table 2. For both competitions, if we had submitted during competition time, our algorithms would have gotten the 1st place.

### 4.4. Single Cell Analysis

Single-cell RNA-sequencing (RNA-seq) aims to help us understand how organisms work and get sick by resolving biology at the level of individual cells; measuring which genes each cell is using to reveal cell types, states, and how they change. Practically, it isolates single cells, tags their mRNA with a Unique Molecular Identifier (UMI), sequences it, and outputs a per-cell gene-by-count table. RNA-seq protocols suffer from measurement noise in the observed UMI counts. Thus, denoising algorithms significantly increases the realized value of expensive experiments. Each sequencing run costs thousands of dollars, and better denoising methods

reduce the need for deeper sequencing.

We apply our method to one of the recent benchmarks Open-Problems (Luecken et al., 2025), an important set of open problems for single-cell analysis. We use the denoising task therein. (Batson et al., 2019) demonstrated that partitioning the observed molecules of a single dataset into training and test sets via binomial sampling and evaluating the denoised training set against the held-out test counts provides a proxy for accuracy against true expression values, providing an evaluation framework without requiring external ground truth data.

**Environment.** The state $s$ is an algorithm implementation. The action $a$ consists of thinking tokens followed by code. The dynamics parse the code from the action: $s' = \text{Parse}(a)$. The benchmark evaluates denoising quality using two complementary metrics: mean squared error (MSE) in log-normalized space, which measures overall reconstruction accuracy, and Poisson negative log-likelihood, which assesses how well the denoised counts match the statistical properties expected of count data. In our context, the reward is the MSE score, or zero if it violates constraints we add for the Poisson score or the algorithm exceeds the time limit of 400 seconds. The Denoising benchmark offers 3 datasets: PBMC, Pancreas, and Tabula Muris Senis Lung, in order of size. We train our policy by using Pancreas in our environment, and ultimately performance is reported by running the algorithm on the held out PBMC and Tabula.

**Previous state-of-the-art.** We report the state of the art as described by the OpenProblems (Luecken et al., 2025) benchmark. The best result was provided by MAGIC (Van Dijk et al., 2018) using an approximate solver and reversed normalization. MAGIC is a well known technique, frequently used in the literature (Youssef et al., 2024; Venkat et al., 2025), the only method different from MAGIC that provides good performance is ALRA (Linderman et al., 2022), ranked third. We also compare with OpenEvolve and Best-of-25600.

**Results.** The improved function obtained via TTT-Discover shows consistent improvements on both datasets (see Figure 4). TTT-Discover is initialized with MAGIC code. TTT-Discover adds gene-adaptive transform ensembling, low-rank SVD refinement, and log-space polishing steps that directly optimize the benchmark metric.

**Expert Review.** We provide the review of Prof. Eric Sun of MIT in Appendix F.

### 4.5. Ablations

We have three sets of ablations. First, we ablate the design choices for the `train` method, while keeping our `reuse` method, PUCT, fixed. We test (i) TTT with entropic objective using constant $\beta = 2$ ((Jiang et al., 2025)),

(ii) TTT with no entropic objective (expected reward), (iii) No TTT (only reuse). Second, we ablate the choice of the `Reuse method`, while keeping our `train` method, TTT with entropic objective using adaptive $\beta$, fixed. We replace PUCT with (i) $\epsilon-$greedy reuse with $\epsilon = 0.1$ as this is perhaps the most naive reuse method, and (ii) no reuse. Finally, we report the naive RL baseline, where we use the expected reward objective with no reuse, and the Best-of-25600 baseline.

For each ablation, we report the runtime of the best kernel found in Table 6, and the reward distribution in Figure 5. The rewards distributions and best kernel runtimes are computed with our evaluator, not the leaderboard.

Only the full TTT-Discover algorithm achieves the best performance in the TriMul competition. When using a constant $\beta$, the improvements diminish later in the training. Using the expected reward objective, improvements are slower overall. Without any test-time training, both the mean reward and the max reward stagnates. $\epsilon$-greedy reuse works reasonably well, especially with an early lucky kernel. In early experiments with other applications, the lack of exploration was also a bigger problem than it is in kernel engineering tasks. Naive RL and no reuse make minimal improvements.

We refer to the Appendix for two ablations: Section K shows the effect of base model choice on the Erdős and TriMul tasks and Section J shows that TTT-Discover does not cause significant catastrophic forgetting when trained on the AC1 and AC2 tasks.

## 5. Related Works

Here, we provide an overview of the closest related works. In Appendix Extended Related Works section A, we provide a broader overview of the field.

Evolutionary search with frozen LLMs is the most direct precursor to our setting: these methods treat the LLM as an immutable generator and improve solutions by recycling prior attempts via hand-designed mutation/crossover, diversity, and fitness heuristics (Novikov et al., 2025; Georgiev et al., 2025; Lange et al., 2025; Sakana AI, 2026; Yuksekgonul et al., 2025; Liu et al., 2024b; Sharma, 2025). Our work differs mainly by training the model at test time.

Three closest and concurrent works perform test-time training: MiGRATe (Phan et al., 2025), ThetaEvolve (Wang et al., 2025a), and EvoTune (Surina et al., 2025). Both combine per-instance RL updates with various replay/reuse mechanisms, and typically use PPO/GRPO/DPO-style updates for LMs (Schulman et al., 2017; Guo et al., 2025; Rafailov et al., 2023). Relative to these, our contribution is to tailor both the learning objective and the reuse rule to the discov-

ery goal, rather than largely standard RL or evolutionary baselines; we also test in more realistic discovery tasks with human expert baselines. In an earlier paper (Bello et al., 2016), they train a neural policy with policy gradients to directly output solutions to combinatorial problems like TSP, using (negative) tour length as reward, and they study both training across many instances and per-instance learning at test time.

## 6. Future Work

The current form of our method targets discovery problems with continuous, verifiable rewards; the most important next step is extending test-time training to sparse/binary rewards and to partially or non-verifiable domains. Even within the verifiable setting, we did minimal tuning of the reuse mechanism: there is substantial room to explore better (Q)-estimation (e.g., uncertainty-aware or multi-step credit), richer priors over states, and more principled exploration strategies.

## Impact Statement

This paper presents work whose goal is to advance the field of Machine Learning. We are performing scientific discovery, so all results should be treated as preliminary and subject to careful human review; we also solicited expert reviews to help assess validity, limitations, and potential impacts.

Our single cell analysis study is an experimental application demonstrating TTT-Discover's ability to find algorithms that excel on specific benchmarks. While our discovered algorithm outperforms existing methods on the OpenProblems denoising benchmark, benchmark metrics are inherently incomplete and do not guarantee biological validity for downstream tasks.

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

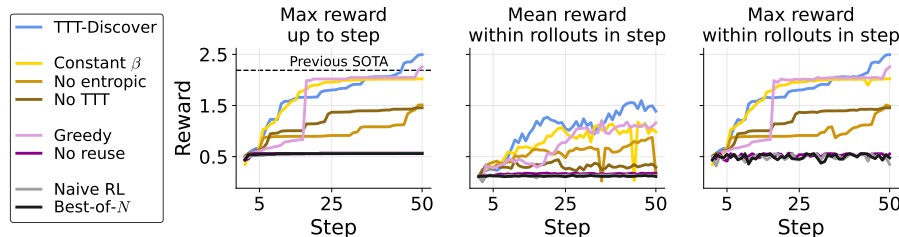

*Figure 5.* Reward distributions for each ablation. We match the sampling budget across all ablations. We sample 512 rollouts in each step. For example, for Best-of-$N$, we have $N = 50 \times 512 = 25600$ rollouts.

| Method | Model | TriMul ($\mu s$, $\downarrow$) | | | |
|---|---|---|---|---|---|
| | | **A100** | **H100** | **B200** [95% CI] | **AMD MI300X** [95% CI] |
| 1st human | – | 4531.5 | 1371.1 | 1038.9 [1016.3, 1061.6] | 2515.8 [2510.9, 2520.7] |
| 2nd human | – | 4918.5 | 2368.0 | 2362.4 [2335.7, 2389.1] | 5165.0 [5163.0, 5167.0] |
| 3rd human | – | 5182.2 | 2545.7 | 1931.0 [1910.9, 1951.1] | 5359.3 [5343.5, 5375.1] |
| 4th human | – | 6097.8 | 3654.8 | 2248.9 [2089.4, 2408.4] | 5981.4 [5978.4, 5984.4] |
| 5th human | – | 8345.0 | 4233.1 | 6503.8 [6400.5, 6607.1] | 8365.1 [8347.7, 8382.5] |
| Best-of-25600 | gpt-oss-120b | 9219.7 | 5390.3 | 3254.9 [3252.5, 3257.4] | 4902.0 [4897.6, 4906.4] |
| TTT-Discover | gpt-oss-120b | **2198.2** | **1161.2** | **914.2** [907.3, 921.1] | **1555.7** [1550.8, 1560.6] |

*Table 1.* For the TriMul competition, we train a single model using H100 runtime as the reward function and report the runtime of the single best kernel. We only trained using H100 for evaluating kernels during training. The generated kernels happened to generalize to other GPU types. We also report the top-5 human submissions in the leaderboard for comparison (each GPU type has its own top-5 human submissions). For A100 and H100, we submitted to the official leaderboard and report the runtime returned. For B200 and MI300X, we could not submit our kernels due to an infra problem on GPU Mode's server, and therefore conduct 10 trials for each kernel and report mean and confidence intervals using the same infrastructure as GPUMode, verified by the organizers. Our state-of-the-art kernels are released and can be validated in our codebase.

| Method | Model | Geometry (ahc039) | Scheduling (ahc058) |
|---|---|---|---|
| 1st human | – | 566, 997 | 847, 674, 723 |
| 2nd human | – | 557, 212 | 846, 938, 871 |
| 3rd human | – | 554, 334 | 846, 350, 877 |
| 4th human | – | 552, 933 | 845, 489, 747 |
| 5th human | – | 549, 746 | 845, 324, 831 |
| ALE-Agent | Ensemble (see caption) | 550, 647 | 848, 373, 282 |
| ShinkaEvolve | Ensemble (see caption) | 558, 026 | n/a |
| Best-of-25600 | gpt-oss-120b | 554, 171 | 772, 429, 752 |
| TTT-Discover | gpt-oss-120b | **567, 062** | **848, 414, 228** |

*Table 2.* Results in two AtCoder Heuristic Competitions. We train our models with local public tests, and submit the best program we get during training to the official submission platform. Our algorithms are released and can be validated in our codebase. ALE-Agent uses Gemini-2.5 Pro for ahc039, and Gemini-3 Pro Preview high and gpt-5.2-high for ahc058. ShinkaEvolve uses an ensemble of gpt-5, gpt-5-mini, Gemini-2.5 Pro and Flash, Claude Sonnet 4, o4-mini.

## A. Extended Related Works

In this section, we first provide a broad overview of continual learning and test-time training, using some of the exposition in (Tandon et al., 2025). Then towards the end of §A.2, we discuss the most relevant work on test-time training: MiGrATe (Phan et al., 2025) and ThetaEvolve (Wang et al., 2025a). Finally, we discuss two pieces of work with tangential formulations: RL on a single training problem that is not the test problem (Wang et al., 2025b) (§A.3), and RL on the entire test set (Zuo et al., 2025) (§A.4).

### A.1. Continual Learning

Most of today's AI systems remain static after deployment, even though the world keeps changing. The high-level goal of continual learning is to enable AI systems to keep changing with the world, similar to how humans improve throughout their

| Method | Model | PBMC | | | Tabula | | |
|---|---|---|---|---|---|---|---|
| | | Score (↑) | MSE (↓) | Poisson (↓) | Score (↑) | MSE (↓) | Poisson (↓) |
| MAGIC (A, R) | – | 0.64 | 0.19 | 0.05 | 0.64 | 0.18 | 0.03 |
| MAGIC (R) | – | 0.64 | 0.19 | 0.05 | 0.64 | 0.18 | 0.03 |
| ALRA (S, RN) | – | 0.50 | 0.26 | 0.05 | 0.47 | 0.27 | 0.03 |
| MAGIC (A) | – | 0.42 | 0.19 | 0.16 | 0.40 | 0.18 | 0.12 |
| MAGIC | – | 0.42 | 0.19 | 0.16 | 0.40 | 0.18 | 0.12 |
| OpenEvolve | gpt-oss-120b | 0.70 | 0.16 | 0.05 | 0.71 | 0.15 | 0.03 |
| Best-of-25600 | gpt-oss-120b | 0.62 | 0.20 | 0.05 | 0.65 | 0.18 | 0.03 |
| TTT-Discover | gpt-oss-120b | **0.71** | **0.15** | **0.05** | **0.73** | **0.14** | **0.03** |

*Table 3.* Denoising task for single cell data analysis. We report the score (mean of normalized MSE and Poisson scores), MSE, and Poisson metrics for each dataset. Our state-of-the-art algorithm is released and can be validated in our codebase. MAGIC (A, R) = MAGIC (Van Dijk et al., 2018) approximate with reversed normalization; MAGIC (R) = MAGIC with reversed normalization; ALRA (Linderman et al., 2022) (S, R) = ALRA sqrt norm with reversed normalization; MAGIC (A) = MAGIC approximate.

lives (Hassabis et al., 2017; De Lange et al., 2021).

Conventionally, continual learning as a research field has focused on learning from a *distribution* that gradually changes over time (Lopez-Paz & Ranzato, 2017; Van de Ven & Tolias, 2019; Hadsell et al., 2020). For example, one could update a chatbot model every hour using new knowledge from the Internet, while typical use cases of the model may require knowledge from both the past and the present (Scialom et al., 2022; Ke et al., 2023; Wang et al., 2024). More formally, at each timestep, we sample new training and test data from the current distribution, update our model using the new training data, and then evaluate it on all the test data up to the current timestep. Under this setting, most algorithms focus on not forgetting the past when learning from the present (Santoro et al., 2016; Li & Hoiem, 2017; Kirkpatrick et al., 2017; Gidaris & Komodakis, 2018).

## A.2. Test-Time Training

The algorithmic framework of test-time training has the same high-level goal as continual learning, but it focuses on two aspects where human learning stands out from the forms of continual learning in the conventional literature.

First, each person has a unique brain that learns within the context of their individual life. This personalized form of continual learning is quite different from, for example, the chatbot model that is fine-tuned hourly using the latest information available worldwide. While such a model does change over time, it is still the same at any given moment for every user and every problem instance.

Second, most human learning happens without a boundary between training and testing. Consider your commute to work this morning. It is both "testing" because you did care about getting to work this very morning, and "training" because you were also gaining experience for future commutes. But in machine learning, the train-test split has always been a fundamental concept.

The concept of test-time training is introduced to realize these two special aspects of human learning. *Training* typically involves formulating a learning problem (such as empirical risk minimization) and then solving it. Following (Sun et al., 2023), *test-time training* is defined as any kind of training that formulates a potentially different learning problem based on each individual test instance.

This concept has a rich history in AI. A well-known example in NLP is dynamic evaluation, pioneered by Mikolov et al. (Mikolov et al., 2013) and extended by Krause et al. (Krause et al., 2018). In computer vision, early examples have also emerged in applications such as face detection (Jain & Learned-Miller, 2011), video segmentation (Mullapudi et al., 2018), super-resolution (Shocher et al., 2018), and 3D reconstruction (Luo et al., 2020). Next, we discuss three popular forms of test-time training today, with an emphasis on their connections to each other and to historical examples.

### A.2.1. TTT ON NEAREST NEIGHBORS: LARGER EFFECTIVE CAPACITY

One simple form of test-time training was called locally weighted regression in the 1970s (Stone, 1977; Cleveland, 1979), local learning in the 1990s (Bottou & Vapnik, 1992), and KNN-SVM in the 2000s (Zhang et al., 2006): Given a test instance, find its nearest neighbors in the training set, and then train (or fine-tune) the model on these neighbors before making a prediction. This procedure can significantly increase the effective capacity of the model; for example, it allows a linear

model to fit a highly nonlinear ground truth (Stone, 1977).

This simple form captures one of the key intuitions of test-time training. In the conventional view of machine learning, a model, once trained, no longer changes at test time. As a consequence, it must prepare to be good at all possible inputs in the future. This task can be very hard, because being good at all possible futures limits the model's capacity to be good at any particular one. But only one future is actually going to happen. So why not train our model once this future happens?

Recently, (Hardt & Sun, 2023) extended this idea to modern language models and observed a similar benefit of larger effective model capacity after test-time training, and (Hübotter et al., 2024) further improved these results through better strategies for neighbor selection. In addition, (Hübotter et al., 2025) showed that test-time training on neighbors from the training set is also effective with RL for reasoning tasks, and (Bagatella et al., 2025) developed the same idea for visual-motor tasks.

### A.2.2. TTT FOR NOVEL INSTANCES: BETTER GENERALIZATION

As models become larger today, their competence is often limited not by their capacity, but by the amount of available training data, especially when they need to generalize to novel test instances that are "out-of-distribution". In this case, it is even harder to prepare for all possible test instances in the future, especially the novel ones, with a static model. But once a specific test instance is given, we can use it to generate relevant data, which we can then use for training (Sun et al., 2020). In other words, the "neighbors" for TTT do not have to come from the training set; they can also be generated on-the-fly.

Since the test instance is unlabeled, one way to make it useful for training is through self-supervision, which generates new pairs of inputs and labels for an auxiliary task such as masked reconstruction (e.g., BERT (Devlin et al., 2018) and MAE(He et al., 2021)). While the auxiliary task is different from the main prediction task, improving performance in one can help the other through their shared representations. This form of TTT can significantly improve generalization under distribution shifts (Sun et al., 2020; Gandelsman et al., 2022).

Recently, TTT has been an important part of AlphaProof (Hubert et al., 2025), which achieved IMO silver-medal standard in 2024. Given each test problem, their system first generates a targeted curriculum of easier problems by prompting a language model, and then performs reinforcement learning on the generated data. Another recent work, Akyurek et al. (Akyürek et al., 2024), found TTT effective for few-shot reasoning tasks such as ARC-AGI. Their system generates augmentations of the few-shot demonstrations in the test problem then performs supervised learning.

MiGrATe (Phan et al., 2025) and ThetaEvolve (Wang et al., 2025a) are two concurrent works that share our high-level idea of performing RL at test time on a single problem. MiGrATe combines on-policy and off-policy RL and tests on simpler environments such as word search. ThetaEvolve is more similar to our work: it uses OpenEvolve, a variant of AlphaEvolve, for state-action reuse. Both methods use GRPO variants for training. Compared to ThetaEvolve, TTT-Discover using the same model and compute budget still produces significant improvements (Table 7), which we attribute to our entropic objective and PUCT-based reuse instead of more complicated and brittle heuristics in evolutionary algorithms.

### A.3. RL on One Example

One Example RL (Wang et al., 2025b) is relevant as they also train on a single problem. To be specific, they train on one example from a dataset, such as the MATH training set. They show that a policy trained on one such problem with RL generalizes to other problems in the same dataset. In contrast, TTT-Discover trains on the test problem itself, where the goal is not to generalize but to solve this specific problem.

### A.4. RL on the Test Set

TTRL (Zuo et al., 2025) trains on an entire test set of problems using majority voting as pseudo-labels for reward estimation. In contrast, TTT-Discover trains on a single test problem with a continuous verifiable reward, where the goal is not to improve average performance across a set of problems but to find one exceptional solution.

### A.5. RL for specific tasks

There is a large body of work on fine-tuning LLMs with RL for targeted domains. For mathematical reasoning, Wizard-Math (Luo et al., 2023) uses step-level PPO with evolved instructions, DeepSeekMath (Shao et al., 2024) introduces GRPO to improve competition-level math without a critic network. For code generation, CodeRL (Le et al., 2022) trains an actor-critic

| Problem | State $s$ | Action $a$ | Transition | Reward $R(s)$ |
|---|---|---|---|---|
| Erdős Minimum Overlap | Step function | Thinking | $s' = \texttt{Python}(\texttt{Parse}(a))$ | 1/Upper bound |
| Autocorr. Inequality (1st) | certificate | tokens and | | 1/Upper bound |
| Autocorr. Inequality (2nd) | | code | | Lower bound |
| Kernel Engineering | Kernel code | Thinking | | 1/Runtime |
| Algorithm Competition | Algorithm code | tokens and | $s' = \texttt{Parse}(a)$ | Test score |
| Single Cell Analysis | Analysis code | code | | 1/MSE |

*Table 4.* Overview of the science and engineering problems in our paper, and the environments they induce (§2.1). Note that the reward is 0 if $s$ fails validity checks.

| Parameters | Value |
|---|---|
| **General** | |
| Model | gpt-oss-120b (Agarwal et al., 2025) |
| Reasoning effort | high |
| **Rollout** | |
| Context window | 32768 tokens |
| Sampling temperature | 1.0 |
| Maximum tokens to generate | 32768-prompt length |
| Prompt length + thinking token limit | 26000 |
| Teacher forcing | *... okay, I am out of thinking tokens. I need to send my final message now.* |
| **Training** | |
| Batch size | 512 (8 groups with 64 rollouts each) |
| Training steps | 50 |
| Optimizer | Adam (Kingma & Ba, 2014), lr $4 \times 10^{-5}$, $\beta_1 = 0.9$, $\beta_2 = 0.95$, $\epsilon = 10^{-8}$ |
| LoRA (Hu et al., 2022) rank | 32 |
| KL coefficient ($\lambda$) | $\{0.01, 0.1\}$ |
| Objective | Entropic; adaptive $\beta(s_{\text{init}})$ with KL constraint $\gamma = \ln(2)$. |
| **Reuse** | |
| PUCT $c$ (exploration coefficient) | 1.0 |
| Further details | Appendix B |

*Table 5.* We fix these hyperparameters across all applications.

with unit test feedback, and RLTF (Liu et al., 2023) extends this with finer-grained compiler signals. Self-improvement methods such as STaR (Zelikman et al., 2022) and ReST (Gulcehre et al., 2023) iteratively fine-tune on the model's own correct outputs in math and coding. TTT-Discover differs in that it performs RL on the model weights at test time for a single problem instance.

# B. Training details

We set the reasoning effort to high. The context window of gpt-oss-120b is limited to $32{,}768$ tokens on Tinker. Thus, each rollout stops when the context window is exhausted or the LM produces the end of sequence token. In most domains, we limit the total length of the prompt and the thinking tokens to 26000 tokens, so as to leave enough tokens to generate the final response, e.g., to allow generating longer algorithm code. We enforce this by token forcing the model to generate its final response. All hyperparameters reported in Table 5, and are fixed unless otherwise stated.

Our hyperparameters are fixed throughout almost all experiment. For almost all applications we used a KL penalty coefficient of 0.1. For algorithm engineering, we used a KL coefficient of 0.01. We present details on our objective function and the reuse algorithm below.

## B.1. Entropic utility objective

We define the entropic utility objective explored also in the concurrent work (Jiang et al., 2025):

$$J_\beta(\theta; s) := \log \mathbb{E}_{\tau \sim \pi_\theta(\cdot | s)} \left[ e^{\beta r(\tau; s)} \right].$$

The gradient of this objective yields

$$\nabla_\theta J_\beta(\theta; s) = \mathbb{E}_{\tau \sim \pi_\theta(\cdot | s)} [\nabla_\theta \log \pi_\theta(\tau | s) \, w_\beta(\tau | s)], \quad w_\beta(\tau | s) = \frac{e^{\beta r(\tau; s)}}{\mathbb{E}_{\pi_\theta}[e^{\beta r(\tau; s)}]}, \quad A_\beta(\tau | s) = w_\beta(\tau | s) - 1,$$

since $\mathbb{E}_{\pi_\theta}[w_\beta(\tau | s)] = 1$, we get $A_\beta$ as the mean baselined advantage. The remaining question is how to set $\beta$. (Jiang et al., 2025) recommends value $\beta = 2$, yet we found it tricky to set it. Later in the training, improvements become harder, and unless $\beta$ is adjusted carefully advantages can become very small. Early in the training, a large $\beta$ can cause instabilities.

**Adaptive $\beta$.** Define the auxiliary tilted distribution induced by the entropic weights,

$$q_\beta(\tau | s) = \frac{\pi_\theta(\tau | s) \exp(\beta r(\tau; s))}{\mathbb{E}_{\pi_\theta}[\exp(\beta r(\tau; s))]}, \qquad w_\beta(\tau | s) = \frac{q_\beta(\tau | s)}{\pi_\theta(\tau | s)}.$$

Then $w_\beta$ is exactly the density ratio that appears in the entropic policy-gradient update, so $\beta$ controls the effective step size induced by this reweighting. We choose $\beta(s)$ by enforcing a KL budget on the auxiliary distribution,

$$\mathrm{KL}\big(q_{\beta(s)}(\cdot | s) \,\|\, \pi_\theta(\cdot | s)\big) = \gamma,$$

analogous to Relative Entropy Policy Search, where the temperature is set by an exponential tilt under a relative-entropy constraint (Peters et al., 2010). In words, $\beta(s)$ is increased only until the KL budget is exhausted, ensuring the induced reweighting, and hence the update, does not move too far from $\pi_\theta(\cdot | s)$. We fix $\gamma = \ln 2$ throughout our experiments.

**Batch estimator.** Given $N$ rollouts from the same $s$ with rewards $\{r_n\}_{n=1}^N$, the empirical sampling distribution is uniform on the batch, $u(n) = 1/N$. The induced reweighting on the batch is

$$q_\beta(n) = \frac{e^{\beta r_n}}{\sum_{m=1}^N e^{\beta r_m}},$$

and we set $\beta(s)$ by solving the weight-concentration constraint

$$\mathrm{KL}(q_\beta \| u) = \sum_{n=1}^N q_\beta(n) \log\big(N q_\beta(n)\big) = \gamma$$

via simple bisection search over $\beta \geq 0$. With $\hat{\beta}(s)$, we compute LOO entropic advantages using $r_{\max} = \max_n r_n$, and an $\epsilon$ in the denominator for numerical stability:

$$\hat{Z}_{-n} = \frac{1}{N-1} \sum_{m \neq n} \exp(\hat{\beta}(s)(r_m - r_{\max})), \qquad A_n = \frac{\exp(\hat{\beta}(s)(r_n - r_{\max}))}{\hat{Z}_{-n} + \varepsilon} - 1.$$

**Discussion.** States where improvements are consistently small (e.g. high-value / near-goal states) tend to make the batch weights $q_\beta(n)$ less peaky for a given $\beta$, so the constraint typically permits a larger $\beta(s)$. In contrast, states that occasionally yield a few very large improvements (often earlier in training or low-value states with large headroom) make $q_\beta$ concentrate quickly as $\beta$ grows; the same KL budget then forces a smaller $\beta(s)$, preventing the update from being dominated by a handful of outlier trajectories while still preferring better-than-average rollouts. Finally, this estimator is invariant to shifting or scaling the reward by a constant, i.e., $r(\tau)$ and $r'(\tau) = wr(\tau) + b$ yield the same advantage for $w \in \mathbb{R}^+$ and $b \in \mathbb{R}$.

## B.2. PUCT Prioritization

We maintain an archive $\mathcal{H}_t$ of previously discovered states $s$ with reward $R(s) \in \mathbb{R}$. To choose the next start state, we score each $s \in \mathcal{H}_t$ by a PUCT-inspired rule, analogous to applying PUCT at a virtual root whose actions correspond to selecting a start state from the archive (Rosin, 2011; Silver et al., 2016; 2017; 2018):

$$\text{score}(s) = Q(s) + c \cdot \text{scale} \cdot P(s) \frac{\sqrt{1+T}}{1+n(s)},$$

where $n(s)$ is a visitation count, $T$ is the number of expanded parents so far, $c > 0$ is an exploration coefficient, and scale $= R_{\max} - R_{\min}$ is the reward range over the archive. The prior $P(s)$ is a linear rank distribution:

$$P(s) = \frac{|\mathcal{H}_t| - \text{rank}(s)}{\sum_{s' \in \mathcal{H}_t}(|\mathcal{H}_t| - \text{rank}(s'))},$$

where $\text{rank}(s) \in \{0, \ldots, |\mathcal{H}_t| - 1\}$ orders states by descending reward (rank 0 is the best state). The term $Q(s)$ uses the best one-step reachable reward $m(s)$:

$$Q(s) = \begin{cases} m(s) & n(s) > 0 \\ R(s) & n(s) = 0 \end{cases}.$$

After expanding parent $p$ and observing its best child reward $y = \max_{s' \in \text{Child}(p)} R(s')$, we update:

$$
\begin{aligned}
m(p) &\leftarrow \max(m(p), y) && \text{(direct parent only)} \\
n(a) &\leftarrow n(a) + 1 \quad \forall a \in \{p\} \cup \text{Anc}(p) && \text{(backprop visitation)} \\
T &\leftarrow T + 1.
\end{aligned}
$$

For the archive update, we keep the top-2 children per expanded parent (largest $R$) before inserting, then enforce a global size constraint by retaining the top-1000 states in $\mathcal{H}_t$ by $R$, while always keeping the initial seed states.

**Comparison to AlphaZero PUCT.** AlphaZero's PUCT operates over a tree of state-action edges, selecting actions via $a = \arg\max_a[Q(s,a) + c \cdot P(s,a) \cdot \sqrt{\sum_b N(s,b)}/(1 + N(s,a))]$, where $Q(s,a)$ is the mean value of simulations through edge $(s,a)$, $P(s,a)$ is a learned policy prior, and $N(s,a)$ counts visits to that edge (Silver et al., 2017; 2018). Our formulation differs in three ways: (i) $Q(s)$ tracks the maximum child reward rather than the mean, favoring optimistic expansion; (ii) $P(s)$ is a rank-based prior over archived states rather than a learned action distribution; (iii) visitation counts backpropagate to all ancestors, so expanding any descendant reduces the exploration bonus for the entire lineage.

| | **train** | **reuse** | **Best runtime** $(\mu s, \downarrow)$ |
|---|---|---|---|
| Best Human Kernel | – | – | 1371.1 |
| TTT-Discover | TTT with adaptive entropic | PUCT | **1203.10** |
| Ablations for `train` | TTT with constant $\beta$ entropic | PUCT | 1483.83 |
| | TTT with expected reward (no entropic) | PUCT | 1985.67 |
| | No TTT | PUCT | 2060.70 |
| Ablations for `reuse` | TTT with adaptive entropic | $\epsilon$-greedy | 1328.89 |
| | TTT with adaptive entropic | no reuse | 5274.03 |
| Naive Test-time RL | TTT with expected reward | no reuse | 5328.73 |
| Best-of-$N$ | no TTT | no reuse | 5352.36 |

*Table 6.* Ablation results for the TriMul GPUMode competition where we time the kernels with an H100 GPU. We report the best kernel we get in each run. We report the reward distributions across steps in Figure 5.

## C. Mathematics

| Method | Model | Erdős' (↓) | AC1 (↓) | AC2 (↑) |
|--------|-------|-----------|---------|---------|
| best human | – | 0.380927 | 1.50973 | 0.9015 |
| AlphaEvolve (Novikov et al., 2025) | Gemini-2.0 Pro + Flash | 0.380924 | 1.50530 | 0.8962 |
| AlphaEvolve V2 (Georgiev et al., 2025) | Gemini-2.0 Pro + Flash | 0.380924 | 1.50317 | **0.9610** |
| ThetaEvolve (Wang et al., 2025a) | R1-Qwen3-8B | n/a | 1.50681 | 0.9468 |
| ThetaEvolve w/ SOTA reuse (1.50317) | R1-Qwen3-8B | n/a | 1.50314 | n/a |
| OpenEvolve (Sharma, 2025) | gpt-oss-120b | 0.380965 | 1.50719 | 0.9449 |
| Best-of-25600 | gpt-oss-120b | 0.380906 | 1.51004 | 0.9344 |
| TTT-Discover | Qwen3-8B | 0.380932 | 1.50525 | 0.9472 |
| TTT-Discover | gpt-oss-120b | **0.380876** | **1.50287** | 0.9591 |

*Table 7.* Results in mathematics problems. In the Erdős' Minimum Overlap Problem and First Autocorrelation Inequality (AC1), TTT-Discover sets the new state-of-the-art. We also report TTT-Discover with Qwen3-8B, for a better comparison to ThetaEvolve. Notably, TTT-Discover with Qwen3-8B outperforms not only ThetaEvolve, baselines including AlphaEvolve which uses Gemini-2.0 family models for the autocorrelation inequalities. Our state-of-the-art constructions are released and can be validated in our codebase.

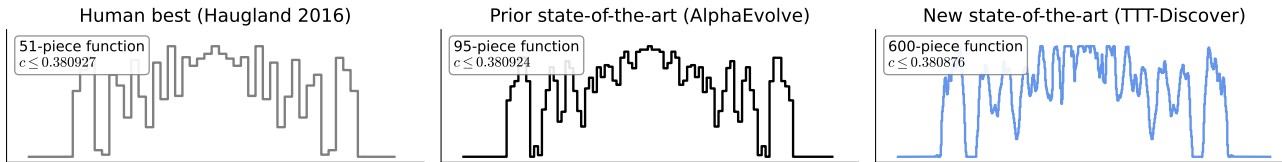

*Figure 6.* We show the normalized step functions including the prior state-of-the-art from AlphaEvolve. The step function $f(x)$ is the limiting density of set $A$. Unlike the previous state-of-the-art, the solution from TTT-Discover is an asymmetric construction. TTT-Discover found a 600-piece step function, while AlphaEvolve construction was 95-piece. The best human result was a 51-piece construction (Haugland, 2016).

> **Human Expert Review — Prof. Davide Torlo (Università di Roma La Sapienza)**
>
> Erdős' minimum overlap problem and the autocorrelation inequalities are classical problems in combinatorics with applications in, among others, discrepancy theory, combinatorial optimization, and signal analysis. Both problems can be formulated as min–max problems, in which the minimization is taken over a class of functions with bounded norm, while the maximization is performed over a set of evaluation points. Closed-form solutions are not known; instead, only lower and upper bounds can be derived. Obtaining sharp bounds for these problems remains a challenging mathematical task and is essential for improving our understanding and resolution of such questions. The upper bounds obtained by TTT-Discover for the Erdős' minimum overlap and the AC1 autocorrelation problems are achieved by specific piecewise-constant functions. It is straightforward to verify that the provided functions give bounds that improve upon the state of the art: one simply evaluates the quantity of interest and its maximum over a discrete set of points determined by the step size of the piecewise-constant functions, and checks that the corresponding norm constraints are satisfied.

### C.1. Circle Packing

In Circle packing, the goal is to maximize the sum of radii of $n$ non-overlapping circles packed inside a unit square. We follow the setup from prior work (Novikov et al., 2025; Georgiev et al., 2025). The state $s$ is a list of circle centers and radii. The action $a$ consists of thinking tokens followed by Python code that optimizes circle positions and radii. The reward is the sum of radii achieved for valid packings, and 0 otherwise. We present the results below mostly for comparison purposes, as several recent works on evolutionary algorithms reported their performance using this task.

Table 8 shows results. TTT-Discover with Qwen3-8B matches the best known constructions for both $n = 26$ and $n = 32$. We make no improvements here, but include these results for completeness. The algorithms found by TTT-Discover are presented in Appendix C.1. Algorithms initialize circles in staggered or hexagonal grid arrangements, then refine positions and radii using sequential least squares programming with boundary and pairwise non-overlap constraints. This solution is a lot simpler than recent work, such as ShinkaEvolve (Lange et al., 2025), especially in terms of initialization, where their

solution uses an initialization based on simulated annealing algorithm, while TTT-Discover initializes only with a simple geometric arrangement.

| Method | Model | $n = 26$ (↑) | $n = 32$ (↑) |
|---|---|---|---|
| AlphaEvolve (Novikov et al., 2025) | Gemini-2.0 Pro + Flash | 2.635862 | 2.937944 |
| AlphaEvolve V2 (Georgiev et al., 2025) | Gemini-2.0 Pro + Flash | 2.635983 | 2.939572 |
| ShinkaEvolve (Lange et al., 2025) | Ensemble (see caption) | 2.635982 | n/a |
| ThetaEvolve (Wang et al., 2025a) | R1-Qwen3-8B | 2.635983 | n/a |
| TTT-Discover | Qwen3-8B | 2.635983 | 2.939572 |

*Table 8.* Results for circle packing. ShinkaEvolve uses an ensemble of Claude Sonnet-4, gpt-4.1, gpt-4.1-mini, gpt-4.1-nano, o4-mini.

**Circle Packing ($n = 26$)**

```python
import numpy as np
from scipy.optimize import minimize

def run_packing():
    n = 26
    initial_centers = []
    initial_radii = []

    # Adjusted initial radius and spacing parameters
    r_initial = 0.102  # Slightly smaller for better flexibility
    buffer = 1e-6  # Small buffer to prevent boundary violations

    # Generate staggered grid with 5 rows and varying number of circles per row
    for row in range(5):  # 5 rows total
        # Even rows start at r_initial, odd rows also start with buffer
        if row % 2 == 0:
            x_start = r_initial + buffer  # Even rows start slightly inside
        else:
            x_start = r_initial + buffer  # Odd rows also start with buffer

        # Varying number of circles per row to fit better
        if row == 0 or row == 2 or row == 4:
            num_circles = 5  # Even rows (0, 2, 4) have 5 circles
        elif row == 1:
            num_circles = 6  # First odd row has 6 circles
        else:  # row == 3
            num_circles = 5  # Second odd row has 5 circles

        if num_circles == 0:
            continue

        # Calculate horizontal spacing for this row
        if num_circles == 1:
            spacing_row = 0.0
        else:
            # Ensure horizontal spacing is at least 2*r_initial to prevent overlaps
            max_horizontal = 1 - 2 * r_initial
            spacing_row = max_horizontal / (num_circles - 1) if max_horizontal > 0 else 0.0

        # Place circles in this row
        for col in range(num_circles):
            x = x_start + col * spacing_row
            # Vertical positioning with refined vertical spacing
            if row == 0:
                y = r_initial + buffer  # First row starts with buffer
            else:
                # Vertical spacing with a refined factor for denser packing
                y = r_initial + buffer + row * 1.0 * np.sqrt(3) * r_initial

            # Ensure y does not exceed 1 - r_initial
            if y + r_initial > 1 + 1e-6:
                y = 1 - r_initial - 1e-6  # Clamp to prevent overflow

            initial_centers.append([x, y])
            # Assign initial radii based on row (middle row gets a slight boost)
            if row == 2:
                initial_radii.append(r_initial + 0.003)  # Increased boost for central row
            else:
```

```
60                      initial_radii.append(r_initial)
61
62          # Flatten the initial variables for optimization
63          variables_initial = []
64          for i in range(n):
65              variables_initial.extend(initial_centers[i])
66              variables_initial.append(initial_radii[i])
67
68          # Objective function to maximize sum of radii
69          def objective(vars):
70              total = 0.0
71              for i in range(n):
72                  idx = i * 3
73                  total += vars[idx + 2]
74              return -total  # Minimize negative sum to maximize
75
76          # Define constraints
77          constraints = []
78
79          # Constraints for center positions and radii
80          for i in range(n):
81              # x_i >= r_i
82              def constraint1(vars, i=i):
83                  idx = i * 3
84                  return vars[idx] - vars[idx + 2]
85              constraints.append({'type': 'ineq', 'fun': constraint1})
86
87              # x_i + r_i <= 1
88              def constraint2(vars, i=i):
89                  idx = i * 3
90                  return 1 - (vars[idx] + vars[idx + 2])
91              constraints.append({'type': 'ineq', 'fun': constraint2})
92
93              # y_i >= r_i
94              def constraint3(vars, i=i):
95                  idx = i * 3
96                  return vars[idx + 1] - vars[idx + 2]
97              constraints.append({'type': 'ineq', 'fun': constraint3})
98
99              # y_i + r_i <= 1
100             def constraint4(vars, i=i):
101                 idx = i * 3
102                 return 1 - (vars[idx + 1] + vars[idx + 2])
103             constraints.append({'type': 'ineq', 'fun': constraint4})
104
105         # Pairwise distance constraints
106         for i in range(n):
107             for j in range(i + 1, n):
108                 def constraint_pair(vars, i=i, j=j):
109                     idx_i = i * 3
110                     idx_j = j * 3
111                     x_i, y_i, r_i = vars[idx_i], vars[idx_i + 1], vars[idx_i + 2]
112                     x_j, y_j, r_j = vars[idx_j], vars[idx_j + 1], vars[idx_j + 2]
113                     dist = np.sqrt((x_i - x_j)**2 + (y_i - y_j)**2)
114                     return dist - (r_i + r_j)
115                 constraints.append({'type': 'ineq', 'fun': constraint_pair})
116
117         # Optimize using Sequential Least Squares Programming with refined parameters
118         result = minimize(
119             objective,
120             variables_initial,
121             method='SLSQP',
122             constraints=constraints,
123             options={
124                 'ftol': 1e-14,
125                 'maxiter': 1000000,
126                 'disp': False,
127                 'eps': 1e-12,
128                 'iprint': 0,  # Suppress verbose output
129                 'finite_diff_rel_step': np.sqrt(np.finfo(float).eps)
130             }
131         )
132
133         # Extract optimized centers and radii
134         optimized_vars = result.x
135         centers = []
136         radii = []
137         for i in range(n):
138             idx = i * 3
```

```python
139            centers.append([optimized_vars[idx], optimized_vars[idx + 1]])
140            radii.append(optimized_vars[idx + 2])
141
142    sum_radii = sum(radii)
143    return np.array(centers), np.array(radii), sum_radii
144 ```
```

## Circle Packing ($n = 32$)

```python
1  ```python
2  import numpy as np
3  from scipy.optimize import minimize
4
5  def run_packing():
6      n = 32
7      r_initial = 1.0 / (2.0 + 5.0 * np.sqrt(3))  # Maximum radius for vertical constraint
8
9      # Generate hexagonal arrangement for 30 circles
10     centers = []
11     for row in range(6):  # 6 rows with 5 columns each
12         y = r_initial * (1 + row * np.sqrt(3))
13         if row % 2 == 0:
14             x_start = (1.0 - 9.0 * r_initial) / 2  # Adjusted to use more horizontal space
15         else:
16             x_start = (1.0 - 9.0 * r_initial) / 2 + r_initial
17         for col in range(5):  # 5 columns
18             x = x_start + col * 2 * r_initial
19             # Ensure the circle is within the square and properly spaced
20             centers.append([x, y])
21
22     # Add two extra circles near the top-right and bottom-right corners with adjusted initial positions
23     extra_x = 1.0 - r_initial - 0.0005
24     extra_y_top = 0.5
25     extra_y_bottom = r_initial + 0.0005
26     centers.append([extra_x, extra_y_top])
27     centers.append([extra_x, extra_y_bottom])
28
29     # Initial radii for all circles
30     radii = [r_initial] * n
31
32     # Flatten the centers and radii into a single array for optimization
33     x0 = np.concatenate([np.array(centers).ravel(), np.array(radii)])
34
35     # Objective function to maximize: sum of radii
36     def objective(x):
37         # Unflatten x into centers and radii
38         centers_flat = x[:n*2].reshape(n, 2)
39         radii_flat = x[n*2:]
40         return -np.sum(radii_flat)  # Negative because we minimize
41
42     # Constraints: for each circle, x_i - r_i >= -1e-12, x_i + r_i <= 1 + 1e-12, same for y
43     # and for each pair, distance >= r_i + r_j - 1e-12
44
45     # Define constraint functions
46     def constraint_boundary(x):
47         centers_flat = x[:n*2].reshape(n, 2)
48         radii_flat = x[n*2:]
49         constraints = []
50         epsilon = 1e-12
51         for i in range(n):
52             x_i, y_i = centers_flat[i]
53             r_i = radii_flat[i]
54             constraints.append(x_i - r_i + epsilon)  # x_i - r_i >= -epsilon
55             constraints.append(1 + epsilon - x_i - r_i)  # x_i + r_i <= 1 + epsilon
56             constraints.append(y_i - r_i + epsilon)  # y_i - r_i >= -epsilon
57             constraints.append(1 + epsilon - y_i - r_i)  # y_i + r_i <= 1 + epsilon
58         return np.array(constraints)
59
60     # Define constraint for non-overlapping
61     def constraint_overlap(x):
62         centers_flat = x[:n*2].reshape(n, 2)
63         radii_flat = x[n*2:]
64         constraints = []
65         for i in range(n):
66             for j in range(i + 1, n):
67                 dx = centers_flat[i, 0] - centers_flat[j, 0]
68                 dy = centers_flat[i, 1] - centers_flat[j, 1]
```

```
69                    dist = np.sqrt(dx**2 + dy**2)
70                    constraints.append(dist - radii_flat[i] - radii_flat[j] + 1e-12)
71            return np.array(constraints)
72
73        # Combine all constraints
74        cons = []
75        # Boundary constraints
76        cons.append({'type': 'ineq', 'fun': lambda x: constraint_boundary(x)})
77        # Overlap constraints
78        cons.append({'type': 'ineq', 'fun': lambda x: constraint_overlap(x)})
79
80        # Perform optimization with adjusted parameters
81        result = minimize(
82            objective,
83            x0,
84            method='SLSQP',
85            constraints=cons,
86            tol=1e-10,
87            options={'disp': False,
88            'maxiter': 200000, 'ftol': 1e-12, 'eps': 1e-8}
89        )
90
91        # Extract the result
92        optimized_x = result.x
93        centers_opt = optimized_x[:n*2].reshape(n, 2)
94        radii_opt = optimized_x[n*2:]
95
96        # Check if optimization was successful
97        if not result.success:
98            print("Optimization failed")
99            # Fallback to initial guess
100           centers_opt = np.array(centers)
101           radii_opt = np.array(radii)
102       else:
103           # Validate the packing
104           valid = validate_packing(centers_opt, radii_opt)
105           if not valid:
106               print("Validation failed")
107               # Fallback to initial guess
108               centers_opt = np.array(centers)
109               radii_opt = np.array(radii)
110
111       sum_radii = np.sum(radii_opt)
112       return centers_opt, radii_opt, sum_radii
113   ```
```

## C.2. Autocorrelation Inequalities

**Insights.** For the first inequality, early improvements down to $1.510$ came from trying and improving gradient-based optimization (e.g., using Adam with softmax parameterization). To reduce the bound from around $1.510$ to $1.504$, the policy mostly used linear programming (LP), following the insights in (Matolcsi & Vinuesa, 2010). The key insight for the later steps, that gradually achieved the state-of-the-art, was using heuristics to focus optimization only on the constraints that are close to being tight—where each constraint in the LP bounds one position of the convolution. Heuristics included picking the top K positions where the convolution was largest and only including those in the LP, as well as computing gradients from all near-maximum positions rather than just the single largest for gradient-based methods. Unlike AlphaEvolve (Georgiev et al., 2025), which mentions the authors suggested ideas such as using Newton type methods, we never intervened on the optimization process.

For autocorrelation inequalities, initial sequences are created by sampling a random value in $[0, 1]$ and repeating it between 1,000 and 8,000 times (or loading a state-of-the-art construction when available). For the first inequality, the verifier computes the upper bound $2n \cdot \max(f * f)/(\sum f)^2$ where $f * f$ denotes discrete autocorrelation; it validates that inputs are non-empty lists of non-negative floats clamped to $[0, 1000]$ with sum $\geq 0.01$, and returns $\infty$ for invalid constructions. For the second inequality, verifier computes the lower bound $C_2 = \|f * f\|_2^2/(\|f * f\|_1 \cdot \|f * f\|_\infty)$ using piecewise-linear integration for the $L^2$ norm (Simpson-like rule with endpoint zeros) over the normalized interval $[-1/2, 1/2]$. Each algorithm is run with 1 GB with 2 CPUs each and a timeout of up to 1100 seconds.

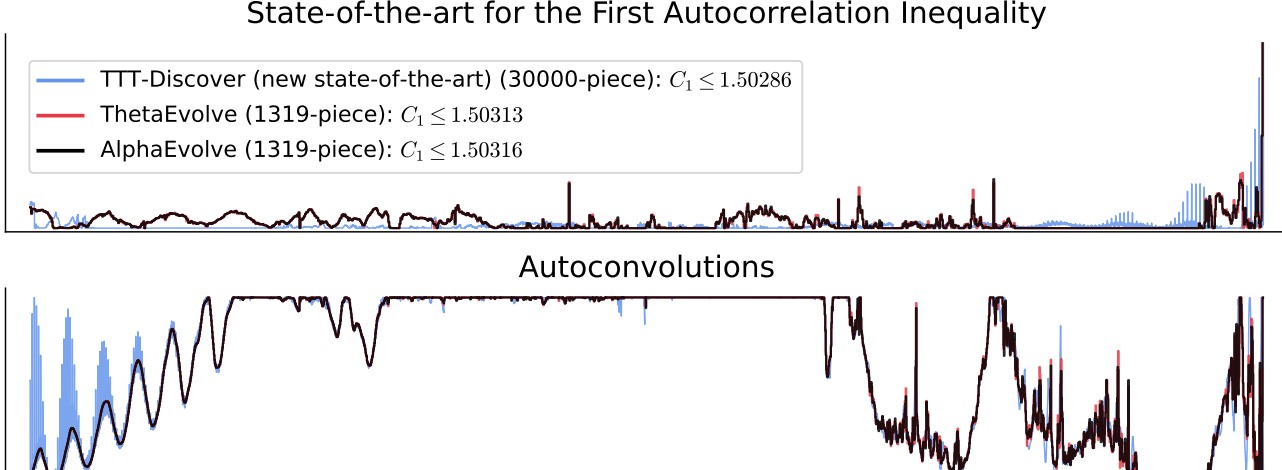

*Figure 7.* We show the prior and new state-of-the-art, with the (normalized) step functions and their autoconvolutions. Both AlphaEvolve and TTT-Discover starts the discovery process from scratch, while ThetaEvolve initializes from the AlphaEvolve construction, and thus is very similar to the AlphaEvolve construction. TTT-Discover found a 30,000-piece step function that certifies that the upper bound $C_1 \leq 1.50286$, while AlphaEvolve and ThetaEvolve constructions are 1319-piece. We overlay the step functions and their autoconvolution visually for qualitative comparison.

### C.3. Erdős'

We initialize TTT-Discover with random constructions of 40-100 samples around 0.5 with random perturbations. We filter out sequences with more than 1000 values in the verifier. Each algorithm is run with 1 GB with 2 CPUs each and a timeout of up to 1100 seconds.

**Insights.** The discovered algorithms minimize the correlation bound using FFT-accelerated gradient descent combined with random hill climbing and simulated annealing. The code maintains feasibility by projecting onto the constraint set where $f(x) \in [0, 1]$ with with $\int f = 1$. Interestingly, the solution found by TTT-Discover is asymmetric.

## D. Kernel engineering

For TriMul, we provide a matrix multiplication kernel that triton provides in README, mostly for syntax purposes. For MLA-Decode, we first put a softmax kernel in a preliminary prompt to let the base model generate a correct but unoptimized MLA-Decode kernel, and then use that as the initial state with the earlier softmax example removed.

**Discovered MLA-Decode kernels.** The kernels shown in Table 9 mainly rely on `torch.compile()` for optimization. Specifically, they adopt a specific configuration of `torch.compile`. However, these kernels do not leverage Triton for fine-grained optimization, which may limit further improvements and more flexible use case. We additionally filter and evaluate generated kernels that explicitly use Triton despite their slightly slower runtime and report in Table 10.

> **Human Expert Review — Matej Sirovatka, Alex Zhang, Mark Saroufim (GPUMode)**
>
> The referenced solution correctly determined that the problem is memory bound because of the surrounding point-wise operations so the agent focuses as much as possible on operation fusions, lowering the memory traffic and kernel launch overhead.
>
> It also stores activations in fp16, while this is fully aligned with the problem definition and defined tolerances, it could potentially lead to numerical stability issues in full workloads.
>
> Overall the agent's strategy is to reduce memory bandwidth via fusions, lower precision and delegating the big matrix multiplications to cuBLAS, as those are non-trivial to beat. This is similar to the current best human solutions, but executed on better. Most of the human solutions lag behind in fusing some of the more complex operators together, resulting in this solution outperforming them by a large margin.

| Method | Model | AMD MI300X - MLA Decode ($\mu s$, $\downarrow$) [95% CI] | | |
|---|---|---|---|---|
| | | Instance 1 | Instance 2 | Instance 3 |
| 1st human | – | 1653.8 [1637.3, 1670.3] | 1688.6 [1672.8, 1704.3] | 1668.7 [1637.0, 1700.3] |
| 2nd human | – | 1662.8 [1648.8, 1676.8] | 1688.6 [1677.6, 1699.5] | 1679.7 [1653.4, 1705.9] |
| 3rd human | – | 1723.0 [1711.5, 1734.5] | 1765.8 [1758.1, 1773.5] | 1718.0 [1698.3, 1737.7] |
| 4th human | – | 1768.7 [1750.3, 1787.2] | 1769.9 [1755.2, 1784.6] | 1767.0 [1736.2, 1797.8] |
| 5th human | – | 2038.6 [2017.8, 2059.3] | 2037.3 [2021.0, 2053.6] | 2041.9 [1989.0, 2094.8] |
| Best-of-25600 | gpt-oss-120b | 2286.0 [2264.2, 2307.8] | 2324.1 [2306.0, 2342.1] | 2275.2 [2267.3, 2283.1] |
| TTT-Discover | gpt-oss-120b | 1669.1 [1649.2, 1688.9] | 1706.1 [1685.9, 1726.3] | 1671.3 [1646.0, 1696.5] |

*Table 9.* AMD MLA Decode runtimes on AMD MI300X across three instances. Values are mean runtime across 10 trials with 95% confidence intervals. Top-5 human submissions are from the GPUMode leaderboard. We trained our kernels using an H200 GPUs even though the task is to minimize runtime on MI300X GPUs, since those were not available at scale in online providers. We only selected kernels using a single MI300X GPU. There is significant variance across AMD MI300X instances available via AMD Developer Cloud. Thus, we performed our kernel selection and evaluation across three different instances. In each instance, our best kernel was different, and in none of the cases our best kernel where better than the top human submission with statistical significance.

| Method | Model | AMD MI300X - MLA Decode ($\mu s$, $\downarrow$) [95% CI] | | |
|---|---|---|---|---|
| | | Instance 1 | Instance 2 | Instance 3 |
| 1st human | – | 1653.8 [1637.3, 1670.3] | 1688.6 [1672.8, 1704.3] | 1668.7 [1637.0, 1700.3] |
| 2nd human | – | 1662.8 [1648.8, 1676.8] | 1688.6 [1677.6, 1699.5] | 1679.7 [1653.4, 1705.9] |
| 3rd human | – | 1723.0 [1711.5, 1734.5] | 1765.8 [1758.1, 1773.5] | 1718.0 [1698.3, 1737.7] |
| 4th human | – | 1768.7 [1750.3, 1787.2] | 1769.9 [1755.2, 1784.6] | 1767.0 [1736.2, 1797.8] |
| 5th human | – | 2038.6 [2017.8, 2059.3] | 2037.3 [2021.0, 2053.6] | 2041.9 [1989.0, 2094.8] |
| Best-of-25600 | gpt-oss-120b | 2286.0 [2264.2, 2307.8] | 2324.1 [2306.0, 2342.1] | 2275.2 [2267.3, 2283.1] |
| TTT-Discover | gpt-oss-120b | 1740.6 [1697.9, 1783.2] | 1754.4 [1736.7, 1772.2] | 1707.1 [1664.5, 1749.8] |

*Table 10.* Results of TTT MLA-Decode kernels filtered with Triton kernels.

### D.1. Kernel evaluation details

**Setup of verifier for training.** We follow the exact same practice for evaluating kernel correctness and runtime as the original GPUMode competitions. Specifically, the verifier used in our training jobs uses the same code as the official GPU Mode Competition Github repository, with minor adjustment to integrate into our training codebase. The verification process includes a correctness check that compare output values between the custom kernel and a pytorch reference program under a designated precision, followed by runtime benchmarking of the custom kernel across multiple iterations. All the details in our verification procedure follow the official competition exactly, including the test cases used for correctness check and benchmarking, hyper-parameters such as matching precision and iterations used for timing, etc. We run our verifier on H100s for TriMul, and H200s for MLA-Decode, both from the Modal cloud platform.

**Setup of environments for final report.** For final report, we submit to the official TriMul A100/H100 leaderboard and report the runtime shown. For TriMul B200/MI300X and MLA-Decode MI300X tasks, due to an infra problem on GPU Mode's server, we could not submit to the official leaderboard. For these tasks, we work with the GPU Mode team closely to set up our local environment, which replicates the official environment and gets GPU Mode team's review and confirmation.

**Selection protocol for best kernels.** For TriMul H100 task, we select 20 kernels with the best verifier score throughout training. For other tasks, since our verifier hardware in training is different from the target hardware, we select 20 kernels with the best training scores plus 20 random correct kernels every 10 steps of training. Finally, we used our verifier **with the target hardware** to verify each selected kernels for three times, and submit the kernel with the smallest average runtime for final report.

## D.2. Analysis of best generated kernels

**TriMul H100 kernels.** The below code shows the best TriMul kernels discovered by TTT for H100 GPU. At the high level, the kernel correctly identifies a major bottleneck of the problem, which is the heavy memory I/O incurred by a series of elementwise operations, and then focuses on fusing them with Triton. Specifically, the kernel fuses: (i) operations in the input LayerNorm, (ii) elementwise activation and multiplication for input gating, and (iii) operations in the output Layernorm and output gating. As for the compute-heavy operation, which is an $O(N^3)$ matmul, the kernel converts its inputs to fp16 and delegates the computation to cuBLAS to effectively leverage the TensorCores on H100 GPU.

Compared with kernels generated early in training, the final kernel achieves a big improvement by (i) fusing more operations together, and (ii) deeper optimization of the memory access pattern inside fused kernels. For example, a kernel generated early fuses LayerNorm operations, but does not fuse the input gating process. A kernel generated in the middle of training fuses the same operations as the final kernel, but has less efficient memory access pattern in the fused kernel for output LayerNorm, gating, and output projection.

Compared with the best human leaderboard kernel, the TTT discovered kernel adopts a similar fusion strategy for the input LayerNorm and input gating. Different from human kernel, the TTT kernel does not perform as much auto-tuning of block size, which could be a limitation. However, the TTT kernel fuses the output LayerNorm and gating with output projection whereas the human kernel does not, which could explain the moderate advantage of the former.

**TriMul H100**

```
1  """
2  Outgoing TriMul (AlphaFold-3) - Triton accelerated forward pass.
3
4  The implementation follows the reference ``TriMul`` module but fuses the
5  expensive kernels:
6
7  1. Row-wise LayerNorm over the last dimension (FP16 output, FP32 reduction).
8  2. Fused projection, gating and optional scalar mask:
9       * left_proj, right_proj  = x_norm @ W_proj
10      * left_gate, right_gate, out_gate = sigmoid(x_norm @ W_gate)
11      * left  = left_proj  * left_gate  * mask
12      * right = right_proj * right_gate * mask
13  3. Pairwise multiplication across the sequence dimension (batched GEMM on
14      fp16 tensors).
15  4. Fused hidden-dim LayerNorm -> out-gate multiplication -> final linear
16      projection (all in one kernel, FP16 matmul with FP32 accumulation).
17
18  The output tensor has shape ``[B, N, N, dim]`` and dtype ``float32``.
19  """
20
21  from typing import Tuple, Dict
22  import torch
23  import triton
24  import triton.language as tl
25
26
27  # -----------------------------------------------------------------------
28  # 1) Row-wise LayerNorm (FP16 output, FP32 accumulator)
29  # -----------------------------------------------------------------------
30  @triton.jit
31  def _row_ln_fp16_kernel(
32      X_ptr, Y_ptr,          # (M, C) input / output
33      w_ptr, b_ptr,          # LN weight & bias (fp32)
34      M, C: tl.constexpr,    # rows, columns (C is compile-time constant)
35      eps,
36      BLOCK_M: tl.constexpr,
37      BLOCK_C: tl.constexpr,
38  ):
39      pid = tl.program_id(0)
40      row_start = pid * BLOCK_M
41      rows = row_start + tl.arange(0, BLOCK_M)
42      row_mask = rows < M
43
44      # ---------- mean / var (fp32) ----------
45      sum_val = tl.zeros([BLOCK_M], dtype=tl.float32)
46      sumsq_val = tl.zeros([BLOCK_M], dtype=tl.float32)
47
48      for c in range(0, C, BLOCK_C):
49          cur_c = c + tl.arange(0, BLOCK_C)
50          col_mask = cur_c < C
```

```
51          x = tl.load(
52              X_ptr + rows[:, None] * C + cur_c[None, :],
53              mask=row_mask[:, None] & col_mask[None, :],
54              other=0.0,
55          ).to(tl.float32)                    # (BLOCK_M, BLOCK_C)
56
57          sum_val += tl.sum(x, axis=1)
58          sumsq_val += tl.sum(x * x, axis=1)
59
60      mean = sum_val / C
61      var = sumsq_val / C - mean * mean
62      inv_std = 1.0 / tl.sqrt(var + eps)
63
64      # ---------- normalize + affine (fp16) ----------
65      for c in range(0, C, BLOCK_C):
66          cur_c = c + tl.arange(0, BLOCK_C)
67          col_mask = cur_c < C
68          x = tl.load(
69              X_ptr + rows[:, None] * C + cur_c[None, :],
70              mask=row_mask[:, None] & col_mask[None, :],
71              other=0.0,
72          ).to(tl.float32)
73
74          y = (x - mean[:, None]) * inv_std[:, None]
75
76          w = tl.load(w_ptr + cur_c, mask=col_mask, other=0.0)
77          b = tl.load(b_ptr + cur_c, mask=col_mask, other=0.0)
78
79          y = y * w[None, :] + b[None, :]
80          tl.store(
81              Y_ptr + rows[:, None] * C + cur_c[None, :],
82              y.to(tl.float16),
83              mask=row_mask[:, None] & col_mask[None, :],
84          )
85
86
87  def _row_layernorm_fp16(
88      x: torch.Tensor,
89      weight: torch.Tensor,
90      bias: torch.Tensor,
91      eps: float = 1e-5,
92  ) -> torch.Tensor:
93      """Row-wise LayerNorm over the last dim -> FP16 output."""
94      B, N, _, C = x.shape
95      M = B * N * N
96      x_flat = x.view(M, C).contiguous()
97      y_flat = torch.empty((M, C), dtype=torch.float16, device=x.device)
98
99      BLOCK_M = 128
100     BLOCK_C = 128
101     grid = lambda meta: (triton.cdiv(M, meta["BLOCK_M"]),)
102
103     _row_ln_fp16_kernel[grid](
104         x_flat,
105         y_flat,
106         weight,
107         bias,
108         M,
109         C,
110         eps,
111         BLOCK_M=BLOCK_M,
112         BLOCK_C=BLOCK_C,
113         num_warps=8,
114     )
115     return y_flat.view(B, N, N, C)
116
117
118 # ------------------------------------------------------------------------
119 # 2) Fused projection + gating + optional mask
120 # ------------------------------------------------------------------------
121 @triton.jit
122 def _proj_gate_mask_kernel(
123     x_ptr,                          # (M, C) fp16
124     mask_ptr,                       # (M,)  fp16 (if MASKED==1)
125     left_proj_w_ptr,                # (C, H) fp16
126     left_gate_w_ptr,                # (C, H) fp16
127     right_proj_w_ptr,               # (C, H) fp16
128     right_gate_w_ptr,               # (C, H) fp16
129     out_gate_w_ptr,                 # (C, H) fp16
```

```
130        left_ptr,                        # (B, H, N, N) fp16
131        right_ptr,                       # (B, H, N, N) fp16
132        out_gate_ptr,                    # (B, N, N, H) fp16
133        M, N, C: tl.constexpr, H: tl.constexpr,
134        BLOCK_M: tl.constexpr,
135        BLOCK_H: tl.constexpr,
136        BLOCK_K: tl.constexpr,
137        MASKED: tl.constexpr,
138 ):
139        pid_m = tl.program_id(0)   # row block
140        pid_h = tl.program_id(1)   # hidden block
141
142        row_start = pid_m * BLOCK_M
143        hid_start = pid_h * BLOCK_H
144
145        rows = row_start + tl.arange(0, BLOCK_M)          # (BLOCK_M,)
146        hids = hid_start + tl.arange(0, BLOCK_H)          # (BLOCK_H,)
147
148        row_mask = rows < M
149        hid_mask = hids < H
150
151        # --------------- mask (scalar per row) ---------------
152        if MASKED:
153            mask_val = tl.load(mask_ptr + rows, mask=row_mask, other=0.0).to(tl.float32)  # (BLOCK_M,)
154        else:
155            mask_val = tl.full([BLOCK_M], 1.0, dtype=tl.float32)
156
157        # --------------- accumulators (fp32) ---------------
158        acc_lp = tl.zeros((BLOCK_M, BLOCK_H), dtype=tl.float32)  # left proj
159        acc_lg = tl.zeros((BLOCK_M, BLOCK_H), dtype=tl.float32)  # left gate
160        acc_rp = tl.zeros((BLOCK_M, BLOCK_H), dtype=tl.float32)  # right proj
161        acc_rg = tl.zeros((BLOCK_M, BLOCK_H), dtype=tl.float32)  # right gate
162        acc_og = tl.zeros((BLOCK_M, BLOCK_H), dtype=tl.float32)  # out gate
163
164        for k in range(0, C, BLOCK_K):
165            cur_k = k + tl.arange(0, BLOCK_K)
166            k_mask = cur_k < C
167
168            # input tile (fp16 -> fp32)
169            a = tl.load(
170                x_ptr + rows[:, None] * C + cur_k[None, :],
171                mask=row_mask[:, None] & k_mask[None, :],
172                other=0.0,
173            )  # (BLOCK_M, BLOCK_K) fp16
174
175            # weight tiles (C,H) column-major
176            w_lp = tl.load(
177                left_proj_w_ptr + cur_k[:, None] * H + hids[None, :],
178                mask=k_mask[:, None] & hid_mask[None, :],
179                other=0.0,
180            )
181            w_lg = tl.load(
182                left_gate_w_ptr + cur_k[:, None] * H + hids[None, :],
183                mask=k_mask[:, None] & hid_mask[None, :],
184                other=0.0,
185            )
186            w_rp = tl.load(
187                right_proj_w_ptr + cur_k[:, None] * H + hids[None, :],
188                mask=k_mask[:, None] & hid_mask[None, :],
189                other=0.0,
190            )
191            w_rg = tl.load(
192                right_gate_w_ptr + cur_k[:, None] * H + hids[None, :],
193                mask=k_mask[:, None] & hid_mask[None, :],
194                other=0.0,
195            )
196            w_og = tl.load(
197                out_gate_w_ptr + cur_k[:, None] * H + hids[None, :],
198                mask=k_mask[:, None] & hid_mask[None, :],
199                other=0.0,
200            )
201
202            # fp16*fp16 -> fp32 dot products
203            acc_lp += tl.dot(a, w_lp)
204            acc_lg += tl.dot(a, w_lg)
205            acc_rp += tl.dot(a, w_rp)
206            acc_rg += tl.dot(a, w_rg)
207            acc_og += tl.dot(a, w_og)
208
```

```
209        # ---------------- sigmoid gates ------------------------
210        left_gate  = 1.0 / (1.0 + tl.exp(-acc_lg))
211        right_gate = 1.0 / (1.0 + tl.exp(-acc_rg))
212        out_gate   = 1.0 / (1.0 + tl.exp(-acc_og))
213
214        # ---------------- apply mask and per-row gates ----------
215        left_out  = acc_lp * left_gate * mask_val[:, None]
216        right_out = acc_rp * right_gate * mask_val[:, None]
217
218        # ---------------- store left/right (B,H,N,N) ------------
219        N_sq = N * N
220        b_idx = rows // N_sq
221        rem   = rows - b_idx * N_sq
222        i_idx = rem // N
223        k_idx = rem - i_idx * N
224
225        # layout for left/right: (B, H, N, N) -> flat index:
226        off = ((b_idx[:, None] * H + hids[None, :]) * N_sq) + i_idx[:, None] * N + k_idx[:, None]
227
228        tl.store(
229            left_ptr + off,
230            left_out.to(tl.float16),
231            mask=row_mask[:, None] & hid_mask[None, :],
232        )
233        tl.store(
234            right_ptr + off,
235            right_out.to(tl.float16),
236            mask=row_mask[:, None] & hid_mask[None, :],
237        )
238
239        # ---------------- store out_gate (B,N,N,H) --------------
240        off_gate = rows[:, None] * H + hids[None, :]
241        tl.store(
242            out_gate_ptr + off_gate,
243            out_gate.to(tl.float16),
244            mask=row_mask[:, None] & hid_mask[None, :],
245        )
246
247
248    # ------------------------------------------------------------------------
249    # 3) Fused hidden-dim LayerNorm -> out-gate -> final linear
250    # ------------------------------------------------------------------------
251    @triton.jit
252    def _ln_gate_out_linear_fused_kernel(
253        hidden_ptr,          # (B*H*N*N,) fp16 flattened
254        out_gate_ptr,        # (B*N*N*H,) fp16 flattened
255        ln_w_ptr, ln_b_ptr,  # (H,) fp32
256        w_out_ptr,           # (H, D) fp16
257        out_ptr,             # (B, N, N, D) fp32
258        B, N, H, D: tl.constexpr,
259        eps: tl.constexpr,
260        BLOCK_M: tl.constexpr,
261        BLOCK_H: tl.constexpr,
262        BLOCK_D: tl.constexpr,
263    ):
264        pid = tl.program_id(0)
265        row_start = pid * BLOCK_M
266        rows = row_start + tl.arange(0, BLOCK_M)            # flat index for (b,i,j)
267        row_mask = rows < (B * N * N)
268
269        N_sq = N * N
270        b_idx = rows // N_sq
271        rem = rows - b_idx * N_sq
272        i_idx = rem // N
273        j_idx = rem - i_idx * N
274
275        # ----- load hidden slice (BLOCK_M, BLOCK_H) ------------
276        hids = tl.arange(0, BLOCK_H)
277        hid_mask = hids < H
278
279        hidden_off = ((b_idx[:, None] * H + hids[None, :]) * N_sq) + i_idx[:, None] * N + j_idx[:, None]
280        hidden_tile = tl.load(
281            hidden_ptr + hidden_off,
282            mask=row_mask[:, None] & hid_mask[None, :],
283            other=0.0,
284        )  # fp16
285        hidden_fp32 = hidden_tile.to(tl.float32)
286
287        # ----- mean / var across H (fp32) -----
```

```
288        sum_val = tl.sum(hidden_fp32, axis=1)                    # (BLOCK_M,)
289        sumsq_val = tl.sum(hidden_fp32 * hidden_fp32, axis=1)    # (BLOCK_M,)
290        mean = sum_val / H
291        var = sumsq_val / H - mean * mean
292        inv_std = 1.0 / tl.sqrt(var + eps)
293
294        # ----- LayerNorm (fp32) -----
295        w_ln = tl.load(ln_w_ptr + hids, mask=hid_mask, other=0.0)   # (H,)
296        b_ln = tl.load(ln_b_ptr + hids, mask=hid_mask, other=0.0)   # (H,)
297        hidden_norm = (hidden_fp32 - mean[:, None]) * inv_std[:, None]
298        hidden_norm = hidden_norm * w_ln[None, :] + b_ln[None, :]   # (BLOCK_M, BLOCK_H)
299
300        # ----- out-gate (fp32) -----
301        out_gate_off = rows[:, None] * H + hids[None, :]
302        out_gate_tile = tl.load(
303            out_gate_ptr + out_gate_off,
304            mask=row_mask[:, None] & hid_mask[None, :],
305            other=0.0,
306        ).to(tl.float32)                                         # (BLOCK_M, BLOCK_H)
307
308        gated = hidden_norm * out_gate_tile                      # (BLOCK_M, BLOCK_H)
309
310        # ----- final linear projection (fp16 matmul, fp32 acc) -----
311        gated_fp16 = gated.to(tl.float16)
312
313        for d0 in range(0, D, BLOCK_D):
314            cols = d0 + tl.arange(0, BLOCK_D)
315            col_mask = cols < D
316
317            w_out = tl.load(
318                w_out_ptr + hids[:, None] * D + cols[None, :],
319                mask=hid_mask[:, None] & col_mask[None, :],
320                other=0.0,
321            )   # (BLOCK_H, BLOCK_D) fp16
322
323            out = tl.dot(gated_fp16, w_out)                      # (BLOCK_M, BLOCK_D) fp32
324
325            tl.store(
326                out_ptr + rows[:, None] * D + cols[None, :],
327                out,
328                mask=row_mask[:, None] & col_mask[None, :],
329            )
330

331
332 # --------------------------------------------------------------------------
333 # 4) Entrypoint
334 # --------------------------------------------------------------------------
335 def custom_kernel(
336     data: Tuple[torch.Tensor, torch.Tensor, Dict[str, torch.Tensor], Dict]
337 ) -> torch.Tensor:
338     """
339     Forward pass of the outgoing TriMul operator (no gradients).
340
341     Arguments
342     ---------
343     data : (input, mask, weights, config)
344         - input  : Tensor[B, N, N, C] (float32)
345         - mask   : Tensor[B, N, N] (bool/float) or None
346         - weights: dict of module parameters (float32)
347         - config : dict with ``dim`` (C) and ``hidden_dim`` (H) and optional ``nomask``
348
349     Returns
350     -------
351     Tensor[B, N, N, C] (float32)
352     """
353     inp, mask, weights, cfg = data
354     dim = cfg["dim"]                # C
355     hidden_dim = cfg["hidden_dim"]  # H
356     nomask = cfg.get("nomask", True)
357     eps = 1e-5
358
359     device = inp.device
360     B, N, _, _ = inp.shape
361     M = B * N * N                   # total rows for row-wise ops
362
363     # ------------------------------------------------------------
364     # 1) Row-wise LayerNorm (fp16 output)
365     # ------------------------------------------------------------
366     x_norm = _row_layernorm_fp16(
```

```
367            inp,
368            weights["norm.weight"],
369            weights["norm.bias"],
370            eps=eps,
371        )   # (B, N, N, C) fp16
372
373        # ----------------------------------------------------------------
374        # 2) Prepare projection / gate weights  (C, H) fp16, column-major
375        # ----------------------------------------------------------------
376        left_proj_w_T  = weights["left_proj.weight"].t().contiguous().to(torch.float16)
377        right_proj_w_T = weights["right_proj.weight"].t().contiguous().to(torch.float16)
378        left_gate_w_T  = weights["left_gate.weight"].t().contiguous().to(torch.float16)
379        right_gate_w_T = weights["right_gate.weight"].t().contiguous().to(torch.float16)
380        out_gate_w_T   = weights["out_gate.weight"].t().contiguous().to(torch.float16)
381
382        # ----------------------------------------------------------------
383        # 3) Mask handling (optional)
384        # ----------------------------------------------------------------
385        if not nomask and mask is not None:
386            mask_flat = mask.reshape(M).to(torch.float16).contiguous()
387            MASKED = 1
388        else:
389            mask_flat = torch.empty(0, dtype=torch.float16, device=device)
390            MASKED = 0
391
392        # ----------------------------------------------------------------
393        # 4) Allocate buffers for fused projection + gating
394        # ----------------------------------------------------------------
395        left = torch.empty((B, hidden_dim, N, N), dtype=torch.float16, device=device)
396        right = torch.empty_like(left)
397        out_gate = torch.empty((B, N, N, hidden_dim), dtype=torch.float16, device=device)
398
399        # ----------------------------------------------------------------
400        # 5) Fused projection / gating / optional mask
401        # ----------------------------------------------------------------
402        BLOCK_M = 64
403        BLOCK_H = 64
404        BLOCK_K = 32
405        grid_proj = (triton.cdiv(M, BLOCK_M), triton.cdiv(hidden_dim, BLOCK_H))
406
407        _proj_gate_mask_kernel[grid_proj](
408            x_norm,
409            mask_flat,
410            left_proj_w_T,
411            left_gate_w_T,
412            right_proj_w_T,
413            right_gate_w_T,
414            out_gate_w_T,
415            left,
416            right,
417            out_gate,
418            M,
419            N,
420            dim,
421            hidden_dim,
422            BLOCK_M=BLOCK_M,
423            BLOCK_H=BLOCK_H,
424            BLOCK_K=BLOCK_K,
425            MASKED=MASKED,
426            num_warps=4,
427        )
428
429        # ----------------------------------------------------------------
430        # 6) Pairwise multiplication (batched GEMM) - left @ right^T
431        # ----------------------------------------------------------------
432        left_mat = left.view(B * hidden_dim, N, N)                     # (B*H, N, N)
433        right_mat = right.view(B * hidden_dim, N, N).transpose(1, 2)   # (B*H, N, N)^T
434        hidden_fp16 = torch.bmm(left_mat, right_mat)                   # (B*H, N, N) fp16
435        hidden = hidden_fp16.view(B, hidden_dim, N, N)                 # (B, H, N, N) fp16
436
437        # ----------------------------------------------------------------
438        # 7) Fused hidden-dim LayerNorm -> out-gate -> final linear
439        # ----------------------------------------------------------------
440        to_out_norm_w = weights["to_out_norm.weight"]   # (H,) fp32
441        to_out_norm_b = weights["to_out_norm.bias"]     # (H,) fp32
442        to_out_w_T = weights["to_out.weight"].t().contiguous().to(torch.float16)  # (H, C)
443
444        out = torch.empty((B, N, N, dim), dtype=torch.float32, device=device)
445
```

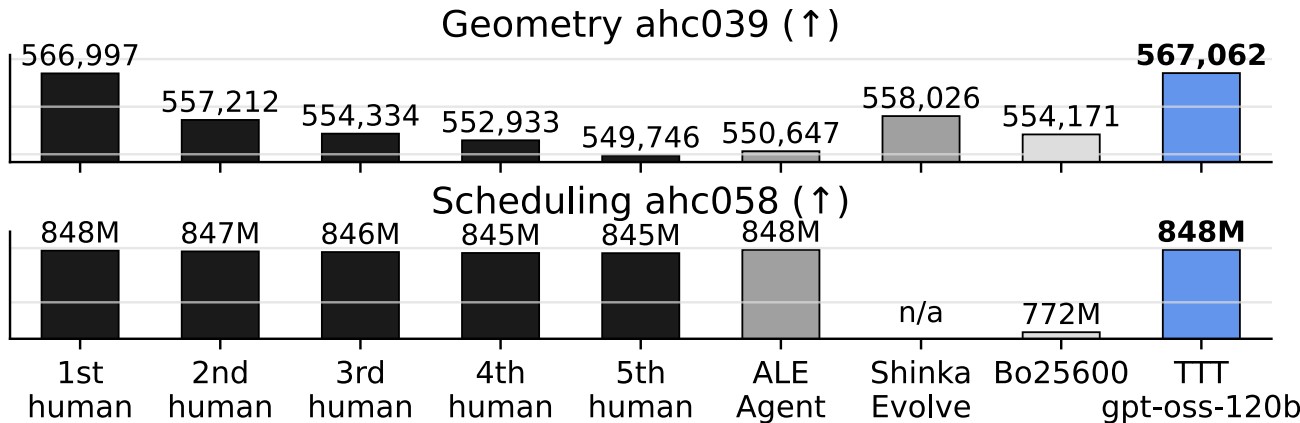

*Figure 8.* Results in algorithm engineering. We evaluate on AtCoder Heuristic Contests ahc039 (geometry) and ahc058 (scheduling). TTT-Discover generates solutions that outscore the top human submissions in both competitions. We also report comparisons to prior AI baselines, including ALE-Agent and ShinkaEvolve. Full numbers are reported in Table 2.

```
446     BLOCK_M_OUT = 64
447     BLOCK_H_OUT = hidden_dim      # cover the whole hidden dim in one kernel launch
448     BLOCK_D_OUT = 64
449
450     grid_out = (triton.cdiv(B * N * N, BLOCK_M_OUT),)
451
452     _ln_gate_out_linear_fused_kernel[grid_out](
453         hidden.view(-1),              # flat fp16 hidden
454         out_gate.view(-1),            # flat fp16 out-gate
455         to_out_norm_w,
456         to_out_norm_b,
457         to_out_w_T,
458         out,
459         B,
460         N,
461         hidden_dim,
462         dim,
463         eps,
464         BLOCK_M=BLOCK_M_OUT,
465         BLOCK_H=BLOCK_H_OUT,
466         BLOCK_D=BLOCK_D_OUT,
467         num_warps=4,
468     )
469
470     return out
```

## E. Algorithm Engineering

Hard optimization problems like package-delivery routing, crew scheduling, factory production planning, power-grid balancing—appear throughout industries and must be solved repeatedly at scale. We apply our method to these algorithm engineering problems, where a new state-of-the-art would be writing a higher-scoring algorithm than existing ones written by human experts.

AtCoder Heuristic Contest (AHC) is a series of programming competitions focused on optimization problems drawn from real-world industrial challenges (AtCoder Inc., 2025), attracting hundreds of participants including industry experts. We attempted to evaluate on two past contests, ahc039 and ahc058. ahc039 ("Purse Seine Fishing") is a computational geometry problem where you design a simple closed net on a 2D map, restricted to horizontal/vertical edges, to capture many target points while avoiding penalty points under a budget. ahc058 ("Apple Incremental Game") is a production planning problem where upgrades trade off immediate output versus growing future production capacity, and the goal is to schedule upgrades to maximize final output.

We select ahc039 because ShinkaEvolve (Lange et al., 2025) reported a solution that would have placed 2nd, and ahc058 because Sakana AI's ALE-Agent achieved the first-ever AI victory in an AHC (Sakana AI, 2026). We use the evaluation harness from ALE-Bench (Imajuku et al., 2025). We use the public test case generator to create local tests, select our

best-performing algorithm, and submit it to be scored on the official platform.

**Environment:** The state $s$ is an algorithm implementation in C++. The action $a$ consists of thinking tokens followed by C++ code. The dynamics parse the code from the action: $s' = \texttt{Parse}(a)$. The reward is the score on locally generated test cases, or zero if the algorithm fails correctness checks or exceeds the time limit of 2 seconds and memory limit of 1024MB. We select the best-performing algorithm and submit it to be scored on the official private tests. We use the evaluation harness released by (Imajuku et al., 2025). For initial states, for the ahc039 competition we use the same initial program as (Lange et al., 2025), which is based on ALE-Agent (Imajuku et al., 2025) best program, that would have placed 5th in the competition leaderboard. For ahc058 we start from scratch, similar to ALE-Agent (Sakana AI, 2026).

**Previous state-of-the-art.** We report the top human submissions on each contest leaderboard. For AI baselines, we compare to ALE-Agent (Imajuku et al., 2025) and ShinkaEvolve (Lange et al., 2025), which use ensembles of models including the gpt, Gemini, and Claude families of models. ALE-Agent (Imajuku et al., 2025) starts from scratch for both problems. ShinkaEvolve (Lange et al., 2025) reports results in ahc039 where they start from ALE-Agent solution, and improve it from 5th place to 2nd place.

**Results.** We report results in Table 2. For both competitions, if we had submitted during competition time, our algorithms would have gotten the 1st place. For ahc039, we marginally improve upon the best human, while there is a significant gap between next best AI and human scores. For ahc039, we follow ShinkaEvolve by starting from the ALE-Agent solution and improve it from 5th place to 1st place, while ShinkaEvolve reaches the 2nd place using significantly more capable frontier models such as Gemini 2.5 Pro. For ahc058, we start from scratch and outscore all submissions in the competition. We provide insights about the discovered solutions in Appendix E.

During the contest, AtCoder provides an official input generator, tester to evaluate program correctness, and a scoring function used for the final ranking. For training, we generate 150 test cases using seeds 0 through 149 from the input generator and run our program on each of these cases with an ALE-Bench provided C++20 container (yimjk/ale-bench:cpp20-202301). A program receives a non-zero reward only if it passes all correctness checks and executes within the problem time limit (2 seconds) across all 150 test cases. The per-test case score is problem-specific and matches the scoring used in the AtCoder contest. For ahc039, we use ShinkaEvolve's performance metric, which is determined by the score's relative placement among the final contest's scores, and for ahc058, we directly use the contest score.

For the final evaluation, we select the top three highest-scoring programs from our local training runs and submit them to the official AtCoder submission website. For our language, we specify C++23 (GCC 15.2.0). The submission is evaluated using the same scoring and validation process as the original contest, including checks for incorrect output, time limit violations, and compilation or runtime errors on AtCoder's hidden test cases. The resulting score is used as the final evaluation.

For AHC training runs, we make a slight modification from our standard hyperparameters. For AHC039, we decrease the prompt length + thinking token limit to 22000 due to the large initial program. For AHC058, we similarly decrease the prompt length + thinking token limit to 25000 and found that a learning rate of $2 \times 10^{-5}$ performed slightly better. For both AHC problems, we use a KL coefficient of $1 \times 10^{-2}$. Other hyperparameters are set to our standard values.

**Insights about discovered solutions:** For AHC039, the solution builds a large pool of promising axis-aligned rectangles using prefix sum scoring, then greedily seeds a connected union and uses simulated annealing with add, remove, replace, expand, shrink, and slide moves to optimize the rectangle union score under perimeter and vertex constraints, followed by cleanup and final greedy refinement.

For AHC058, the solution first builds several reasonable plans using greedy rules, different biases, and a short beam search to explore promising early decisions. Then, the program improves the best plan with simulated annealing that makes random edits, swaps, and partial rebuilds before finishing with a small local cleanup pass. It estimates the value of actions using a simple formula for how much future production an upgrade is likely to create, which guides both greedy choices and pruning. For performance, it caches intermediate states so it only recomputes parts of the plan that change. Overall, the program balances broad exploration early with focused local improvement later.

# F. Single cell analysis

> **Human Expert Review — Prof. Eric Sun (MIT)**
>
> Single-cell transcriptomics provides a high-dimensional readout on cellular gene expression patterns and has enabled new insights into both biological and disease processes. One challenge in the analysis of single-cell transcriptomics is the sparsity of the data, characterized by zero counts detected for many genes (i.e. "dropouts") due to low expression or other technical issues. MAGIC addresses this challenge by de-noising single-cell transcriptomics using diffusion or smoothing, and it has been widely incorporated in the pre-processing of single-cell data for studying multiple diseases and tissue biology. The proposed improvement on the MAGIC algorithm is simple, aligns with the underlying smoothing-based approach of MAGIC, and yields empirical improvements on key metrics. However, improvements on metrics for single-cell data analysis tasks may not always transfer to enhanced ability to obtain new biological insights, which is often difficult to quantify and therefore benchmark. Further evaluation of the proposed algorithm against MAGIC and other existing methods for biologically relevant tasks would be necessary to fully understand the extent of the reported improvements.

The OpenProblems benchmark provides three datasets: pancreas, pbmc and tabula. We select the Pancreas dataset to compute MSE and Poisson loss scores and use the other two datasets to assess generalization. MSE and Poisson loss scores are normalized with respect to the scores that no denoising and perfect denoising would get on this task. The main score metric in the OpenProblems denoising benchmark is the mean between the normalized MSE and the normalized Poisson. During verification, we reject all the solutions that obtain a normalized Poisson lower than 0.97 or larger than 1 so that we can focus only on improving a single metric, MSE.

In the prompt we also include instructions regarding what makes a solution taking inspiration from the Supplementary Materials of the OpenProblems paper (Luecken et al., 2025). For this specific applications, considering the size of the datasets, the memory limit is increased to 3GB. To force generalization, we reduce the time limits for the execution to 400 seconds.

We ran the OpenEvolve baseline with 25,600 samples. After sample 17,000, we observed the OpenEvolve database filling up with programs that timed out. Consequently, we selected the best program found up to that point.

Both TTT-Discover and the Best-of-25600 baselines are run with max tokens equal to 20,000.

Both MAGIC and the solution found by TTT-Discover are run with default parameters.

> **Denoising**

```
1
2  # ---------------------------------------------------------------------
3  # Imports
4  # ---------------------------------------------------------------------
5  import warnings
6  import numpy as np
7  import scipy.sparse as sp
8  from graphtools import Graph
9  import scprep
10 from scprep.utils import toarray
11 from scprep.normalize import library_size_normalize
12 from sklearn.decomposition import TruncatedSVD
13 import scanpy as sc
14 import sklearn.metrics
15
16 # ---------------------------------------------------------------------
17 # Helper utilities (identical to the reference implementation – unchanged)
18 # ---------------------------------------------------------------------
19
20
21 def _inverse_anscombe_refined(Y: np.ndarray, n_iter: int = 12) -> np.ndarray:
22     """Newton-iteration inverse of the Anscombe variance-stabilising transform."""
23     Y = np.asarray(Y, dtype=np.float64)
24     x = (Y / 2.0) ** 2 - 3.0 / 8.0
25     for _ in range(n_iter):
26         sqrt_term = np.sqrt(np.maximum(x + 3.0 / 8.0, 0.0))
27         x -= (2.0 * sqrt_term - Y) * sqrt_term
28     np.maximum(x, 0.0, out=x)
```

```
29        return x
30
31
32    def _inverse_ft_refined(Y: np.ndarray, n_iter: int = 12) -> np.ndarray:
33        """Newton-iteration inverse of the Freeman-Tukey transform."""
34        Y = np.asarray(Y, dtype=np.float64)
35        out = np.zeros_like(Y)
36        mask = Y > 0
37        y = Y[mask]
38
39        # Analytic start: s = (y^2-1) / (2y)   (s = √x)
40        s = np.maximum((y * y - 1.0) / (2.0 * y), 0.0)
41        x = s * s
42        for _ in range(n_iter):
43            sqrtx = np.sqrt(np.maximum(x, 0.0))
44            sqrtx1 = np.sqrt(np.maximum(x + 1.0, 0.0))
45            f = sqrtx + sqrtx1 - y
46            fprime = 0.5 / np.maximum(sqrtx, 1e-12) + 0.5 / np.maximum(sqrtx1, 1e-12)
47            x -= f / fprime
48            x = np.maximum(x, 0.0)
49        out[mask] = x
50        return out
51
52
53    def _calc_dropout(counts: np.ndarray) -> np.ndarray:
54        """Fraction of zero entries per gene."""
55        return np.mean(counts == 0, axis=0)
56
57
58    def _adaptive_blend_weights(
59        dropout: np.ndarray,
60        var_orig: np.ndarray,
61        var_diff: np.ndarray,
62        corr: np.ndarray,
63        mu: np.ndarray,
64        max_alpha: float = 0.55,
65        eps: float = 1e-12,
66    ) -> np.ndarray:
67        """
68        Compute a diffusion-blend weight for each gene.
69
70        Larger weight → gene benefits more from diffusion.
71        """
72        var_reduction = (var_orig - var_diff) / (var_orig + eps)
73        var_reduction = np.clip(var_reduction, 0.0, 1.0)
74
75        mu_norm = (mu - mu.min()) / (mu.max() - mu.min() + eps)
76        expr_factor = 1.0 - mu_norm
77
78        raw = dropout * var_reduction * (1.0 - corr) * expr_factor
79        raw = np.where(dropout > 0.8, raw * 1.2, raw)
80        w = np.clip(raw, 0.0, max_alpha)
81        return w
82
83
84    def _select_hvg_scanpy(X_norm: np.ndarray, n_hvg: int = 3000) -> np.ndarray:
85        """HVG selection using Scanpy's Seurat-flavour method."""
86        if n_hvg is None or n_hvg >= X_norm.shape[1]:
87            return np.arange(X_norm.shape[1])
88        adata = sc.AnnData(X=X_norm)
89        sc.pp.highly_variable_genes(
90            adata,
91            n_top_genes=n_hvg,
92            flavor="seurat",
93            batch_key=None,
94            subset=False,
95            inplace=True,
96        )
97        return np.where(adata.var["highly_variable"].values)[0]
98
99
100   def _row_normalize_sparse(M: sp.spmatrix) -> sp.spmatrix:
101       """Row-stochastic normalisation for a CSR/CSC matrix."""
102       row_sums = np.asarray(M.sum(axis=1)).ravel()
103       row_sums[row_sums == 0] = 1.0
104       return M.multiply(1.0 / row_sums[:, None])
105
106
```

```
107  def _symmetrize_diffusion(P: sp.spmatrix) -> sp.spmatrix:
108      """Produce a symmetric, row-stochastic diffusion operator."""
109      sym = (P + P.transpose()) * 0.5
110      return _row_normalize_sparse(sym)
111
112
113  def _add_self_loop(P: sp.spmatrix, alpha: float = 0.5) -> sp.spmatrix:
114      """Mix the identity matrix with the transition matrix."""
115      n = P.shape[0]
116      I = sp.eye(n, format=''csr``)
117      P_mix = (1.0 - alpha) * I + alpha * P
118      return _row_normalize_sparse(P_mix)
119
120
121  def _gene_correlation(X1: np.ndarray, X2: np.ndarray, eps: float = 1e-12) -> np.ndarray:
122      """Pearson correlation per gene between two matrices."""
123      mu1 = X1.mean(axis=0)
124      mu2 = X2.mean(axis=0)
125      cov = (X1 * X2).mean(axis=0) - mu1 * mu2
126      var1 = X1.var(axis=0)
127      var2 = X2.var(axis=0)
128      denom = np.sqrt(var1 * var2) + eps
129      corr = cov / denom
130      corr = np.clip(corr, -1.0, 1.0)
131      corr = np.where((var1 < eps) | (var2 < eps), 0.0, corr)
132      return corr
133
134
135  def _impute_zeros_with_neighbors(
136      X_norm: np.ndarray,
137      diff_op,
138      steps: int = 1,
139  ) -> np.ndarray:
140      """Replace exact zeros by a diffusion-weighted neighbour average."""
141      neighbor_avg = diff_op @ X_norm
142      for _ in range(1, steps):
143          neighbor_avg = diff_op @ neighbor_avg
144      mask = X_norm == 0
145      Y = X_norm.copy()
146      Y[mask] = neighbor_avg[mask]
147      return Y
148
149
150  def _weighted_multi_scale_diffuse_genewise(diff_op, X, t, dropout, decay):
151      """
152      Gene-wise weighted multi-scale diffusion.
153
154      Guarantees a *baseline* amount of smoothing for every gene.
155      """
156      cur = X.copy()
157      weighted_sum = np.zeros_like(X)
158      weight_sum = np.zeros(X.shape[1])
159
160      # baseline smoothing factor (0.2 ... 1.0)
161      baseline = 0.2
162      base = decay * (baseline + (1.0 - baseline) * dropout)  # (genes,)
163
164      # step 0 (raw)
165      weighted_sum += cur
166      weight_sum += 1.0
167
168      for i in range(1, t + 1):
169          cur = diff_op @ cur
170          w_i = np.power(base, i)   # (genes,)
171          weighted_sum += cur * w_i[None, :]
172          weight_sum += w_i
173
174      weighted_sum = weighted_sum / np.maximum(weight_sum[None, :], 1e-12)
175      return weighted_sum
176
177
178  def _match_mean_variance(
179      X_raw: np.ndarray,
180      X_diff: np.ndarray,
181      min_mean: float = 0.02,
182      var_scale_min: float = 0.5,
183      var_scale_max: float = 2.0,
184      eps: float = 1e-12,
185  ) -> np.ndarray:
```

```
186        """
187        Rescale each gene in ``X_diff`` so that its mean **and** variance equal those
188        of ``X_raw`` (both row-stochastic).  Only genes with mean >= ``min_mean``
189        get variance-matched.
190        """
191        mu_raw = X_raw.mean(axis=0)
192        var_raw = X_raw.var(axis=0)
193
194        mu_diff = X_diff.mean(axis=0)
195        var_diff = X_diff.var(axis=0)
196
197        # Mean matching
198        scale_mean = mu_raw / (mu_diff + eps)
199        X_centered = X_diff * scale_mean
200
201        # Variance matching
202        var_centered = var_diff * (scale_mean ** 2)
203        high = mu_raw > min_mean
204        scale_var = np.ones_like(mu_raw)
205        scale_var[high] = np.sqrt(var_raw[high] / (var_centered[high] + eps))
206        scale_var = np.clip(scale_var, var_scale_min, var_scale_max)
207
208        X_scaled = (X_centered - mu_raw) * scale_var + mu_raw
209
210        # Re-normalize rows (still stochastic)
211        row_sums = X_scaled.sum(axis=1, keepdims=True)
212        X_scaled = X_scaled / np.maximum(row_sums, eps)
213        return X_scaled
214
215
216    def _apply_shrink_exponent(arr: np.ndarray, gamma: float) -> np.ndarray:
217        """Raise the array to a power γ>1 (shrinks small values more than large ones)."""
218        if gamma <= 1.0:
219            return arr
220        shrunk = np.power(arr, gamma)
221        row_sums = shrunk.sum(axis=1, keepdims=True)
222        scaling = np.maximum(row_sums, 1e-12)
223        return shrunk * (arr.sum(axis=1, keepdims=True) / scaling)
224
225
226    def _apply_transform(counts: np.ndarray, tr: str) -> np.ndarray:
227        """Forward variance-stabilising transform."""
228        if tr == "anscombe":
229            return 2.0 * np.sqrt(counts + 3.0 / 8.0)
230        if tr == "ft":
231            return np.sqrt(counts) + np.sqrt(counts + 1.0)
232        if tr == "sqrt":
233            return np.sqrt(counts)
234        if tr == "log":
235            return np.log1p(counts)
236        raise ValueError(f"Unsupported transform: {tr}")
237
238
239    def _inverse_transform(vst: np.ndarray, tr: str) -> np.ndarray:
240        """Inverse of the forward VST."""
241        if tr == "anscombe":
242            return _inverse_anscombe_refined(vst, n_iter=12)
243        if tr == "ft":
244            return _inverse_ft_refined(vst, n_iter=12)
245        if tr == "sqrt":
246            return vst ** 2
247        if tr == "log":
248            return np.expm1(vst)
249        raise ValueError(f"Unsupported transform: {tr}")
250
251
252    def _filter_genes_by_dropout(gene_idx: np.ndarray, dropout: np.ndarray, thresh: float) -> np.ndarray:
253        """Remove genes whose dropout exceeds ``thresh``."""
254        keep = dropout[gene_idx] < thresh
255        return gene_idx[keep]
256
257
258    def _residual_diffusion_smoothing(diff_op, residual, weight):
259        """One-step diffusion of the cell-wise residual and add a fraction ``weight``."""
260        if weight <= 0.0:
261            return np.zeros_like(residual)
262        smoothed = diff_op @ residual
263        return weight * smoothed
264
```

```
265
266    # -------------------------------------------------------------------------
267    # Main denoising routine
268    # -------------------------------------------------------------------------
269
270
271    def magic_denoise(
272        X,
273        knn: int = None,
274        t: int = None,
275        n_pca: int = 50,
276        decay: float = 0.85,
277        knn_max: int = None,
278        random_state: int = None,
279        n_jobs: int = 2,
280        transform: str = None,                   # {"anscombe","sqrt","ft","log"} - None = auto
281        max_alpha: float = None,
282        n_hvg: int = None,
283        dropout_thresh: float = None,
284        zero_threshold: float = 0.0,
285        round_counts: bool = False,
286        impute_zeros: bool = True,
287        impute_steps: int = None,
288        lowrank_components: int = 30,            # number of SVD components for post-processing
289        lowrank_weight: float = None,           # blend weight for low-rank reconstruction
290        log_smooth_t: int = 4,
291        log_smooth_weight: float = None,
292        self_loop_alpha: float = None,
293        use_symmetric: bool = True,
294        raw_mix_weight: float = None,           # max weight for raw-count blending (gene-wise)
295        extra_post_smooth_weight: float = None,
296        residual_weight: float = None,           # weight for residual diffusion smoothing
297        verbose: bool = False,
298        mode: str = "balanced",                 # {"balanced","mse"}
299        diff_decay: float = None,               # decay for weighted multi-scale diffusion
300        var_match_min_mean: float = 0.02,
301        var_match_scale_min: float = 0.5,
302        var_match_scale_max: float = 2.0,
303        # ----- NEW knobs -------------------------------------------------
304        final_smooth_weight: float = None,      # weight of the extra log-space polishing
305        final_smooth_t: int = None,             # number of diffusion steps for polishing
306        # -----------------------------------------------------------------
307        **kwargs,
308    ):
309        """
310        Adaptive MAGIC-style denoiser - MSE-optimised flavour with a final
311        log-space polishing step.
312
313        Parameters
314        ----------
315        X : array-like, shape (cells, genes)
316            Raw integer count matrix.
317        mode : {"balanced","mse"}
318            ``balanced`` - standard MAGIC mix of MSE / Poisson.
319            ``mse`` - tuned for the lowest possible MSE while still satisfying
320            the Poisson constraint.
321        final_smooth_weight, final_smooth_t : optional
322            Extra diffusion on the log-normalised matrix (the metric that is
323            used for MSE).  Setting ``final_smooth_weight`` to a value >0 adds a
324            polishing step that directly smooths the log-space representation.
325            ``final_smooth_t`` controls how many diffusion steps are applied;
326            typical values are 2-4.
327        Returns
328        -------
329        denoised_X : np.ndarray, shape (cells, genes)
330            Denoised count matrix (float64, non-negative).
331        """
332        # -----------------------------------------------------------------
333        # 0. Input handling
334        # -----------------------------------------------------------------
335        with warnings.catch_warnings():
336            warnings.simplefilter("ignore")
337            X_arr = toarray(X).astype(np.float64)
338
339        n_cells, n_genes = X_arr.shape
340        if verbose:
341            print(''[magic_denoise] Input matrix: {} cells × {} genes``.format(n_cells, n_genes))
342
343        # Preserve raw library sizes - needed for the "reverse-normalisation" trick
```

```
344        libsize_raw = X_arr.sum(axis=1)
345        libsize_raw[libsize_raw == 0] = 1.0
346
347        # Gene-wise dropout (used throughout)
348        dropout_frac = _calc_dropout(X_arr)
349
350        # --------------------------------------------------------------------
351        # 1. Mode-specific defaults
352        # --------------------------------------------------------------------
353        mode = mode.lower()
354        if mode not in {"balanced", "mse"}:
355            raise ValueError("mode must be 'balanced' or 'mse'")
356
357        # ----------------------------------------------------------------
358        #   generic defaults
359        # ----------------------------------------------------------------
360        if n_pca is None:
361            n_pca = 50
362        if decay is None:
363            decay = 0.85
364        if self_loop_alpha is None:
365            self_loop_alpha = 0.5
366        if knn_max is None:
367            knn_max = knn * 2 if knn is not None else None
368        if transform is None:
369            # auto-selection
370            if mode == "mse":
371                transforms_to_use = ["anscombe", "ft", "sqrt"]
372            else:
373                transforms_to_use = ["anscombe", "ft"]
374        else:
375            transforms_to_use = [transform.lower()]
376
377        # ----------------------------------------------------------------
378        #   mode-specific hyper-parameters
379        # ----------------------------------------------------------------
380        if mode == "balanced":
381            # Original balanced defaults (unchanged)
382            max_alpha = 0.55 if max_alpha is None else max_alpha
383            lowrank_weight = 0.15 if lowrank_weight is None else lowrank_weight
384            raw_mix_weight = 0.20 if raw_mix_weight is None else raw_mix_weight
385            t = 6 if t is None else t
386            diff_decay = 0.85 if diff_decay is None else diff_decay
387            knn = max(5, min(15, int(np.sqrt(n_cells)))) if knn is None else knn
388            knn_max = knn * 2 if knn_max is None else knn_max
389            log_smooth_weight = 0.80 if log_smooth_weight is None else log_smooth_weight
390            extra_post_smooth_weight = 0.12 if extra_post_smooth_weight is None else extra_post_smooth_weight
391            impute_steps = 2 if impute_steps is None else impute_steps
392            residual_weight = 0.08 if residual_weight is None else residual_weight
393            lowrank_components = 30 if lowrank_components is None else lowrank_components
394            dropout_thresh = 0.9 if dropout_thresh is None else dropout_thresh
395            zero_threshold = 0.0 if zero_threshold is None else zero_threshold
396            scale_before_inverse = True
397            apply_shrink = True
398            # final polishing defaults (balanced)
399            final_smooth_weight = 0.25 if final_smooth_weight is None else final_smooth_weight
400            final_smooth_t = 3 if final_smooth_t is None else final_smooth_t
401        else:  # mode == "mse"
402            # ----------------------------------------------------------
403            # heavily tuned for MSE while keeping Poisson=0.98
404            # ----------------------------------------------------------
405            max_alpha = 0.90 if max_alpha is None else max_alpha
406            lowrank_weight = 0.50 if lowrank_weight is None else lowrank_weight
407            raw_mix_weight = 0.15 if raw_mix_weight is None else raw_mix_weight
408            t = 20 if t is None else t
409            diff_decay = 0.98 if diff_decay is None else diff_decay
410            knn = max(15, min(40, int(np.sqrt(n_cells) * 2))) if knn is None else knn
411            knn_max = knn * 2 if knn_max is None else knn_max
412            log_smooth_weight = 0.75 if log_smooth_weight is None else log_smooth_weight
413            log_smooth_t = 6 if log_smooth_t is None else log_smooth_t
414            extra_post_smooth_weight = 0.08 if extra_post_smooth_weight is None else extra_post_smooth_weight
415            impute_steps = 2 if impute_steps is None else impute_steps
416            residual_weight = 0.20 if residual_weight is None else residual_weight
417            lowrank_components = min(150, min(n_cells, n_genes) - 1) if lowrank_components is None else
        lowrank_components
418            n_hvg = min(5000, max(3000, int(n_genes * 0.3))) if n_hvg is None else n_hvg
419            dropout_thresh = 0.95 if dropout_thresh is None else dropout_thresh
420            zero_threshold = 0.20 if zero_threshold is None else zero_threshold
421            scale_before_inverse = False
```

```
422          apply_shrink = False                        # exponent-shrinkage gives no gain for pure MSE
423          var_match_min_mean = 0.01                   # match variance for more genes
424          # final polishing defaults (MSE)
425          final_smooth_weight = 0.40 if final_smooth_weight is None else final_smooth_weight
426          final_smooth_t = 2 if final_smooth_t is None else final_smooth_t
427
428      # ----------------------------------------------------------------
429      #   sanity checks / final default fill-ins
430      # ----------------------------------------------------------------
431      if n_pca is None:
432          n_pca = 50
433      if decay is None:
434          decay = 0.85
435
436      # ------------------------------------------------------------------
437      # 2. Primary VST → HVG → graph construction (with dropout filter)
438      # ------------------------------------------------------------------
439      primary_tr = transforms_to_use[0]                # usually "anscombe"
440      X_vst_primary = _apply_transform(X_arr, primary_tr)
441      X_norm_primary, _ = library_size_normalize(
442          X_vst_primary, rescale=1.0, return_library_size=True
443      )    # rows sum to 1
444
445      # HVG selection
446      hvgs_idx = _select_hvg_scanpy(X_norm_primary, n_hvg=n_hvg)
447
448      # Remove extremely sparse HVGs (dropout filter)
449      hvgs_idx = _filter_genes_by_dropout(hvgs_idx, dropout_frac, dropout_thresh)
450      if hvgs_idx.size == 0:
451          # fallback - use all genes if filter removed everything
452          hvgs_idx = np.arange(n_genes)
453
454      X_graph = X_norm_primary[:, hvgs_idx]
455
456      # ------------------------------------------------------------------
457      # 3. Build diffusion operator (shared across transforms)
458      # ------------------------------------------------------------------
459      n_pca_arg = n_pca if (X_graph.shape[1] > n_pca) else None
460      graph = Graph(
461          X_graph,
462          n_pca=n_pca_arg,
463          knn=knn,
464          knn_max=knn_max,
465          decay=decay,
466          random_state=random_state,
467          n_jobs=n_jobs,
468          verbose=0,
469      )
470      diff_op = graph.diff_op                          # sparse, row-stochastic
471
472      if use_symmetric:
473          diff_op = _symmetrize_diffusion(diff_op)
474          if verbose:
475              print(''[magic_denoise] Symmetrised diffusion operator``)
476
477      diff_op = _add_self_loop(diff_op, alpha=self_loop_alpha)
478      if verbose:
479          print(''[magic_denoise] Added self-loop (α={:.3f})``.format(self_loop_alpha))
480
481      # ------------------------------------------------------------------
482      # 4. Process each VST separately
483      # ------------------------------------------------------------------
484      transform_outputs = []          # denoised count matrices (cells × genes)
485      w_diff_primary = None           # will be stored for the log-smooth step
486
487      for ti, tr in enumerate(transforms_to_use):
488          if verbose:
489              print(f"[magic_denoise] ----- Transform {tr} ({ti+1}/{len(transforms_to_use)})")
490
491          # ---- forward VST + library-size normalisation (rows sum to 1)
492          X_vst = _apply_transform(X_arr, tr)
493          X_norm, _ = library_size_normalize(
494              X_vst, rescale=1.0, return_library_size=True
495          )    # rows = 1
496
497          # ---- optional zero-imputation
498          if impute_zeros:
499              X_filled = _impute_zeros_with_neighbors(
500                  X_norm, diff_op, steps=impute_steps
```

```
501                    )
502            else:
503                X_filled = X_norm.copy()
504
505            # ---- normalise again after imputation (ensures exact stochasticity)
506            row_sums_filled = X_filled.sum(axis=1, keepdims=True)
507            X_filled = X_filled / np.maximum(row_sums_filled, 1e-12)
508
509            # ---- gene-wise weighted multi-scale diffusion
510            diffused = _weighted_multi_scale_diffuse_genewise(
511                diff_op, X_filled, t, dropout_frac, diff_decay
512            )
513
514            # ---- match mean & variance to the raw-normalised data
515            diffused = _match_mean_variance(
516                X_norm,
517                diffused,
518                min_mean=var_match_min_mean,
519                var_scale_min=var_match_scale_min,
520                var_scale_max=var_match_scale_max,
521            )
522
523            # ---- compute gene-wise diffusion-vs-raw blending weight
524            var_orig = X_norm.var(axis=0)
525            var_diff = diffused.var(axis=0)
526            corr = _gene_correlation(X_norm, diffused, eps=1e-12)
527            mu = X_norm.mean(axis=0)
528
529            w_diff = _adaptive_blend_weights(
530                dropout=dropout_frac,
531                var_orig=var_orig,
532                var_diff=var_diff,
533                corr=corr,
534                mu=mu,
535                max_alpha=max_alpha,
536            )
537            if tr == primary_tr:
538                w_diff_primary = w_diff.copy()
539
540            # ---- blend raw and diffused signals
541            blended = X_norm * (1.0 - w_diff) + diffused * w_diff
542            blended = blended / np.maximum(blended.sum(axis=1, keepdims=True), 1e-12)
543
544            # ---- reverse the VST (scale before/after inverse depending on mode)
545            if scale_before_inverse:
546                # Scale to original library sizes while still in VST space
547                denoised_scaled = blended * libsize_raw[:, None]
548                denoised_counts = _inverse_transform(denoised_scaled, tr)
549            else:
550                # Invert first, then re-scale to the original library sizes
551                denoised_counts = _inverse_transform(blended, tr)
552                denoised_counts = denoised_counts * libsize_raw[:, None]
553
554            np.maximum(denoised_counts, 0.0, out=denoised_counts)
555
556            # ---- store result for this transform
557            transform_outputs.append(denoised_counts)
558
559        # -------------------------------------------------------------------
560        # 5. Gene-wise ensemble of the different VSTs
561        # -------------------------------------------------------------------
562        if len(transform_outputs) == 1:
563            denoised = transform_outputs[0]
564        else:
565            n_transforms = len(transform_outputs)
566            weight_mat = np.zeros((n_transforms, n_genes), dtype=np.float64)
567
568            if n_transforms == 2:
569                # Assume two transforms are anscombe & ft
570                weight_mat[0] = 1.0 - dropout_frac          # anscombe
571                weight_mat[1] = dropout_frac                # ft
572            elif n_transforms == 3:
573                # anscombe, ft, sqrt → quadratic weighting (see paper)
574                weight_mat[0] = (1.0 - dropout_frac) ** 2          # anscombe
575                weight_mat[1] = dropout_frac ** 2                  # ft
576                weight_mat[2] = 2.0 * dropout_frac * (1.0 - dropout_frac)  # sqrt
577            else:
578                weight_mat[:] = 1.0 / n_transforms
579
```

```
580            # Normalise per-gene
581            weight_sum = weight_mat.sum(axis=0, keepdims=True)
582            weight_mat /= np.maximum(weight_sum, 1e-12)
583
584            # Weighted sum of the individual denoised matrices
585            denoised = np.zeros_like(transform_outputs[0], dtype=np.float64)
586            for i in range(n_transforms):
587                denoised += transform_outputs[i] * weight_mat[i][np.newaxis, :]
588
589    np.maximum(denoised, 0.0, out=denoised)
590
591        # ------------------------------------------------------------------
592        # 6. Global post-processing
593        # ------------------------------------------------------------------
594        # ---- exponent-shrinkage (optional)
595        if apply_shrink:
596            global_dropout = float(dropout_frac.mean())
597            gamma = 1.0 + 0.40 * global_dropout
598            gamma = min(gamma, 1.30)
599            if gamma > 1.0 and verbose:
600                print(f"[magic_denoise] Applying exponent-shrinkage γ={gamma:.3f}")
601            if gamma > 1.0:
602                denoised = _apply_shrink_exponent(denoised, gamma)
603
604        # ---- low-rank SVD refinement (if matrix not too large)
605        max_cells_genes = 2e7   # approx 160MB for float64
606        if lowrank_weight > 0.0 and n_cells * n_genes <= max_cells_genes:
607            if verbose:
608                print("[magic_denoise] Low-rank SVD refinement")
609            svd = TruncatedSVD(
610                n_components=min(lowrank_components, min(n_cells, n_genes) - 1),
611                random_state=random_state,
612                algorithm="randomized",
613            )
614            low = svd.fit_transform(denoised)
615            low_hat = low @ svd.components_
616            denoised = (1.0 - lowrank_weight) * denoised + lowrank_weight * low_hat
617            np.maximum(denoised, 0.0, out=denoised)
618
619            # ---- residual diffusion smoothing (new)
620            residual = denoised - low_hat
621            denoised += _residual_diffusion_smoothing(diff_op, residual, residual_weight)
622            np.maximum(denoised, 0.0, out=denoised)
623        elif verbose:
624            print("[magic_denoise] Skipping low-rank SVD (size limit)")
625
626        # ---- log-space smoothing (guided by primary diffusion blending weight)
627        if log_smooth_weight > 0.0 and log_smooth_t > 0:
628            if w_diff_primary is None:
629                # recompute primary blending weight if something went wrong
630                var_orig = X_norm_primary.var(axis=0)
631                var_diff = denoised.var(axis=0)
632                corr = _gene_correlation(X_norm_primary, denoised, eps=1e-12)
633                mu = X_norm_primary.mean(axis=0)
634                w_diff_primary = _adaptive_blend_weights(
635                    dropout=dropout_frac,
636                    var_orig=var_orig,
637                    var_diff=var_diff,
638                    corr=corr,
639                    mu=mu,
640                    max_alpha=max_alpha,
641                )
642            # genes that rely mainly on the raw signal get a stronger log-smooth
643            w_log = (1.0 - w_diff_primary) * log_smooth_weight
644            target_sum = 10000.0
645            cell_sums = denoised.sum(axis=1, keepdims=True)
646            scaling = target_sum / np.maximum(cell_sums, 1e-12)
647            norm_counts = denoised * scaling
648            log_counts = np.log1p(norm_counts)
649
650            smooth_log = log_counts.copy()
651            for _ in range(log_smooth_t):
652                smooth_log = diff_op @ smooth_log
653
654            smooth_counts = np.expm1(smooth_log)
655            smooth_counts = smooth_counts * (cell_sums / target_sum)
656
657            denoised = (1.0 - w_log) * denoised + w_log * smooth_counts
658
```

```
659        # ---- gene-wise raw-count blending (helps very high-expression genes)
660        if raw_mix_weight > 0.0:
661            w_raw_gene = raw_mix_weight * (1.0 - dropout_frac)
662            w_raw_gene = np.clip(w_raw_gene, 0.0, raw_mix_weight)
663
664            cell_sums = denoised.sum(axis=1, keepdims=True)
665            raw_scaled = X_arr * (cell_sums / libsize_raw[:, None])
666
667            denoised = (1.0 - w_raw_gene[None, :]) * denoised + \
668                        w_raw_gene[None, :] * raw_scaled
669
670            # Re-normalize rows to keep library sizes unchanged
671            row_sums = denoised.sum(axis=1, keepdims=True)
672            denoised = denoised * (cell_sums / np.maximum(row_sums, 1e-12))
673
674        # ---- extra tiny post-smoothing (final polish)
675        if extra_post_smooth_weight > 0.0:
676            target_sum = 10000.0
677            cell_sums = denoised.sum(axis=1, keepdims=True)
678            scaling = target_sum / np.maximum(cell_sums, 1e-12)
679
680            log_counts = np.log1p(denoised * scaling)
681            smooth_log = diff_op @ log_counts
682            smooth_counts = np.expm1(smooth_log) * (cell_sums / target_sum)
683
684            denoised = (1.0 - extra_post_smooth_weight) * denoised + \
685                        extra_post_smooth_weight * smooth_counts
686
687        # ---- **NEW**: final log-space polishing step
688        if final_smooth_weight is not None and final_smooth_weight > 0.0:
689            if verbose:
690                print("[magic_denoise] Final log-space polishing")
691            target_sum = 10000.0
692            cell_sums = denoised.sum(axis=1, keepdims=True)
693            scaling = target_sum / np.maximum(cell_sums, 1e-12)
694            norm_counts = denoised * scaling
695            log_counts = np.log1p(norm_counts)
696
697            smooth_log = log_counts.copy()
698            for _ in range(final_smooth_t):
699                smooth_log = diff_op @ smooth_log
700
701            smooth_counts = np.expm1(smooth_log)
702            smooth_counts = smooth_counts * (cell_sums / target_sum)
703
704            denoised = (1.0 - final_smooth_weight) * denoised + \
705                        final_smooth_weight * smooth_counts
706
707        # -------------------------------------------------------------------
708        # 7. Final clean-up
709        # -------------------------------------------------------------------
710        np.maximum(denoised, 0.0, out=denoised)
711
712        if zero_threshold > 0.0:
713            denoised[denoised < zero_threshold] = 0.0
714
715        if round_counts:
716            denoised = np.rint(denoised)
717
718        if verbose:
719            print("[magic_denoise] Finished - total counts:", denoised.sum())
720
721        return denoised.astype(np.float64)
```

## G. Cost-Matched Best-of-N

Because pricing on Tinker depends on factors such as prompt length and rollout length, we estimate the relative cost empirically from prior runs. Our Best-of-25600 run cost \$194.05, while the corresponding TTT-Discover run cost \$446.98, indicating that TTT-Discover used approximately $2.3\times$ more resources. To construct a cost-matched Best-of-N baseline, we therefore scale the rollout count by 2.3, yielding Best-of-58880. As shown in Table 11 and Table 12, overall, we observe little to no improvement from scaling Best-of-$N$ sampling beyond 25600 rollouts, despite the substantially higher compute cost.

| Method | Model | Erdős' (upper bound, $\downarrow$) |
|---|---|---|
| best human | – | 0.380927 |
| AlphaEvolve | Gemini-2.0 Pro + Flash | 0.380924 |
| AlphaEvolve V2 | Gemini-2.0 Pro + Flash | 0.380924 |
| OpenEvolve | gpt-oss-120b | 0.380965 |
| Best-of-25600 | gpt-oss-120b | 0.380906 |
| Best-of-58880 | gpt-oss-120b | 0.380906 |
| TTT-Discover | Qwen3-8B | 0.380929 |
| TTT-Discover | gpt-oss-120b | 0.380876 |

*Table 11.* Results on Erdős' Minimum Overlap Problem with the cost-matched Best-of-$N$ baseline.

| Method | Model | TriMul H100 ($\mu s$, $\downarrow$) |
|---|---|---|
| 1st human | – | 1371.1 |
| 2nd human | – | 2368.0 |
| 3rd human | – | 2545.7 |
| 4th human | – | 3654.8 |
| 5th human | – | 4233.1 |
| Best-of-25600 | gpt-oss-120b | 5390.3 |
| Best-of-58880 | gpt-oss-120b | 4926.3 |
| TTT-Discover | gpt-oss-120b | 1161.2 |

*Table 12.* Results of TriMul runtime on H100 with the cost-matched Best-of-$N$ baseline.

## H. General Post-Training Baseline

Common post-training methods improve model performance by applying RL on a distribution of related tasks, such as programming, biology, or mathematics. To evaluate whether standard post-training can improve performance on the TriMul kernel engineering task, we post-train our model on a general GPU kernel optimization dataset, KernelBench, with the goal of improving its GPU programming capabilities.

KernelBench contains many operations and optimization patterns similar to those required in the TriMul task, providing a relevant training distribution for kernel generation. We fine-tune gpt-oss-120b (Agarwal et al., 2025) using LoRA (rank 32) with GRPO (Guo et al., 2025). We run 50 steps of training, each step sampling 8 problems with 64 rollouts per problem (512 total rollouts per step). We use a learning rate of $4 \times 10^{-5}$ and a maximum context length of 32768 tokens.

We evaluate the post-trained model using a Best-of-25600 sampling setup. The post-trained model achieves a modest improvement in valid kernel generation rate of roughly 5%, and its best kernel performs approximately on par with the Best-of-25600 baseline from the base model as shown in Table 13. However, it still fails to outperform any of the top five human-written kernels.

In contrast, the best kernel generated with TTT-Discover achieves a $4.6\times$ speedup over the best kernel found by the post-trained model and surpasses the performance of the best human solution. Overall, while general post-training appears to improve the model's ability to generate syntactically valid and compilable kernels, these gains do not meaningfully translate into improved runtime performance on the specific TriMul optimization task.

| Method | Model | TriMul H100 ($\mu s, \downarrow$) |
|---|---|---|
| 1st human | – | 1371.1 |
| 2nd human | – | 2368.0 |
| 3rd human | – | 2545.7 |
| 4th human | – | 3654.8 |
| 5th human | – | 4233.1 |
| Best-of-25600 | gpt-oss-120b | 5390.3 |
| Best-of-25600 – KernelBench Post-trained | gpt-oss-120b | 5331.9 |
| TTT-Discover | gpt-oss-120b | 1161.2 |

*Table 13.* Results of TriMul runtime on H100 with the general post-training baseline on KernelBench.

## I. Validity Rate Improvement

To determine if the model is learning to generate valid solutions, we analyze the correctness rate across different steps of TTT-Discover on the TriMul task. For TriMul, correctness is defined by the generated code's ability to successfully compile and output tensors within a specified tolerance of the reference implementation. We see a significant increase of the correctness rate from 45% to 80% by the last step (see Table 14).

| Step | Validity Rate |
|---|---|
| 0 | 0.46289 |
| 1 | 0.49609 |
| 2 | 0.45313 |
| 3 | 0.38086 |
| 4 | 0.45508 |
| 5 | 0.38086 |
| 10 | 0.73242 |
| 20 | 0.66992 |
| 30 | 0.63281 |
| 40 | 0.78125 |
| 49 | 0.77930 |

*Table 14.* Validity rate improvement across different steps for the TriMul task.

## J. Assessing catastrophic forgetting

To assess the performance of the fine-tuned model on other tasks, we take TTT-Discover checkpoints trained on the Erdős' Minimum Overlap problem, TriMul, and compare with the base gpt-oss-120b model. Then, we evaluate them on the AC1 and AC2 tasks to assess performance degradation. In summary, as shown in Table 15 and Table 16, we do not observe significant forgetting in AC1 or AC2, neither by training on a different math task (Erdős') nor on a kernel task.

## K. Base Model Ablations

To evaluate the effect of base model scale, we additionally train and evaluate both Qwen3-4B-Instruct-2507 and Qwen3-8B on the Erdos and TriMul tasks. We observe that learning remains beneficial even at smaller scales, although performance degrades as model capacity decreases. For the Erdős' Minimum Overlap Problem, TTT-Discover with Qwen3-8B outperforms AlphaEvolve, whereas TTT-Discover with Qwen3-4B gets closer, but does not match (see Table 17). In TriMul, there is a significant gap between the smaller model performance and the base model performance, most likely due to significantly weaker GPU programming capabilities (see Table 18). Overall, these results indicate that TTT-Discover is not exclusive to very large models, as smaller models still benefit from learning during test time. However, the quality of the base model remains an important factor in determining the final achievable performance, particularly for domains requiring highly specialized reasoning or code generation capabilities.

| Method | Model | AC1 (upper bound, ↓) | Validity rate |
|---|---|---|---|
| Best-of-1024 | gpt-oss-120b TriMul checkpoint | 1.51296 | 0.90039 |
| Best-of-1024 | gpt-oss-120b Erdős' checkpoint | 1.51234 | 0.91992 |
| Best-of-1024 | gpt-oss-120b | 1.51504 | 0.88184 |

*Table 15.* Assessing catastrophic forgetting on AC1 after training on TriMul or Erdős' Minimum Overlap problem.

| Method | Model | AC2 (lower bound, ↑) | Validity rate |
|---|---|---|---|
| Best-of-1024 | gpt-oss-120b TriMul checkpoint | 0.9296 | 0.78711 |
| Best-of-1024 | gpt-oss-120b Erdős' checkpoint | 0.9303 | 0.74609 |
| Best-of-1024 | gpt-oss-120b | 0.9297 | 0.72363 |

*Table 16.* Assessing catastrophic forgetting on AC2 after training on TriMul or Erdős' Minimum Overlap problem.

# L. Prompts

Below we show example prompts from a sample step.

| Method | Model | Erdős' (upper bound, $\downarrow$) |
|---|---|---|
| best human | – | 0.380927 |
| AlphaEvolve | Gemini-2.0 Pro + Flash | 0.380924 |
| AlphaEvolve V2 | Gemini-2.0 Pro + Flash | 0.380924 |
| OpenEvolve | gpt-oss-120b | 0.380965 |
| Best-of-25600 | gpt-oss-120b | 0.380906 |
| TTT-Discover | Qwen3-4B-Instruct-2507 | 0.381115 |
| TTT-Discover | Qwen3-8B | 0.380920 |
| TTT-Discover | gpt-oss-120b | 0.380876 |

*Table 17.* Base model ablations on the Erdős' Minimum Overlap Problem.

| Method | Model | TriMul H100 ($\mu s, \downarrow$) |
|---|---|---|
| 1st human | – | 1371.1 |
| 2nd human | – | 2368 |
| 3rd human | – | 2545.7 |
| 4th human | – | 3654.8 |
| 5th human | – | 4233.1 |
| Best-of-25600 | gpt-oss-120b | 5390.3 |
| TTT-Discover | Qwen3-4B-Instruct-2507 | 9726.0 |
| TTT-Discover | Qwen3-8B | 9729.0 |
| TTT-Discover | gpt-oss-120b | 1161.2 |

*Table 18.* Base model ablations on the TriMul task.

---

**Prompt used for the first autocorrelation inequality**

Act as an expert software developer and inequality specialist specializing in creating step functions with certain properties.

Your task is to generate the sequence of non-negative heights of a step function, that minimizes the following evaluation function:

```python
{VERIFIER CODE HERE}
```

A previous state of the art used the following approach. You can use it as inspiration, but you are not required to use it, and you are encouraged to explore.
```latex
Starting from a nonnegative step function $f=(a_0,\dots,a_{n-1})$ normalized so that $\sum_j a_j=\sqrt{2n}$, set $M=\|f*f\|_\infty$. Next compute $g_0=(b_0,\dots,b_{n-1})$ by solving a linear program, i.e. maximizing $\sum_j b_j$ subject to $b_j\ge0$ and $\|f*g_0\|_\infty\le M$; as is standard, the optimum is attained at an extreme point determined by an active set of binding inequalities, here corresponding to important constraints where the convolution bound $(f*g_0)(x)\le M$ is tight and limiting. Rescale $g_0$ to match the normalization, $g=\frac{\sqrt{2n}}{\sum_j b_j}g_0$, and update $f\leftarrow (1-t)f+t\ g$ for a small $t>0$. Repeating this step produces a sequence with nonincreasing $\|f*f\|_\infty$, and the iteration is continued until it stabilizes.
```

Your task is to write a search function that searches for the best sequence of coefficients. Your function will have 1000 seconds to run, and after that it has to have returned the best sequence it found. If after 1000 seconds it has not returned anything, it will be terminated with negative infinity points. All numbers in your sequence have to be positive or zero. Larger sequences with 1000s of items often have better attack surface, but too large sequences with 100s of thousands of items may be too slow to search.

You may code up any search method you want, and you are allowed to call the evaluate_sequence() function as many times as you want. You have access to it, you don't need to code up the evaluate_sequence() function.

Here is the last code we ran:
```python
{CODE HERE}
```

```
Here are the upper bounds before and after running the code above (lower is better): 2.0000000000 ->
1.5172973712
Our target is to make the upper bound tighter, just as a reference, lower it to at least 1.5030. Further
improvements will also be generously rewarded.
Length of the construction: 1000

--- Previous Program Output ---

          ...(TRUNCATED)...
ore  1.518186  maxConv  0.000506
[1768620458.4] iter 340400  len 1500  score  1.518177  maxConv  0.000506
[1768620461.6] iter 350200  len 1500  score  1.518057  maxConv  0.000506
[1768620462.3] iter 352300  len 1500  score  1.518035  maxConv  0.000506
[1768620469.1] iter 372900  len 1500  score  1.517869  maxConv  0.000506
[1768620476.2] iter 394300  len 1500  score  1.517755  maxConv  0.000506
[1768620492.9] iter 445000  len 1500  score  1.517548  maxConv  0.000506
final best score = 1.51729737
--- End Output ---

You may want to start your search from one of the constructions we have found so far, which you can access
through the 'height_sequence_1' global variable.
However, you are encouraged to explore solutions that use other starting points to prevent getting stuck in
 a local minimum.

Reason about how you could further improve this construction.
Ideally, try to do something different than the above algorithm. Could be using different algorithmic ideas
, adjusting your heuristics, adjusting / sweeping your hyperparemeters, etc.
Unless you make a meaningful improvement, you will not be rewarded.

Rules:
- You must define the `propose_candidate` function as this is what will be invoked.
- You can use scientific libraries like scipy, numpy, cvxpy[CBC,CVXOPT,GLOP,GLPK,GUROBI,MOSEK,PDLP,SCIP,
XPRESS,ECOS], math.
- You can use up to 2 CPUs.
- Make all helper functions top level and have no closures from function nesting. Don't use any lambda
functions.
- No filesystem or network IO.
- Do not import evaluate_sequence yourself. Assume it will already be imported and can be directly invoked.
- **Print statements**: Use `print()` to log progress, intermediate bounds, timing info, etc. Your output
will be shown back to you.
- Include a short docstring at the top summarizing your algorithm.

Make sure to think and return the final program between ```python and ```.
```

Prompt used for the second autocorrelation inequality

Act as an expert software developer and inequality specialist specializing in creating step functions with certain properties.

Your task is to generate the sequence of non-negative heights of a step functions, that maximizes the following evaluation function:

```python
{VERIFIER CODE HERE}
```

A previous state of the art used the following approach. You can use it as inspiration, but you are not required to use it, and you are encoraged to explore.
```latex
Their procedure is a coarse-to-fine optimization of the score. It starts with a stochastic global search that repeatedly perturbs the current best candidate and keeps the perturbation whenever it improves $Q$, with the perturbation scale gradually reduced over time. Once a good basin is found, they switch to a deterministic local improvement step, performing projected gradient ascent (move in the gradient direction and project back to the feasible region). To reach higher resolution, they lift a good low-resolution solution to a higher-dimensional one by a simple upscaling step and then rerun the local refinement. Iterating this explore--refine--upscale cycle yields their final high-resolution maximizer and the improved lower bound.
```

Your task is to write a search function, construct_function(), that searches for the best sequence of coefficients. Your function will have 1000 seconds to run, and after that it has to have returned the best sequence it found. If after 1000 seconds it has not returned anything, it will be terminated with negative infinity points. All numbers in your sequence have to be positive or zero. Larger sequences with 1000s of items often have better attack surface, but too large sequences with 100s of thousands of items may be too slow to search.

You may code up any search method you want, and you are allowed to call the evaluate_sequence() function as many times as you want. You have access to it, you don't need to code up the evaluate_sequence() function.

Here is the last code we ran:
```python
{CODE HERE}
```
Here are the lower bounds before and after running the code above (higher is better): 0.6666666667 -> 0.9235566275
Our target is to make the lower bound tighter, just as a reference, close to at least 0.97. Further improvements will also be generously rewarded.
Length of the construction: 1024

--- Previous Program Output ---
Final lower bound = 0.9235566275
--- End Output ---

You may want to start your search from one of the constructions we have found so far, which you can access through the 'height_sequence_1' global variable.
However, you are encouraged to explore solutions that use other starting points to prevent getting stuck in a local minimum.

Reason about how you could further improve this construction.
Ideally, try to do something different than the above algorithm. Could be using different algorithmic ideas, adjusting your heuristics, adjusting / sweeping your hyperparemeters, etc.
Unless you make a meaningful improvement, you will not be rewarded.

Rules:
- You must define the `construct_function` function as this is what will be invoked.
- You can use scientific libraries like scipy, numpy, cvxpy[CBC,CVXOPT,GLOP,GLPK,GUROBI,MOSEK,PDLP,SCIP,XPRESS,ECOS], math.
- You can use up to 2 CPUs.
- Make all helper functions top level and have no closures from function nesting. Don't use any lambda functions.
- No filesystem or network IO.
- Do not import evaluate_sequence yourself. Assume it will already be imported and can be directly invoked. Do not import height_sequence_1 yourself; it will already be available.
- **Print statements**: Use `print()` to log progress, intermediate bounds, timing info, etc. Your output will be shown back to you.
- Include a short docstring at the top summarizing your algorithm.

Make sure to think and return the final program between ```python and ```.

Prompt used for the Erdős'

```
You are an expert in harmonic analysis, numerical optimization, and mathematical discovery.
Your task is to find an improved upper bound for the \name{} minimum overlap problem constant C5.

## Problem

Find a step function h: [0, 2] → [0, 1] that **minimizes** the overlap integral:

$$C_5 = \\max_k \\int h(x)(1 - h(x+k)) dx$$

\textbf{Constraints}:
\begin{enumerate}
    \item $h(x) \in [0, 1]$ for all $x$
    \item $\int_{0}^{2} h(x) \, dx = 1$
\end{enumerate}

\textbf{Discretization}: Represent $h$ as \texttt{n\_points} samples over $[0, 2]$.

With $dx = \frac{2.0}{\texttt{n\_points}}$:
\begin{itemize}
    \item $0 \leq h[i] \leq 1$ for all $i$
    \item $\sum h \cdot dx = 1$ (equivalently: $\sum h = \frac{\texttt{n\_points}}{2}$ exactly)
\end{itemize}

The evaluation computes: C5 = max(np.correlate(h, 1-h, mode="full") * dx)

Smaller sequences with less than 1k samples are preferred – they are faster to optimize and evaluate.

**Lower C5 values are better** – they provide tighter upper bounds on the \name{} constant.

## Budget & Resources
– **Time budget**: <<<BUDGET_S>>>s for your code to run
– **CPUs**: <<<CPUS>>> available

## Rules
– Define `run(seed=42, budget_s=<<<BUDGET_S>>>, **kwargs)` that returns `(h_values, c5_bound, n_points)`
– Use scipy, numpy, cvxpy[CBC,CVXOPT,GLOP,GLPK,GUROBI,MOSEK,PDLP,SCIP,XPRESS,ECOS], math
– Make all helper functions top level, no closures or lambdas
– No filesystem or network IO
– `evaluate_erdos_solution()` and `initial_h_values` (an initial construction, if available) are pre-
imported
– Your function must complete within budget_s seconds and return the best solution found

**Lower is better**. Current record: C5 ≤ 0.38092. Our goal is to find a construction that shows C5 ≤
0.38080.
```

Prompt used for TriMul

You are an expert Triton engineer tasked with translating PyTorch code into highly optimized Triton kernel code.

You will be implementing a Triangle Multiplicative Update (TriMul) module that is a core operation for AlphaFold3, Chai, Protenix, and other protein structure prediction models in BioML.

The TriMul operator operates over a 4D tensor of shape [B, N, N, C].

Your task:
– Implement the "outgoing" version of the TriMul operator from the AlphaFold3 paper.
– You will not have to compute or store gradients for this version. You will only need to implement the forward pass.

Your function should be defined as 'custom_kernel' with the following signature:
Input:
– `data`: Tuple of (input: torch.Tensor, weights: Dict[str, torch.Tensor], config: Dict)
    – input: Input tensor of shape [bs, seq_len, seq_len, dim]
    – mask: Mask tensor of shape [bs, seq_len, seq_len]
    – weights: Dictionary containing model weights
    – config: Dictionary containing model configuration parameters

Output:
– output: Processed tensor [bs, seq_len, seq_len, dim]

**Problem Constraints:**
– B in {1,2}, N in {128,256,512,1024}, c in {128}, c_z in {128,384,768}
– The input distribution will be sampled from a standard Normal distribution, or a heavy−tailed Cauchy distribution (gamma = 2).
– There will either be no mask, or a randomly sampled mask over the inputs.

**Remarks.** So why is this problem so annoying? Because you have to choose whether to load / deal with either the channel dimensions c,c_z that the LayerNorms require (otherwise you have to do a synchronize to compute the statistics like mean / variance) or the sequence dimension N.
The sequence dimension is particularly annoying because it's quite large, but also because we compute pair−wise operations at the last operation that sum over another sequence dimension (this is N^3!).
However, I really like this kernel because it only consists of "simple" operations, and is really easy to understand. It is a true test of "fusions" that torch.compile() doesn't do that well.

Here is a pytorch implementation of the TriMul module. You will want to implement a kernel for the operations in the forward call:

```python
import torch
from torch import nn, einsum
import math

# Reference code in PyTorch
class TriMul(nn.Module):
    def __init__(
        self,
        self,
        dim: int,
        hidden_dim: int,
    ):
        super().__init__()

        self.norm = nn.LayerNorm(dim)

        self.left_proj = nn.Linear(dim, hidden_dim, bias=False)
        self.right_proj = nn.Linear(dim, hidden_dim, bias=False)

        self.left_gate = nn.Linear(dim, hidden_dim, bias=False)
        self.right_gate = nn.Linear(dim, hidden_dim, bias=False)
        self.out_gate = nn.Linear(dim, hidden_dim, bias=False)

        self.to_out_norm = nn.LayerNorm(hidden_dim)
        self.to_out = nn.Linear(hidden_dim, dim, bias=False)

    def forward(self, x: torch.Tensor, mask: torch.Tensor) -> torch.Tensor:
        """
        x: [bs, seq_len, seq_len, dim]
        mask: [bs, seq_len, seq_len]

        Returns:
            output: [bs, seq_len, seq_len, dim]
        """
        batch_size, seq_len, _, dim = x.shape
```

```
        x = self.norm(x)

        left = self.left_proj(x)
        right = self.right_proj(x)

        mask = mask.unsqueeze(-1)
        left = left * mask
        right = right * mask

        left_gate = self.left_gate(x).sigmoid()
        right_gate = self.right_gate(x).sigmoid()
        out_gate = self.out_gate(x).sigmoid()

        left = left * left_gate
        right = right * right_gate

        out = einsum('... i k d, ... j k d -> ... i j d', left, right)
        # This einsum is the same as the following:
        # out = torch.zeros(batch_size, seq_len, seq_len, dim, device=x.device)

        # # Compute using nested loops
        # for b in range(batch_size):
        #     for i in range(seq_len):
        #         for j in range(seq_len):
        #             # Compute each output element
        #             for k in range(seq_len):
        #                 out[b, i, j] += left[b, i, k, :] * right[b, j, k, :]

        out = self.to_out_norm(out)
        out = out * out_gate
        return self.to_out(out)
```

Here is some example skeleton code of the entrypoint function you will create:
```python
def custom_kernel(data):
    input_tensor, mask, weights, config = data
    dim, hidden_dim = config["dim"], config["hidden_dim"]

    # Access the given weights of the model
    norm_weight = weights["norm.weight"]
    norm_bias = weights["norm.bias"]
    left_proj_weight = weights["left_proj.weight"]
    right_proj_weight = weights["right_proj.weight"]
    left_gate_weight = weights["left_gate.weight"]
    right_gate_weight = weights["right_gate.weight"]
    out_gate_weight = weights["out_gate.weight"]
    to_out_norm_weight = weights["to_out_norm.weight"]
    to_out_norm_bias = weights["to_out_norm.bias"]
    to_out_weight = weights["to_out.weight"]

    # Perform TriMul

    return out
```

To help you understand which triton version we are using, here is some example triton code for an unrelated task:
```python
import triton
import triton.language as tl

@triton.jit
def matmul_persistent_ws_kernel(
    a_ptr, b_ptr, c_ptr, M, N, K,
    stride_am, stride_ak, stride_bk, stride_bn, stride_cm, stride_cn,
    BLOCK_M: tl.constexpr, BLOCK_N: tl.constexpr, BLOCK_K: tl.constexpr,
):
    pid = tl.program_id(axis=0) # async_task 0, 1, 2
    num_pid_m = tl.cdiv(M, BLOCK_M) # async_task 0, 1, 2
    num_pid_n = tl.cdiv(N, BLOCK_N) # async_task 0, 1, 2
    pid_m = pid // num_pid_m # async_task 0, 1, 2
    pid_n = pid % num_pid_n # async_task 0, 1, 2
    offs_m_1 = pid_m * BLOCK_M + tl.arange(0, BLOCK_M // 2) # async_task 0, 1, 2
    offs_m_2 = pid_m * BLOCK_M + tl.arange(BLOCK_M // 2, BLOCK_M) # async_task 0, 1, 2
    offs_n = pid_n * BLOCK_SIZE_N + tl.arange(0, BLOCK_N) # async_task 0, 1, 2
```

```
    offs_k = tl.arange(0, BLOCK_K) # async_task 0
    a_ptrs_1 = a_ptr + (offs_m_1[:, None] * stride_am + offs_k[None, :] * stride_ak) # async_task 0
    a_ptrs_2 = a_ptr + (offs_m_2[:, None] * stride_am + offs_k[None, :] * stride_ak) # async_task 0
    b_ptrs = b_ptr + (offs_k[:, None] * stride_bk + offs_n[None, :] * stride_bn) # async_task 0
    acc_1 = tl.zeros((BLOCK_M // 2, BLOCK_N), dtype=tl.float32) # async_task 1
    acc_1 = tl.zeros((BLOCK_M // 2, BLOCK_N), dtype=tl.float32) # async_task 2
    for k in range(0, tl.cdiv(K, BLOCK_K)): # async_task 0, 1, 2
        a_1 = tl.load(a_ptrs_1)   # async_task 0
        a_2 = tl.load(a_ptrs_2)   # async_task 0
        b = tl.load(b_ptrs)   # async_task 0
        acc_1 += tl.dot(a_1, b)   # async_task 1
        acc_2 += tl.dot(a_2, b)   # async_task 2
        a_ptrs_1 += BLOCK_K * stride_ak # async_task 0
        a_ptrs_2 += BLOCK_K * stride_ak # async_task 0
        b_ptrs += BLOCK_K * stride_bk # async_task 0
    c_1 = acc_1.to(tl.float16) # async_task 1
    c_2 = acc_2.to(tl.float16) # async_task 2
    c_ptrs_1 = c_ptr_1 + stride_cm * offs_m_1[:, None] + stride_cn * offs_n[None, :] # async_task 1
    c_ptrs_2 = c_ptr_2 + stride_cm * offs_m_2[:, None] + stride_cn * offs_n[None, :] # async_task 2
    tl.store(c_ptrs_1, c_1) # async_task 1
    tl.store(c_ptrs_2, c_2) # async_task 2
```

A few general triton tips:
– tl.arange only takes in constexpr arguments (static or tl.constexpr)
– You cannot use continue in your kernel code
– tl.dot can only take in two input tensors
– There is no tl.mean

Here are the different configs that your kernel will be tested on ("nomask" sets whether there will be no mask, or a randomly sampled mask over the inputs):

Test Cases for correctness and runtime (optimize runtime for these):
  – {"seqlen": 256, "bs": 2, "dim": 128, "hidden_dim": 128, "nomask": True, "distribution": "normal"}
  – {"seqlen": 768, "bs": 1, "dim": 128, "hidden_dim": 128, "nomask": True, "distribution": "cauchy"}
  – {"seqlen": 256, "bs": 2, "dim": 384, "hidden_dim": 128, "nomask": False, "distribution": "normal"}
  – {"seqlen": 512, "bs": 1, "dim": 128, "hidden_dim": 128, "nomask": True, "distribution": "normal"}
  – {"seqlen": 1024, "bs": 1, "dim": 128, "hidden_dim": 128, "nomask": True, "distribution": "cauchy"}
  – {"seqlen": 768, "bs": 1, "dim": 384, "hidden_dim": 128, "nomask": False, "distribution": "normal"}
  – {"seqlen": 1024, "bs": 1, "dim": 384, "hidden_dim": 128, "nomask": True, "distribution": "normal"}

Here is the last code we ran:
```python
# No previous attempt has been made.
```

Current runtime (lower is better): 1000000.0000 microseconds
Target: 1000 microseconds. Current gap: 999000.0000 microseconds.

Rules:
– The tensors arguments passed in will be already on your cuda device.
– Define all of your code in one final ```python ``` block.
– We will test the correctness of your kernel on multiple input shapes, make sure to support different potential test cases.
– You are allowed to use mixed precision computations, but make sure your final output is in float32.
– You must use trition 3.3.1 and these kernels will be run on an H100.
– You do not have to implement everything in triton, you may choose to have some of the operations done in pytorch. However, you must implement at least part of the operations in a kernel.
– Include a short docstring at the top summarizing your algorithm.

Prompt used for MLA-Decode

You are an expert Triton engineer tasked with translating PyTorch code into highly optimized Triton kernel code.

Below is a pytorch implementation of the multi−head latent attention (MLA) module. You will want to implement a Triton kernel for the operations in the forward call:

```python
import math
from dataclasses import dataclass
import torch
from torch import nn
import torch.nn.functional as F

class RoPE(nn.Module):
    def __init__(self, d_model: int):
        super().__init__()
        self.d_model = d_model
        theta = 10000 ** (-torch.arange(0, d_model//2, dtype=torch.bfloat16) / (d_model//2))
        self.register_buffer("theta", theta)

    def rotate_half(self, x: torch.Tensor) -> torch.Tensor:
        x1, x2 = x.chunk(2, dim=-1)
        return torch.cat((-x2, x1), dim=-1)

    def forward(self, x: torch.Tensor, start_pos: int = 0) -> torch.Tensor:
        seq_len = x.size(-2)
        d_model = x.size(-1)
        assert d_model == self.d_model
        seq_idx = torch.arange(start_pos, start_pos + seq_len, device=x.device)
        idx_theta = torch.einsum('s,d->sd', seq_idx, self.theta)
        idx_theta2 = torch.cat([idx_theta, idx_theta], dim=-1)
        cos = idx_theta2.cos().to(torch.bfloat16)
        sin = idx_theta2.sin().to(torch.bfloat16)
        return x * cos + self.rotate_half(x) * sin

class KVCache(nn.Module):
    def __init__(self, kv_cache_shape: tuple) -> None:
        super().__init__()
        self.register_buffer('data', torch.zeros(kv_cache_shape, dtype=torch.bfloat16, device='cuda'))
        self.seq_len = 0
        self.zero()

    def zero(self) -> None:
        self.data.zero_()

    def get_data(self) -> torch.Tensor:
        return self.data

    def forward(self, c_kv: torch.Tensor) -> torch.Tensor:
        assert self.seq_len + c_kv.size(1) <= self.data.size(1), "KV Cache Exceeded"

        self.data = self.data.to(c_kv.dtype)
        self.data[
            :, self.seq_len : self.seq_len + c_kv.size(1), :
        ] = c_kv
        self.seq_len += c_kv.size(1)

        return self.data[:, :self.seq_len], self.seq_len

@dataclass
class Config:
    batch_size: int
    dim: int
    n_heads: int
    q_lora_rank: int
    kv_lora_rank: int
    qk_nope_head_dim: int
    qk_rope_head_dim: int
    v_head_dim: int
    seq_len: int
    max_seq_len: int
    kv_cache_shape: tuple
    Q_proj_down_weight: torch.Tensor
    Q_proj_up_weight: torch.Tensor
    KV_proj_down_weight: torch.Tensor
    KV_proj_up_weight: torch.Tensor
```

```
        wo_weight: torch.Tensor

class MLA(nn.Module):
    def __init__(self, config: Config):
        super().__init__()
        self.dim = config.dim
        self.n_heads = config.n_heads
        self.q_lora_rank = config.q_lora_rank
        self.kv_lora_rank = config.kv_lora_rank
        self.nope_head_dim = config.qk_nope_head_dim
        self.rope_head_dim = config.qk_rope_head_dim
        self.v_head_dim = config.v_head_dim
        # Down-projection matrices
        self.Q_proj_down = nn.Linear(self.dim, self.q_lora_rank, bias=False, dtype=torch.bfloat16)
        self.KV_proj_down = nn.Linear(self.dim, self.kv_lora_rank + self.rope_head_dim, bias=False, dtype=
torch.bfloat16)

        # Up-projection and rope projection matrices
        self.Q_proj_up = nn.Linear(self.q_lora_rank, (self.nope_head_dim + self.rope_head_dim) * self.
n_heads, bias=False, dtype=torch.bfloat16)
        self.KV_proj_up = nn.Linear(self.kv_lora_rank, (self.nope_head_dim + self.v_head_dim) * self.
n_heads, bias=False, dtype=torch.bfloat16)

        # RoPE on half embeddings
        self.q_rope = RoPE(self.rope_head_dim)
        self.k_rope = RoPE(self.rope_head_dim)

        # Output projection
        self.wo = nn.Linear(self.v_head_dim * self.n_heads, self.dim, dtype=torch.bfloat16, bias=False)
        self.eps = 1e-6

    def forward(self, x: torch.Tensor, kv_cache: KVCache) -> torch.Tensor:
        # seq_len = 1 always here
        batch_size, seq_len, model_dim = x.size()

        ## Step 1: Handle down-projection + KV cache ##

        q_lora = self.Q_proj_down(x)
        kv_lora = self.KV_proj_down(x)
        kv_lora, kv_len = kv_cache(kv_lora)
        query_pos = kv_len - 1

        ## Step 2: Up-project and prepare NoPE + RoPE ##

        # Handle queries Q first
        q_nope_and_rope = self.Q_proj_up(q_lora).view(
            batch_size, seq_len, self.n_heads, self.nope_head_dim + self.rope_head_dim)
        q_nope, q_rope = torch.split(q_nope_and_rope, [self.nope_head_dim, self.rope_head_dim], dim=-1)

        # Handle keys and values K/V. V does not need RoPE
        kv_nope, k_rope = torch.split(kv_lora, [self.kv_lora_rank, self.rope_head_dim], dim=-1)
        kv_nope = self.KV_proj_up(kv_nope).view(
            batch_size, kv_len, self.n_heads, self.nope_head_dim + self.v_head_dim)
        k_nope, v = torch.split(kv_nope, [self.nope_head_dim, self.v_head_dim], dim=-1)

        ## Step 3: Handle RoPE Stream ##

        # Compute RoPE for queries and combine with no-RoPE part
        q_rope = q_rope.permute(0, 2, 1, 3) # bs x n_heads x seq_len x rope_head_dim
        q_rope = self.q_rope(q_rope, start_pos=query_pos)

        q_nope = q_nope.permute(0, 2, 1, 3) # bs x n_heads x seq_len x rope_head_dim
        q = torch.concat([q_nope, q_rope], dim=-1)

        # Compute RoPE for keys and combine with no-RoPE part
        k_rope = k_rope[:, None, :, :]
        k_rope = self.k_rope(k_rope).expand(-1,self.n_heads,-1,-1)
        k_nope = k_nope.permute(0, 2, 1, 3) # bs x kv_len x n_heads x rope_head_dim
        k = torch.concat([k_nope, k_rope], dim=-1)

        ## Step 4: Compute Multi-head Attention ##

        v = v.permute(0, 2, 1, 3) # bs x n_heads x kv_len x v_head_dim
        scores = torch.matmul(q, k.transpose(-1, -2)) / math.sqrt(self.rope_head_dim + self.nope_head_dim)
        attn = F.softmax(scores, dim=-1).to(torch.bfloat16)
        y = torch.matmul(attn, v).view(batch_size, 1, -1)
        y = self.wo(y)
```

```
            return y, kv_cache.get_data()
```

Your function should be defined as 'custom_kernel' (skeleton provided below)

```python
### DO NOT CHANGE THIS IMPORT STATEMENTS BLOCK ###
import os
import math
from typing import Tuple
import torch
import torch.nn.functional as F
import triton
from reference import KVCache, Config  # Definition of KVCache and Config classes are shown above. Must
import this way. Do not rewrite yourself.
### END OF IMPORT STATEMENTS BLOCK ###

### Import other packages here if needed

def custom_kernel(data: Tuple[Config, torch.Tensor, KVCache]) -> Tuple[torch.Tensor, KVCache]:
    """
    Optimized Triton-based forward pass for Multi-Head Latent Attention (MLA) decode.

    This function performs:
        1) Q/KV down-projections
        2) KV-cache update
        3) Q/KV up-projections
        4) RoPE application
        5) Multi-head attention (softmax, aggregation)
        6) Final output linear

    Args:
        data: Tuple of (config, x, kv_cache)
            - config: Config object (batch_size, dim, n_heads, lora_ranks, etc.)
            - x: input tensor (bs, 1, dim) of bfloat16
            - kv_cache: KVCache holding (bs, max_seq_len, dkv+d_rope)

    Returns:
        Tuple of (output, kv_cache.data)
        - output: attention output tensor (bs, 1, dim), bfloat16
        - kv_cache.data: updated KV-cache tensor (bs, max_seq_len, dkv+d_rope), bfloat16
    """
    config, x, kv_cache = data

    # ------------------------------------------------------------------
    # Step 1: Extract config parameters
    # ------------------------------------------------------------------
    bs = config.batch_size
    dim = config.dim
    nh = config.n_heads
    dq = config.q_lora_rank
    dkv = config.kv_lora_rank
    d_nope = config.qk_nope_head_dim
    d_rope = config.qk_rope_head_dim
    dv = config.v_head_dim
    msl = config.max_seq_len

    # Weight matrices
    wDQ = config.Q_proj_down_weight        # (dq, dim)
    wDKV = config.KV_proj_down_weight      # (dkv+d_rope, dim)
    wUQ = config.Q_proj_up_weight          # ((d_nope+d_rope)*nh, dq)
    wUKV = config.KV_proj_up_weight        # ((d_nope+dv)*nh, dkv)
    wO = config.wo_weight                  # (dim, nh*dv)

    # ------------------------------------------------------------------
    # Step 2: Down-projections (bs, 1, dim) -> (bs, dq) or (bs, dkv+d_rope)
    # ------------------------------------------------------------------
    q_lora = F.linear(x.squeeze(1), wDQ)       # (bs, dq)
    kv_in = F.linear(x.squeeze(1), wDKV)       # (bs, dkv+d_rope)

    # ------------------------------------------------------------------
    # Step 3: Update KV-cache & retrieve full cached sequence
    # ------------------------------------------------------------------
    kv_lora, kv_len = kv_cache(kv_in.unsqueeze(1))  # (bs, kv_len, dkv+d_rope), int
    query_pos = kv_len - 1
```

```
# -------------------------------------------------------------------
# Step 4: Up-projections
# -------------------------------------------------------------------
# Q: (bs, dq) -> (bs, (d_nope+d_rope)*nh) -> (bs, nh, d_nope+d_rope)
q_nope_rope = F.linear(q_lora, wUQ).view(bs, nh, d_nope + d_rope)
q_nope = q_nope_rope[..., :d_nope]      # (bs, nh, d_nope)
q_rope = q_nope_rope[..., d_nope:]      # (bs, nh, d_rope)

# KV: split the latent vector
kv_nope_input = kv_lora[..., :dkv]      # (bs, kv_len, dkv)
k_rope_input  = kv_lora[..., dkv:]      # (bs, kv_len, d_rope)

# -------------------------------------------------------------------
# Step 5: RoPE - use cached cosine / sine tables
# -------------------------------------------------------------------
cos_table, sin_table = _get_rope_tables(d_rope, msl, x.device)

# query side (single position)
cos_q = cos_table[query_pos].view(d_rope).contiguous()  # (d_rope,)
sin_q = sin_table[query_pos].view(d_rope).contiguous()  # (d_rope,)
rope_inplace_query(q_rope, cos_q, sin_q)

# key side (all cached positions)
cos_k = cos_table[:kv_len]                      # (kv_len, d_rope)
sin_k = sin_table[:kv_len]                      # (kv_len, d_rope)
k_rope = k_rope_input * cos_k + _rotate_half(k_rope_input) * sin_k   # (bs, kv_len, d_rope)

# -------------------------------------------------------------------
# Step 6: Latent projection for the "no-PE" query part
# -------------------------------------------------------------------
# wUKV shape: ((d_nope+dv)*nh, dkv) -> view as (nh, d_nope+dv, dkv)
wUKV_view = wUKV.view(nh, d_nope + dv, dkv)         # (nh, d_nope+dv, dkv)
wK = wUKV_view[:, :d_nope, :]                       # (nh, d_nope, dkv)
# q_nope: (bs, nh, d_nope)  wK: (nh, d_nope, dkv) -> (bs, nh, dkv)
q_nope_latent = torch.einsum('bhd,hdk->bhk', q_nope, wK)   # (bs, nh, dkv)

# -------------------------------------------------------------------
# Step 7: Compute attention scores (latent + RoPE)
# -------------------------------------------------------------------
# latent part: q_nope_latent @ kv_nope_input^T
kv_nope_T = kv_nope_input.transpose(1, 2)       # (bs, dkv, kv_len)
scores_nope = torch.matmul(q_nope_latent, kv_nope_T) # (bs, nh, kv_len)

# RoPE part: q_rope @ k_rope^T
scores_rope = torch.matmul(q_rope, k_rope.transpose(-2, -1))  # (bs, nh, kv_len)

scale = 1.0 / math.sqrt(d_nope + d_rope)
scores = (scores_nope + scores_rope) * scale        # (bs, nh, kv_len)

# -------------------------------------------------------------------
# Step 8: Softmax (Triton) -> attention weights
# -------------------------------------------------------------------
scores_flat = scores.reshape(bs * nh, kv_len)       # (B*H, kv_len)
attn_flat = _triton_softmax(scores_flat)            # (B*H, kv_len) bf16
attn = attn_flat.view(bs, nh, kv_len)               # (bs, nh, kv_len)

# -------------------------------------------------------------------
# Step 9: Weighted sum of latent keys (M)
# -------------------------------------------------------------------
M = torch.matmul(attn, kv_nope_input)               # (bs, nh, dkv)

# -------------------------------------------------------------------
# Step 10: Project aggregated latent keys to per-head values
# -------------------------------------------------------------------
wV = wUKV_view[:, d_nope:, :]                        # (nh, dv, dkv)
wV_T = wV.permute(0, 2, 1)                           # (nh, dkv, dv)
y_head = torch.einsum('bhd,hdk->bhk', M, wV_T)       # (bs, nh, dv)

# -------------------------------------------------------------------
# Step 11: Merge heads & final linear projection
# -------------------------------------------------------------------
y = y_head.reshape(bs, nh * dv)                      # (bs, nh*dv)
y = y.unsqueeze(1)                                   # (bs, 1, nh*dv)
output = F.linear(y, wO)                             # (bs, 1, dim)

# -------------------------------------------------------------------
```

```
      # Return the output and the updated KV-cache tensor
      # -------------------------------------------------------------------
      return output, kv_cache.data

```
```

Current runtime (lower is better): 3846.0450 microseconds
Target: 1700 microseconds. Current gap: 2146.0450 microseconds.

Rules:
- The tensors arguments passed in will be already on your cuda device.
- The weights for all parameters in the MLA will be given as input.
- All weights and data will be in `torch.bfloat16` format.
- Define all of your code in one final ```python ``` block.
- The entrypoint to your code must be named 'custom_kernel'.
- You will be using trition 3.4.0 and your kernels will be run on an Nvidia H200 GPU.
- Consider optimizing multiple operations with triton, not just limited to softmax. E.g., rope, attention, etc.
- You are allowed to use torch.compile().

Important rules in triton 3.4.0:
- `tl.load` does not have an argument called `dtype`. Never use it like `tl.load(..., dtype=...)`.
- Triton dtypes are not callable, so never use them like `tl.float16(1.0)`, `tl.float32(0.0)`.
- `tl.arange(start, end)`:
    - range length (end - start) must be power-of-2
    - start, end must be of type `tl.constexpr`
- `tl.range(start, end, step, num_stages)`:
    - keep loop index type stable, don't reassign it
    - start, end, step do not have to be `tl.constexpr` but must stay scalar integer types
    - num_stages must be `tl.constexpr`
- Do not something like x[0] or offs[0] inside a Triton kernel. Triton tensors are SIMD vectors; scalar indexing like [0] is not generally supported.

Here's an simple example correctly following these rules:

```python
import torch
import triton
import triton.language as tl

@triton.jit
def kernel_right(
    x_ptr, y_ptr, out_ptr,
    n_elements: tl.constexpr,
    BLOCK: tl.constexpr,              # constexpr; also power-of-2 for tl.arange
    ROW_STEP: tl.constexpr,
    NUM_STAGES: tl.constexpr,         # constexpr; used by tl.range(num_stages=...)
):
    pid = tl.program_id(axis=0)

    # ----------------------------------------------------------------
    # arange: constexpr args + power-of-2 range
    # ----------------------------------------------------------------
    offs = pid * BLOCK + tl.arange(0, BLOCK)    # (0, BLOCK) are constexpr
    mask = offs < n_elements

    x = tl.load(x_ptr + offs, mask=mask, other=0.0)
    y = tl.load(y_ptr + offs, mask=mask, other=0.0)

    # ----------------------------------------------------------------
    # Dtypes not callable: typed constants and casting
    # ----------------------------------------------------------------
    one_f32 = tl.full([], 1.0, tl.float32)              # typed scalar
    acc = tl.zeros((BLOCK,), dtype=tl.float32)          # typed vector
    acc = tl.cast(x, tl.float32) + tl.cast(y, tl.float32) + one_f32

    # ----------------------------------------------------------------
    # Avoid x[0]: scalar address load + broadcast
    # ----------------------------------------------------------------
    base = tl.full([], pid * BLOCK, tl.int32)
    x0 = tl.load(x_ptr + base, mask=(base < n_elements), other=0.0)
    x0_vec = tl.full((BLOCK,), x0, tl.float32)

    out_vec = acc + x0_vec

    # ----------------------------------------------------------------
```

```
# tl.range: keep loop index type stable, don't reassign it
#
# WRONG (causes "Loop-carried variable ... type stays consistent" assertion):
#   for row in tl.range(row, n_rows, row_step):
#       row = tl.load(...)  # row (int32) reassigned to tensor/bf16/...
#
# RIGHT:
#   - use a fresh name for loop index (e.g., r)
#   - compute offsets/tensors into *different* vars
#   - keep r as an integer index (int32) throughout
# ------------------------------------------------------------
# We'll do a tiny staged reduction over "rows" just as a demo.
n_rows = tl.full([], 4, tl.int32)  # small fixed count for demo (scalar int32)

extra = tl.zeros((BLOCK,), dtype=tl.float32)
for r in tl.range(0, n_rows, ROW_STEP, num_stages=NUM_STAGES):
    # r is an int32 loop index. Keep it that way.

    # Use r to build an integer shift; keep shifts as ints too.
    shift = r * tl.full([], 1, tl.int32)

    # Compute new offsets (int) without mutating r:
    offs_r = offs + shift

    # Load something; store into a separate var (tensor), not r:
    xr = tl.load(x_ptr + offs_r, mask=(offs_r < n_elements), other=0.0)
    extra += tl.cast(xr, tl.float32)

out_vec = out_vec + extra

tl.store(out_ptr + offs, tl.cast(out_vec, tl.float16), mask=mask)
```

Prompt used for the AHC039

You are a world−class algorithm engineer, and you are very good at programming.
Now, you are participating in a programming contest. You are asked to solve a heuristic problem, known as an NP−hard problem. Here is the problem statement:

Story
--------
Takahashi is a skilled purse seine fisher.
His fishing boat is equipped with state−of−the−art sonar, allowing him to accurately determine the positions of fish within the fishing area.
Additionally, the boat is capable of high−speed movement, enabling him to assume that fish remain stationary while he sets up the fishing net.

The fishing method involves using the boat to deploy nets and form a closed polygon, capturing the fish within the enclosed area.
To optimize efficiency, each edge of the polygon formed by the nets must be aligned either parallel to the east−west or north−south direction.
Furthermore, due to the limited length of the nets equipped on the boat, the polygon must be constructed within these constraints.

The fishing area contains two types of fish: mackerels and sardines.
For resource conservation reasons, sardines are currently prohibited from being caught in this fishing area.
Any sardines caught in the net must be released back into the sea.
Because this process is labor−intensive, Takahashi should focus on maximizing the catch of mackerel while avoiding sardines as much as possible.

Problem Statement
--------
There are $N$ mackerels and $N$ sardines on a two−dimensional plane.
Construct a polygon that satisfies the following conditions and maximize the value obtained by subtracting the total number of sardines inside the polygon from the total number of mackerels inside it.
Note that any points lying on the edges of the polygon are considered to be inside the polygon.

### Conditions
1. The number of vertices in the polygon must not exceed $1000$, and the total length of its edges must not exceed $4 \times 10^5$.
2. The coordinates of each vertex $(x, y)$ must be integers satisfying $0 \leq x, y \leq 10^5$.
3. Each edge of the polygon must be parallel to either the $x$−axis or the $y$−axis.
4. The polygon must not self−intersect: non−adjacent edges must not share any points, and adjacent edges must only meet at their endpoints.

Scoring
--------
Let $a$ be the total number of mackerels inside the polygon and $b$ be the total number of sardines inside the polygon.
Then, you will obtain the score of $\max(0, a - b + 1)$.

There are $150$ test cases, and the score of a submission is the total score for each test case.
If your submission produces an illegal output or exceeds the time limit for some test cases, the submission itself will be judged as WA or TLE, and the score of the submission will be zero.
The highest score obtained during the contest will determine the final ranking, and there will be no system test after the contest.
If more than one participant gets the same score, they will be ranked in the same place regardless of submission time.

Input
--------
Input is given from Standard Input in the following format:
~~~
$N$
$x_0$ $y_0$
$dots$
$x_{2N−1}$ $y_{2N−1}$
~~~

− In all test cases, the number of mackerels and sardines, $N$, is fixed at $5000$.
− For each $i = 0, 1, \dots, N−1$, $(x_i, y_i)$ represents the coordinates of the $i$−th mackerel.
− For each $i = 0, 1, \dots, N−1$, $(x_{N+i}, y_{N+i})$ represents the coordinates of the $i$−th sardine.
− Each coordinate $(x_i, y_i)$ satisfies $0 \leq x_i, y_i \leq 10^5$, and all coordinates are distinct.

Output
--------
Let the number of vertices in the polygon be $m$ ($4 \leq m \leq 1000$), and let $(a_i, b_i)$ denote the coordinates of the $i$-th vertex.
Then, output to Standard Output in the following format:
~~~
$m$
$a_0$ $b_0$
$dots$
$a_{m-1}$ $b_{m-1}$
~~~

The output vertices do not necessarily need to form the actual corners of the polygon.
In other words, three consecutive vertices $(a_i, b_i), (a_{i+1}, b_{i+1}), (a_{i+2}, b_{i+2})$ may lie on a straight line.
However, all vertices must have distinct coordinates.

The vertices can be output in either clockwise or counterclockwise order.

Your program may output multiple solutions.
If multiple solutions are output, only the last one is used for scoring.

Here is the last code we ran:
```cpp
{CODE HERE}
```

Current performance (higher is better): 3668.8333
Target: 5000. Current gap: 1331.1667

Rules:
- You must use cpp20 to solve the problem.
- Define all of your code in one final ```cpp ``` block.
- In your final response, you should only output the code of your program. Do not include any other text.

Try diverse approaches to solve the problem. Think outside the box.

## Prompt used for the AHC058

You are a world-class algorithm engineer, and you are very good at programming.
Now, you are participating in a programming contest. You are asked to solve a heuristic problem, known as an NP-hard problem. You are trying to get the highest score possible to get the best rank on the leaderboard. Here is the problem statement:

# Story

APPLE ARTIS Corporation (commonly known as AA Corporation) is a company engaged in the mass production of apples. Recently, after many years of research, they have successfully developed an innovative machine capable of generating apples from nothing.

However, to begin full-scale mass production of apples using this machine, it is necessary to mass-produce the machines themselves. To achieve this, AA Corporation has established a hierarchical system in which machines are created to produce apple-generating machines, and machines are created to produce those machine-producing machines, and so on.

As an engineer at AA Corporation, you have been tasked with developing a production planning algorithm that utilizes this hierarchy of machines to produce as many apples as possible.

# Problem Statement

There are $N \times L$ types of machines, composed of $N$ types of IDs and $L$ types of Levels. A machine with Level $i$ and ID $j$ is referred to as **machine $j^i$** ($0 \leq i < L,\ 0 \leq j < N$).

The production capacity of machine $j^0$ is $A_j$. The initial cost of machine $j^i$ is $C_{i,j}$.

Your objective is to maximize the total number of apples at the end of $T$ turns, following the procedure of the production plan below.

## Procedure of the Production Plan

Let $B_{i,j}$ be the number of machines $j^i$, and initially all $B_{i,j}$ are set to 1.
Also, let $P_{i,j}$ be the power of machine $j^i$, and initially all $P_{i,j}$ are set to 0.

The initial number of apples at the start of the plan is $K$.
Each turn proceeds according to the following steps:

1. You choose one of the following two actions:
    – Strengthen machine $j^i$: Consume $C_{i,j} \times (P_{i,j} + 1)$ apples to increase $P_{i,j}$ by
1. However, you cannot strengthen if it would result in a negative number of apples.
    – Do nothing.
2. For all machines $j^i$, perform the following in the order of Level 0, 1, 2, 3:
    – For Level 0 machines ($i = 0$):
        – Increase the number of apples by $A_j \times B_{i,j} \times P_{i,j}$.
    – For machines of Level 1 or higher ($i \geq 1$):
        – Increase $B_{i-1,j}$ by $B_{i,j} \times P_{i,j}$.

Choose your actions wisely to maximize the number of apples at the end of $T$ turns.

# Scoring

Let $S$ be the number of apples at the end of $T$ turns. Your score is calculated as $\mathrm{round}(10^5 \times \log_2 S)$.
The higher the score, the better.

The following cases will result in a WA:

– Performing a strengthening action that results in the number of apples becoming less than $0$
– Specifying a non-existent machine Level or ID
– Taking fewer than $T$ actions

There are $150$ test cases, and the score of a submission is the total score for each test case.
If your submission produces an illegal output or exceeds the time limit for some test cases, the submission itself will be judged as WA or TLE, and the score of the submission will be zero.
The highest score obtained during the contest will determine the final ranking, and there will be no system test after the contest.

---

# Input

Input is given from Standard Input in the following format.

```
N L T K
```

```
A_0 A_1 \cdots A_{N-1}
C_{0,0} C_{0,1} \cdots C_{0,N-1}
C_{1,0} C_{1,1} \cdots C_{1,N-1}
dots
C_{L-1,0} C_{L-1,1} \cdots C_{L-1,N-1}
```

- The first line contains four integers $(N, L, T, K)$:
  - $(N)$ is the number of machine IDs, and $(N = 10)$.
  - $(L)$ is the number of machine Levels, and $(L = 4)$.
  - $(T)$ is the total number of turns, and $(T = 500)$.
  - $(K)$ is the number of apples at the start of the plan, and $(K = 1)$.
- The second line contains $(N)$ space-separated integers $(A_0, A_1, \dots, A_{N-1})$ representing the production capacities of Level 0 machines:
  - $(A_j)$ is the production capacity of machine $(j^0)$, satisfying $(1 \leq A_j \leq 100)$.
  - $(A)$ is sorted in ascending order $(A_0 \leq A_1 \leq \cdots \leq A_{N-1})$.
- The following $(L)$ lines each contain $(N)$ space-separated integers $(C_{i,j})$:
  - $(C_{i,j})$ is the initial cost of machine $(j^i)$, satisfying $(1 \leq C_{i,j} \leq 1.25 \quad imes 10^{12})$.

# Output

Output exactly $(T)$ lines.
Each line should describe the action taken on turn $(t)$ $((0 \leq t < T))$, in order from turn 0, using the following format:

- To strengthen machine $(j^i)$:

```
i j
```

- To do nothing:

```
-1
```

Your program may include comment lines in the output that start with `#`.

# Input Generation

The function $(\mathrm{rand\_double}(L, U))$ represents generating a real number uniformly at random between $(L)$ and $(U)$.

## Generation of $(A_j)$

- When $(j = 0)$: set $(A_0 = 1)$
- When $(j
eq 0)$: set $(A_j = \mathrm{round}(10^{\mathrm{rand\_double}(0,2)}))$
- After generating all values, sort the array $(A)$ in ascending order

## Generation of $(C_{i,j})$

- When $(i = 0)$ and $(j = 0)$: set $(C_{0,0} = 1)$
- Otherwise: set $(C_{i,j} = \mathrm{round}(A_j \quad imes 500^i \quad imes 10^{\mathrm{rand\_double}(0,2)}))$

Here is the last code we ran:
```cpp
{CODE HERE}
```

Current performance (higher is better): 5626752.9267
Target: 6500000. Current gap: 873247.0733

Rules:
- You must use cpp20 to solve the problem.
- Define all of your code in one final ```cpp ``` block.
- In your final response, you should only output the code of your program. Do not include any other text.

Try diverse approaches to solve the problem. The best solution will make efficient use of the entire 2 second time limit without exceeding it. Think outside the box.

Prompt used for Denoising

```
You are an expert in computational biology and single-cell RNA-seq analysis.
Your task is to develop a denoising algorithm for scRNA-seq count data. You are experienced in
compuational biology libraries and tools and are familiar with problems in denoising in the single-cell
field.

## Problem

Single-cell RNA-seq data is noisy due to technical dropout and low capture efficiency.
Given noisy count data, predict the true expression levels.

Your prediction is evaluated against held-out molecules using two metrics:
1. **MSE** - Mean Squared Error in log-normalized space
2. **Poisson Loss** - Poisson negative log-likelihood

You need to implement a novel denoising algorithm that outperforms the current state-of-the-art without
overfitting.

## Data Format

- Input `X`: numpy array of shape (n_cells, n_genes) - **raw count data**
- Output: numpy array of same shape - your denoised counts

## Evaluation

Your output is evaluated using these exact functions:

```python
<<<EVALUATE_MSE_FUNC>>>
```

```python
<<<EVALUATE_POISSON_FUNC>>>
```

## Scoring

**Poisson is a HARD CONSTRAINT.** Your solution is REJECTED if `poisson_norm < 0.97`.
- `poisson_norm = (0.257575 - poisson) / (0.257575 - 0.031739)`
- MAGIC baseline achieves 0.97

**Reward = MSE score only** (after passing Poisson constraint).

## Budget & Resources

- **Time budget**: <<<BUDGET_S>>>s for your code to run. You should time your code and make sure it runs
within the time budget.
- **CPUs**: <<<CPUS>>> available

## Function Signature to return

```python
def magic_denoise(X, **kwargs):
    # kwargs may include: budget_s, random_state, knn, t, n_pca, solver, decay, knn_max, n_jobs
    # You can add your own parameters too
    # Your implementation
    return denoised_X  # same shape as X
```

## Rules

- Implement `magic_denoise(X, ...)` that returns denoised data
- Use numpy, scipy, sklearn, graphtools, scprep, scanpy
- Make all helper functions top level, no closures or lambdas
- No filesystem or network IO

## Key Insights from Benchmarks

- NORMALIZATION ORDER MATTERS: Denoise raw/log counts first, then normalize. "Reversed normalization order"
  achieves Poisson ~0.98 vs ~0.55 for standard order.
- Square root transform is variance-stabilizing for Poisson distributions
- Poisson loss is highly affected by low non-zero values - push values < 1 toward zero
- The original MAGIC with reversed normalization achieves best results
```

