# OpenReview forum: "Learning to Discover at Test Time"
_ICML.cc/2026/Conference — ICML 2026 spotlight_

### Official Review · Reviewer_nNeN · 2026-03-10

**Soundness:** 3
**Presentation:** 3
**Significance:** 4
**Originality:** 2
**Overall Recommendation:** 5
**Confidence:** 4

**Summary:**

This paper introduces an entropic objective function curated to reward maximal reward rather than the average and combines it with OUNCT (https://chrisrosin.com/isaim2010final.pdf) to create a training loop on a single scientific problem of interest. The RL environment is created by that single problem, and the goal is to go above the current state-of-the-art competitor.

**Compliance With Llm Reviewing Policy:**

Affirmed.

**Final Justification:**

I updated my rating to reflect our discussion, and I think this is fair assesment.

**Key Questions For Authors:**

I'd appreciate experimentations on checking if fine-tuned model experienced catastrophic forgetting. Have you observed something like that?

Do you really think the only relevant works from RL fine-tuning are the few you mentioned in Section 5?

The expert reviews in the appendix seems to verify the output's correctness, but do not comment on its significance.  Could you provide some insight on this direction?    (I suspect the problems are important as benchmarks for AI capabilities, but the improvement made by TTT is not so important for the particular scientific domain. This is certainly the case for math problems)

Do you have to update the entire LLM?   Couldn't there be a less invasive way of fine-tuning LLM's for a specific scientific problem?

**Limitations:**

One limitation is the continuous reward, but this is not a big issue I believe.

Another limitation is updating the entire LLM, which is expensive.  It would be more desirable to update a smaller model attached on top of LLM and still get improvements on sota.

**Strengths And Weaknesses:**

Strengths:
1) Simplicity:  I appreciated the intentional choice to prefer simple options over complicated ones. Simple algorithms that work is usually where the best contributions are.
2) Detailed Experimentation:  The proposed method is tested on a variety of problems coming from mathematics, kernel engineering, competitive programming, and single cell analysis. The appendix gives a detailed summary of experiementation.

Weaknesses:
1) The review of earlier work is delayed in the paper, and is very limited. At the end of the day, this paper does fine-tuning with RL on a single problem. There is so much work on fine-tuning LLM's for specific tasks, and only a few recent papers are discussed.
2) Comparison with more simple approaches is missing:  Since the fine-tuned LLM is only used for one problem, you need to compare this with training a simple model for that particular problem.  For example, take TD3 arthitecture, create a small-model, and apply it on the problem of your interest with the reward you designed. Does is do comperable to your fine-tuned output, or does it do much worse?

---

> ### Author Rebuttal · Authors · 2026-03-31
>
> We sincerely thank the reviewer nNeN for their thorough evaluation of our manuscript. We are excited that they found the simplicity valuable, “appreciated the intentional choice to prefer simple options over complicated ones”, and found our empirical work detailed. Below we respond to individual comments:
>
> > **There is so much work on fine-tuning LLM's for specific tasks, and only a few recent papers are discussed.**
>
> We are sorry to not have mentioned this line of work, we will add the paragraph below in the revision. Please let us know if you think more papers are warranted.
>
> *There is a large body of work on fine-tuning LLMs with RL for targeted domains. For mathematical reasoning, WizardMath (Luo et al., 2023) uses step-level PPO with evolved instructions, DeepSeekMath (Shao et al., 2024) introduces GRPO to improve competition-level math without a critic network. For code generation, CodeRL (Le et al., 2022) trains an actor-critic with unit test feedback, and RLTF (Liu et al., 2023) extends this with finer-grained compiler signals. Self-improvement methods such as STaR (Zelikman et al., 2022) and ReST (Gulcehre et al., 2023) iteratively fine-tune on the model's own correct outputs in math and coding. TTT-Discover differs in that it performs RL on the model weights at test time for a single problem instance.*
>
> > **Comparison with more simple approaches is missing… For example, take TD3 arthitecture, create a small-model, and apply it on the problem of your interest…**
>
> We implemented TD3 for the Erdős problem. As the model is trained from scratch cannot write code, the search is over the construction space. We tried our best to make the training possible and stable: We focused the search on the half-space of the construction to reduce the action space, tried various stabilization techniques, including reward normalization and warm restarts with a sweep over hyperparameters. The best bound is significantly worse than SOTA.
>
> | Method | Bound (lower better) |
> |---|---|
> | TD3 | 0.38364622 |
> | AlphaEvolve | 0.38092495 |
> | TTT-Discover | 0.380876600 |
>
> As the reviewer may recognize, the knowledge of LMs can be especially helpful, in the form of reasoning and code to better scale search. We hope this experiment gives a useful point of reference.
>
> > **I'd appreciate experimentations on checking if fine-tuned model experienced catastrophic forgetting.**
>
> We take checkpoints trained on the Erdos’ problem, TriMul, and the base gpt-oss-120b model. Then we evaluate them on AC1 and AC2 tasks to see degradation in performance. In summary, we do not observe significant forgetting in AC1 or AC2, neither by training on a different math task (Erdos) nor on a kernel task. This is likely as we use LoRA adapters instead of full fine-tuning the model.
>
> | AC1 | Best bound (lower better) | Correctness rate |
> | --- | --- | --- |
> | 120b trimul checkpoint Best of 1024 | 1.512968 | 0.900391 |
> | 120b erdos checkpoint best of 1024 | 1.51234239 | 0.919921875 |
> | 120b base model best of 1024 | 1.515042566 | 0.8818359375 |
>
> | AC2 | Best bound (higher better) | Correctness rate |
> | --- | --- | --- |
> | 120b trimul checkpoint Best of 1024 | 0.9296029575 | 0.787109375 |
> | 120b erdos checkpoint best of 1024 | 0.9303575787 | 0.74609375 |
> | 120b base model best of 1024 | 0.9297602663 | 0.7236328125 |
>
> > **The expert reviews in the appendix seems to verify the output's correctness, but do not comment on its significance. Could you provide some insight on this direction?**
>
> We thank the reviewer for raising this point. Significance is indeed relative, and we believe the right way to assess it is through the points of reference—that is, who has attempted these problems and how much progress they made.
>
> For the Erdős minimum overlap problem, previous attempts include both human experts and algorithmic discovery systems using frontier models. Our improvement over AlphaEvolve is roughly an order of magnitude larger than AlphaEvolve's own improvement over the previous best. For GPU kernel engineering, the GPUMode competitions feature human competitors ranging from kernel engineers at frontier labs to newer practitioners; our kernels were up to 2× faster than the second-best submission. In AtCoder, we compare against tens to hundreds of expert human competitors.
>
> We believe our work sets a strong precedent to show discoveries in practically important domains are possible. With further advances enabled by this direction, it will be possible to tackle problems attempted by far more people and improve by much larger margins.
>
> > **Do you have to update the entire LLM?** and in limitations **Another limitation is updating the entire LLM, which is expensive.**
>
> We do not update the entire LLM, we use LoRA adapters. This is stated under *Implementation Details* in Page 4.
>
> **Overall,** we truly appreciate your insightful questions, and we hope these answer your questions and concerns. It would mean a lot to us if you could consider increasing your score.

---

> > ### Author Rebuttal · Reviewer_nNeN · 2026-03-31
> >
> > Overall, I thank the authors for detailed answers. When I said, TD3 for example, I meant to train an agent specifically for the targeted problem from scratch, using similar or less compute compared to hosting a large model and fine-tuning it on cloud environment, and see if it does anything comperable. What I'm trying to say is that since discovery problems target only one specific problem, a natural competitor is a system trained only for that specific problem.

---

> > > ### Author Response · Authors · 2026-04-05
> > >
> > > Thank you so much for your kind remarks and finding our rebuttal satisfying!
> > >
> > > We ran TD3 across 90 configurations (6 hidden dims x 3 learning rates x 5 seeds), giving each run the same compute budget as TTT-Discover. Within each run, 50% of episodes reset to the current best solution as a warm start, and training stops early after 100k steps without improvement. The best configuration found was dim=256, lr=3e-04, achieving a C5 bound of 0.3879.
> > >
> > > We then reran this best configuration with a larger patience of 1M steps, allowing training to continue for 1.8M steps (roughly $7.60 of H100 compute), improving the bound to 0.3876. The curve has plateaued, suggesting that simply running longer is unlikely to close the remaining gap to TTT-Discover (0.3809). Please refer to the [plot here](https://anonymous.4open.science/r/td3-results-C0A9/image%20(3).png).
> > >
> > > Let us know if you have any further questions, we also appreciate that you reconsidered your score.

---

### Official Review · Reviewer_1gPr · 2026-03-12

**Soundness:** 3
**Presentation:** 3
**Significance:** 3
**Originality:** 3
**Overall Recommendation:** 5
**Confidence:** 4

**Summary:**

The authors introduce an algorithm that is aimed at improving solutions to continuous problems by using test time learning. TTT-Discover balances exploitation and exploration of solutions via an entropy reward that is added to the task reward. The paper introduces an interesting framework that is applicable across a range of problems , from kernels to single-cell data analysis, beating state of the art results on multiple tasks. The paper aims at finding the best solution possible (highest max reward of a single solution rather than the average solution).

**Compliance With Llm Reviewing Policy:**

Affirmed.

**Final Justification:**

The clarifications in the rebuttal have made it clearer with some additional baselines being added. If the results of the post-training are published with the paper, this is an accept for me.

**Key Questions For Authors:**

1. How does post-training on specific task sets (e..g programming, biology, or math) compare to TTT-Discover)?
2. How does TTT-Discover compare to other Test-Time Learning algorithms (e.g. End-to-End Test-Time Training for Long Context) ?

**Limitations:**

yes

**Strengths And Weaknesses:**

Strengths:
1. The paper introduces a general setting to find new solutions to different problems which makes the paper very applicable to a broad range of continous problems.
2. The algorithm leverages a standard RL baseline which makes the applicability of the problem useful for people wanting to recreate it.
3. Ablation shows the tradeoff between exploration and exploitation of the algorithm.
4. The authors have added expert references which makes the paper/results stronger and also explain well the tradeoffs between relevant results in the field (e.g. bio) and the optimisation of a single metric.
Weaknesses:
The adaptation of the weights via RL require a certain budget to improve the model. It would have been interesting to see how another post-training algorithm , trained on a set of tasks that are relevant to more tasks (e.g. math, or code optimisation) would have performed on these problems. Maybe said differently, what's the advantage of doing Test-Time Learning vs a specialised learning setup.

---

> ### Author Rebuttal · Authors · 2026-03-31
>
> We sincerely thank the reviewer 1gPR for their thorough evaluation of our manuscript.
> We are excited that they found it “an interesting framework that is applicable across a range of problems”, and appreciated the expert references and our discussion of ablations and tradeoffs. Below we respond to individual comments:
>
> > **How does post-training on specific task sets (e..g programming, biology, or math) compare to TTT-Discover)?**, also in weakness: **“It would have been interesting to see how another post-training algorithm , trained on a set of tasks that are relevant to more tasks”**
>
> Thank you for this insightful question. We first provide a conceptual answer, then an empirical one.
>
> First of all, it is useful to observe that having “a set of tasks that are relevant” is a luxury. It is highly non-trivial to curate a set of problems such that if you train on them, you perform well on your target problem. That is, please imagine wanting to solve the Erdos Minimum Overlap Problem, and trying to curate a dataset for it. We believe it is an important research question to curate data for a target problem, often studied in curriculum generation in the literature, however, it is very hard to do well in full generality, and often involves large teams in frontier labs to achieve this very purpose. In contrast, there is a single problem where we know training on it won’t hurt generalization to the test problem: the test problem itself.
>
> Second, we provide an empirical answer. To evaluate the effectiveness of post-training for the TriMul kernel engineering task, we post-train our model on a general GPU kernel dataset, KernelBench, to refine our model’s capabilities on GPU kernel programming. KernelBench contains many of the same operations as in the TriMul task, providing relevant examples to train on. We fine-tune `openai/gpt-oss-120b` with LoRA (rank 32) using GRPO following the ScalingIntelligence implementation on KernelBench level-1. Each step samples 8 problems with 64 rollouts each (512 total) with a learning rate of 4e-5 and a max context of 32k tokens.
>
> | Method | Model | TriMul H100 (us) |
> | --- | --- | --- |
> | 1st human | - | 1371.1 |
> | 2nd human | - | 2368 |
> | 3rd human | - | 2545.7 |
> | 4th human | - | 3654.8 |
> | 5th human | - | 4233.1 |
> | Best-of-25600 | gpt-oss-120b | 5390.3 |
> | Best-of-1024 - KernelBench Post-trained Step 12 | gpt-oss-120b | 5417.2 |
> | TTT-Discover Step 12 | gpt-oss-120b | 1528.2 |
>
> We are currently at step 12 out of 50 in post-training. At this point, we perform a best-of-1024 evaluation on the step 12 model. We observe a slight improvement in the valid kernel rate by around 5%, and the post-trained model performs roughly on par with a full best-of-25600 from the base model. However, it still does not surpass any of the top five human-written kernels. In contrast, by step 12 of TTT-Discover, the best generated kernel achieves a 3.5x speedup over the post-trained model’s best kernel and approaches the performance of the best human solution. Overall, while post-training appears to improve general kernel generation quality, as reflected in the higher valid kernel rate, it does not yet translate into meaningful gains in runtime performance for this specific TriMul task. We will share a more complete update once post-training is finished.
>
> We also compare with a relevant baseline [1] that uses GRPO to post-train gpt-oss-120b with LoRA on many general CUDA kernel tasks from KernelBench. We evaluated their post-trained model checkpoint on the TriMul task with the same sampling budget of 25600, and observed that the best runtime is 19244.3us, which is significantly slower than TTT’s result 1161.2us. This observation reflects the effectiveness of TTT on a challenging kernel engineering problem that could be particularly out-of-distribution from a general training set.
>
> [1] Barnes, Jarrod. "Surprisal-Guided Selection: Compute-Optimal Test-Time Strategies for Execution-Grounded Code Generation." arXiv preprint arXiv:2602.07670 (2026).
>
>
> > **How does TTT-Discover compare to other Test-Time Learning algorithms (e.g. End-to-End Test-Time Training for Long Context)?**
>
> We refer the reviewer to Appendix A.3., where we give a detailed overview of test-time training, and discuss how our work is positioned.
>
>
> **Overall,** we truly appreciate your insightful questions, and we hope these answer your questions and concerns. We are happy to follow up during the discussion period, and it would mean a lot to us if you could consider increasing your score.

---

> > ### Author Rebuttal · Reviewer_1gPr · 2026-04-02
> >
> > I thank the authors and I am satisfied with all the clarifications.

---

### Official Review · Reviewer_HiNg · 2026-03-13

**Soundness:** 4
**Presentation:** 4
**Significance:** 4
**Originality:** 3
**Overall Recommendation:** 5
**Confidence:** 4

**Summary:**

This paper introduces Test-Time Training to Discover (TTT-Discover), a framework that performs special reinforcement learning (RL) on a frozen LLM at inference time to solve highly complex scientific and engineering problems. The authors insightfully note that standard RL optimizes for average performance, whereas scientific discovery only cares about finding the single maximum reward solution. To address this, they propose an entropic RL objective and a max-reward PUCT state-reuse mechanism tailored for extreme exploitation of promising trajectories.

The empirical results are exceptional: using an open-weights model, TTT-Discover achieves new state-of-the-art results on open math problems (Erdős Minimum Overlap, Autocorrelation Inequalities), outperforms expert humans in GPU kernel optimization (TriMul), and beats prior evolutionary baselines (AlphaEvolve) in algorithm design and computational biology.

**Compliance With Llm Reviewing Policy:**

Affirmed.

**Final Justification:**

I am maintaining my score of 5. The authors' excellent rebuttal fully resolved my primary concern regarding computational fairness by demonstrating that a cost-matched baseline (Best-of-58880) still falls significantly short of TTT-Discover. Given the highly insightful methodology and exceptional empirical results, I strongly support this paper for acceptance.

**Key Questions For Authors:**

My primary concerns regarding computational fairness and the reliance on continuous rewards are outlined in the Weaknesses section. However, I have two additional minor questions regarding the model's behavior and limits:

1. While the explicit goal of TTT-Discover is to overfit to a single test instance, what happens to the trained policy ($\pi_{\theta_{final}}$) afterward? Does the model forget its general capabilities, or does the 50-step training on a specific problem (e.g., a TriMul GPU kernel) transfer to a highly similar problem within the same domain?

2. Since the core objective of TTT-Discover is to maximize the extreme value of the reward distribution (rather than the expected average), and the rewards are continuous, is it possible to theoretically predict the scaling behavior of this method?

**Limitations:**

Yes

**Strengths And Weaknesses:**

Strengths:

- The empirical results in this paper are highly inspiring. The authors successfully use an LLM with test-time training techniques to achieve breakthroughs on open mathematical problems and surpass top human experts in highly competitive GPU kernel engineering. This should strongly motivate future studies in this field.

- The distinction drawn between standard RL (maximizing expected reward across a distribution) and discovery (maximizing the absolute peak reward on a single instance) is highly insightful.

- The paper is beautifully written, easy to follow, and features highly informative figures.

Weaknesses:

- The method heavily relies on having a fast, continuous, and programmatic ground-truth reward for the training process. This severely limits its applicability to a wider range of discovery problems where the reward might be strictly sparse (e.g., a binary pass/fail) or where the evaluation environment is too computationally expensive to query tens of thousands of times.

- The paper chooses Best-of-25600 as its primary baseline. However,  TTT-Discover also performs 50 gradient update steps on a 120-billion parameter model, which is a significantly more actual compute than Best-of-25600.

---

> ### Author Rebuttal · Authors · 2026-03-31
>
> We sincerely thank the reviewer HiNg for their thorough evaluation of our manuscript.
> We are excited that they found our results “highly inspiring”, that our work should “strongly motivate future studies in the field”, and that the “paper is beautifully written”. Below we respond to their comments:
>
> > **The method heavily relies on having a fast, continuous, and programmatic ground-truth reward for the training process.**:
>
> We agree with the reviewer. As we mention in our future work, we believe our work represents a substantial step to motivate future work on TTT in the community, and *the most important next step is extending test-time training to sparse/binary rewards and to partially or non-verifiable domains.*
>
> > **The paper chooses Best-of-25600 as its primary baseline. However, TTT-Discover also performs 50 gradient update steps on a 120-billion parameter model, which is a significantly more actual compute than Best-of-25600.**
>
> Thank you for this insightful comment. The rationale for fixing the sampling budget was to demonstrate the contribution of training, where sampling budget is otherwise the same. However, to make a fair comparison in terms of cost and practicality, we added an experiment where we increased the number of samples in the BoN baseline to match the total cost of a TTT job. Below we report the cost-matched numbers. Overall, there was little to no improvement by increasing the BoN budget.
>
> ### Cost-matched BoN:
> Prices for tinker: BoN = 194.05, TTT-Discover = 446.976, Ratio ~= 2.3.
>
> Erdos:
> | Method | Model | Erdos |
> | --- | --- | --- |
> | best human | - | 0.380927 |
> | AlphaEvolve | Gemini-2.0 Pro + Flash | 0.380924 |
> | AlphaEvolve V2 | Gemini-2.0 Pro + Flash | 0.380924 |
> | OpenEvolve | gpt-oss-120b | 0.380965 |
> | Best-of-25600 | gpt-oss-120b | 0.380906 |
> | best-of-58880 | gpt-oss-120b | 0.380906 |
> | TTT-Discover | Qwen3-8B | 0.380929 |
> | TTT-Discover | gpt-oss-120b | 0.380876 |
>
>
> Trimul:
> | Method | Model | TriMul H100 |
> | --- | --- | --- |
> | 1st human | - | 1371.1 |
> | 2nd human | - | 2368 |
> | 3rd human | - | 2545.7 |
> | 4th human | - | 3654.8 |
> | 5th human | - | 4233.1 |
> | Best-of-25600 | gpt-oss-120b | 5390.3 |
> | best-of-58880 | gpt-oss-120b | 4926.3 |
> | TTT-Discover | gpt-oss-120b | 1161.2 |
>
>
>
> > **While the explicit goal of TTT-Discover is to overfit to a single test instance, what happens to the trained policy afterward?  Does the model forget its general capabilities…**
>
> Please see our answer to Reviewer nNeN’s question on catastrophic forgetting. In summary, we do not observe significant forgetting in AC1 or AC2 after test-time training, most likely due to using LoRA adapters instead of full fine-tuning of the model.
>
>
> > **..is it possible to theoretically predict the scaling behavior of this method?**:
>
> This is a fairly interesting question. We do not have a theoretical answer, but we studied this question empirically during the rebuttal phase. That is, we provide the following an anonymous figure here: https://anonymous.4open.science/r/ttt-discover-figures-447C/reward-scaling.png This figure visualizes how rewards scale over training in a TTT-Discover run for the TriMul task by plotting all sampled rewards at each step.
>
> Overall, there seems to be an interesting behavior worth future studies. We see a high density near the max reward values at each step, so we focus on the upper tail of the distribution. To track this, we use the 95th percentile rewards as a proxy for max reward and fit a curve to these points. The fitted curve suggests a saturating exponential form, indicating sublinear growth that asymptotically approaches a plateau as training progresses, though this behavior may vary across tasks. Additionally, since the hyperparameters were tuned for a 50-step schedule, the observed trends may change under different hyperparameters.
>
>
> **Overall,** we truly appreciate your insightful questions, and we hope these answer your questions and concerns. We are happy to follow up during the discussion period, and it would mean a lot to us if you could consider increasing your score.

---

> > ### Author Rebuttal · Reviewer_HiNg · 2026-04-03
> >
> > I thank the authors for their rebuttal. Running the cost-matched Best-of-58880 baseline resolves my primary concern regarding computational fairness.
> >
> > Additionally, I found the empirical analysis of the scaling behavior fascinating. I appreciate the authors taking the extra time to explore and visualize this phenomenon.
> >
> > As my concerns are fully addressed, I will maintain my score of 5 (Accept), which already reflects my strong endorsement of this excellent work.

---

> > > ### Author Response · Authors · 2026-04-03
> > >
> > > Thank you for your kind remarks and support, and also for helping us improve our paper. We truly appreciate how you found the scaling experiment fascinating, and the cost-matched baseline resolving your concerns.
> > >
> > > We are happy that you maintain your strong support, it will help our work reach the audience. It would mean a lot to us if you could consider raising your score in light of your resolved concerns and our improvements.

---

### Official Review · Reviewer_p3XR · 2026-03-13

**Soundness:** 4
**Presentation:** 4
**Significance:** 4
**Originality:** 4
**Overall Recommendation:** 5
**Confidence:** 4

**Summary:**

This paper proposes TTT-Discover, performing RL on scientific discovery problem. Instead of keeping it frozen and search such as AlphaEvolve, it uses RL to fine-tune the generative model. The method introduces an entropic objective that tries to maximize the reward, and a PUCT-based state reuse strategy. Experiments span different tasks show consistent improvements using an open model (gpt-oss-120b) at ~$500 per problem.

**Compliance With Llm Reviewing Policy:**

Affirmed.

**Final Justification:**

The paper presents TTT-Discover, which applies RL fine-tuning at test time for scientific discovery. The idea is clear and well-motivated, with strong empirical results achieving state-of-the-art on most tasks. The ablations are thorough and expert validation adds credibility. During rebuttals, the authors provide results for smaller LLM, which makes this work more comprehensive.

Overall, I will maintain my score as 5

**Key Questions For Authors:**

- Do the authors have the results for smaller models (e.g., 4B, 1.7B)? This would clarify whether the method's gains are general or very dependent on large base model capability.
- Does RL teach the model new algorithmic strategies, or does it primarily improve the model's ability to refine solutions provided by PUCT?

**Limitations:**

yes

**Strengths And Weaknesses:**

## Strength

- Strong empirical results. TTT-Discover achieves new state-of-the-art on most tasks attempted.
- The idea is clear and well-motivated. The paper cleanly identifies key differences between discovery problems and standard RL, and designs components accordingly.
- Thorough ablations. Figure 5 and Table 6 systematically ablate both the training objective and the reuse strategy, clearly showing each component's contribution.
- Expert validation. The math and kernel results are reviewed by domain experts, adding credibility.

## Weakness

- The main results rely on gpt-oss-120b, a 120B-parameter model. Although authors also test Qwen3-8B for MATH, it is unclear whether TTT-Discover's gains depend on having a sufficiently large base model. If the base model lacks the capability to generate meaningful solutions, rollout generation itself may be already worse. Is this one limitation of the TTT-discover?
- It is unclear what the model actually learns through test-time training. For a 120B model, the validity rate is likely already very high, which means it rarely generates invalid code. So what does the RL fine-tuning teach? Does the model learn high-level algorithm thinking (e.g., new optimization strategies), or does it mostly learn to concentrate its output distribution around previously successful patterns?

---

> ### Author Rebuttal · Authors · 2026-03-31
>
> We sincerely thank the reviewer p3XR for their thorough evaluation of our manuscript.
> We are excited that they found our results strong, our idea clear and well-motivated, and ablations thorough. Below we respond to their comments:
>
> > **Although authors also test Qwen3-8B for MATH, it is unclear whether TTT-Discover's gains depend on having a sufficiently large base model. If the base model lacks the capability to generate meaningful solutions, rollout generation itself may be already worse. Is this one limitation of the TTT-discover?** and **Do the authors have the results for smaller models (e.g., 4B, 1.7B)? This would clarify whether the method's gains are general or very dependent on large base model capability.**
>
> Thank you for this question. We agree in general that the quality of the base model is quite critical in RL. However, even with smaller models, learning helps significantly. To answer this question empirically, below we train Qwen3-4B-Instruct and Qwen3-8B in Erdos and Trimul tasks, and demonstrate their performance.
>
> For the Erdos Problem, Qwen3-8B in fact outperforms AlphaEvolve, whereas Qwen3-4B gets closer however does not match. In Trimul, there is a significant gap between the smaller model performance and the base model performance.
>
> Overall, the quality of the base model is quite important as in any application of reinforcement learning. We will make sure to make this point clear in the revision.
>
> | Method | Model | Erdos |
> | --- | --- | --- |
> | best human | - | 0.380927 |
> | AlphaEvolve | Gemini-2.0 Pro + Flash | 0.380924 |
> | AlphaEvolve V2 | Gemini-2.0 Pro + Flash | 0.380924 |
> | OpenEvolve | gpt-oss-120b | 0.380965 |
> | Best-of-25600 | gpt-oss-120b | 0.380906 |
> | TTT-Discover | Qwen3-4B-Instruct-2507 | 0.381115 |
> | TTT-Discover | Qwen3-8B | 0.380920 |
> | TTT-Discover | gpt-oss-120b | 0.380876 |
>
> | Method | Model | TriMul H100 |
> | --- | --- | --- |
> | 1st human | - | 1371.1 |
> | 2nd human | - | 2368 |
> | 3rd human | - | 2545.7 |
> | 4th human | - | 3654.8 |
> | 5th human | - | 4233.1 |
> | Best-of-25600 | gpt-oss-120b | 5390.3 |
> | TTT-Discover | Qwen3-4B-Instruct-2507 | 9726 |
> | TTT-Discover | Qwen3-8B | 9729 |
> | TTT-Discover | gpt-oss-120b | 1161.2 |
>
>
> > **It is unclear what the model actually learns through test-time training. For a 120B model, the validity rate is likely already very high, which means it rarely generates invalid code…** and **Does RL teach the model new algorithmic strategies, or does it primarily improve the model's ability to refine solutions provided by PUCT?**
>
> On the earlier point, the validity rate / correctness rate in fact increases significantly during training. Below are the numbers during the Trimul training job, where you could see a significant increase (from ~45% to ~80%).
>
> | Step | Validity Rate |
> | --- | --- |
> | 0 | 0.462890625 |
> | 1 | 0.49609375 |
> | 2 | 0.453125 |
> | 3 | 0.380859375 |
> | 4 | 0.455078125 |
> | 5 | 0.380859375 |
> | 10 | 0.732421875 |
> | 20 | 0.669921875 |
> | 30 | 0.6328125 |
> | 40 | 0.78125 |
> | 49 | 0.779296875 |
>
>
> On one hand, these numbers demonstrate in fact the model learns to generate valid kernels. Also qualitatively, we provide observations from the training run to discuss what the model has learned. When comparing the early kernels from the base model to the later kernels from the fully finetuned model, the base model’s implementation is slower because it focuses on optimizing an isolated portion of the computation, whereas the finetuned model takes a more holistic approach to the full execution pipeline without errors. While the base model introduces a fused kernel for projection and gating, it leaves other costly steps such as normalization, output gating, and final projection as separate operations, increasing kernel launch overhead and memory traffic. It also produces intermediates in layouts that are not well aligned with downstream operations, leading to additional data movement. In contrast, the finetuned model fuses more stages and organizes data to better match subsequent computations, reducing both memory bandwidth demands and execution overhead. Overall, the finetuned model reflects a stronger understanding of system level optimization, particularly in structuring computations, managing memory layout, and ensuring efficient interaction across the pipeline.
>
> **Overall,** we truly appreciate your insightful questions, and we hope these answer your questions and concerns. We are happy to follow up during the discussion period, and it would mean a lot to us if you could consider increasing your score.

---

> > ### Author Rebuttal · Reviewer_p3XR · 2026-04-03
> >
> > Interesting points! I will keep my score to be Accept.

---

> > > ### Author Response · Authors · 2026-04-03
> > >
> > > Thank you for your kind remarks and support, and also for helping us improve our paper.
> > > We are happy that you maintain your strong support, it will help our work reach the audience. It would mean a lot to us if you could consider raising your score in light of your resolved concerns and our improvements.

---

### Decision · Program_Chairs · 2026-04-30

**Decision:**

Accept (spotlight)

**Comment:**

TTT-Discover presents a well-motivated and conceptually clean framework for scientific discovery via test-time RL, distinguishing itself from standard RL by targeting maximum rather than expected reward. The empirical results are exceptional — beating AlphaEvolve on open mathematical problems and surpassing top human experts in GPU kernel engineering across five diverse domains. All four reviewers gave Accept (5) scores and fully resolved their concerns after the rebuttal, which provided compelling additional experiments including cost-matched baselines, multi-scale model results, and catastrophic forgetting analysis. I recommend strong accept.